# GaussMark: A Practical Approach for Structural Watermarking of Language Models

Adam Block [1 2]   Alexander Rakhlin [3]   Ayush Sekhari [3 4]

## Abstract

Watermarking, the process by which Large Language Model (LLM) servers imbed an imperceptible signal at inference time in order to detect text generated by their own models, has grown in importance due to the significant improvements in natural language processing tasks by modern LLMs. Current approaches are often impractical due to generation latency, detection time, degradation in text quality, or robustness; such problems often arise due to the focus on *token-level* watermarking, which ignores the inherent structure of text. In this work, we introduce a new scheme, GaussMark, that is simple and efficient to implement, has formal statistical guarantees, comes at no cost in generation latency, and embeds the watermark into the weights of the model itself, providing a *structural* watermark. Our approach is based on Gaussian independence testing and is motivated by recent empirical observations that minor additive corruptions to LLM weights can result in models of identical (or even improved) quality. We provide formal statistical bounds on the validity and power of our procedure and, through an extensive suite of experiments, demonstrate that GaussMark is reliable, efficient, relatively robust to corruption, and can be instantiated with essentially no loss in model quality.

## 1. Introduction

Recent advances in Language Models (LMs) have significantly transformed the field of Natural Language Processing, offering unprecedented capabilities in generating high-

quality artificial text (Achiam et al., 2023; Schulman et al., 2022). While LLMs provide many benefits (He-Yueya et al., 2023; Ahn et al., 2024; Jiang et al., 2024; Imani et al., 2023; Wang et al., 2023), their rapid evolution and widespread deployment pose significant risks to social, political, and economic institutions (Mirsky et al., 2023). In particular, the ability of LLMs to produce human-like text has raised concerns about potential misuse, including academic plagiarism (Kasneci et al., 2023) and the spread of tailored misinformation (Islam et al., 2020).

To address these issues, a substantial body of research has focused on *watermarking* LMs, i.e. embedding a signal into generated text that the provider can use to determine whether or not a given example was machine-generated. In order for watermarking to be practical, it must possess a few key properties: (a) watermarks should be easily detectable with formal statistical guarantees; (b) watermarking should not degrade LM performance either in quality or latency; and (c) watermarks should remain detectable after common token- and sequence-level corruptions are applied. Among these desiderata, formal statistical guarantees on the validity of the watermark are particularly important in situations where authorship claims are sensitive, with a recent example being a lawsuit against a Massachusetts school by a student's parents after the school wrongly accused a student of cheating with ChatGPT (Tenbarge, 2024); while such claims do not always rise to the level of litigation, examples of students falsely accused of cheating due to unreliable detection abound (Jimenez, 2023; Klee, 2023; Giray, 2024; Mathewson, 2023; Verma, 2023). As such, watermarking schemes relying on heuristics or trained detectors, such as Dathathri et al. (2024), cannot be reliably deployed.

Prior works have proposed a plethora of watermarking schemes in recent years, discussed in detail in Appendix B. While some, such as that of Dathathri et al. (2024), allow for efficient and practical watermarking, in order to maintain high quality text, they require storage scaling linearly with the number of generated tokens, which can be impractical in large production systems. Others achieve rigorous guarantees for watermarking by embedding statistical patterns into the autoregressive next-token sampling process of language models and further emphasize *indistinguishability*

---

[1]Microsoft Research, New York, USA [2]Department of Computer Science, Columbia University, New York, USA [3]Department of Computer Science, MIT, Cambridge, MA, USA [4]Boston University, MA, USA. Correspondence to: Adam Block <adam.block@columbia.edu>, Ayush Sekhari <sekhari@mit.edu>.

*Proceedings of the $42^{nd}$ International Conference on Machine Learning*, Vancouver, Canada. PMLR 267, 2025. Copyright 2025 by the author(s).

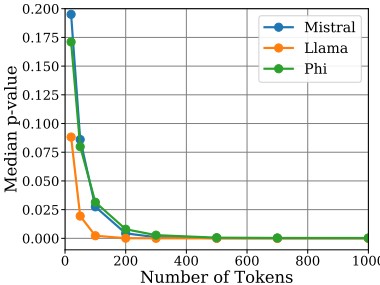

(a) The median p-value of GaussMark averaged on 1K watermarked generations.

---

**Input Prompt**: Sci-Fi fans will rejoice with today's news as details about the upcoming game, Doctor Who: The Etern

**Base (unwatermarked) Llama3.1-8B Model**: ity Clock have just been revealed. Both iOS and Android versions of the game are set to be released in Spring 2012. Also set to be released in 2012 is the much anticipated Movie The Dark Knight Rises and the new Dr. Who series. The official game synopsis, gameplay video, and screenshots follow. Wired, popular UK gadget site, has leaked some screenshots of the upcoming Android OS. According to the leaked screenshots Android 4.0 will feature an updated logo that looks to borrow a bit from HTC's Sense user interface . . .

**Watermarked Llama3.1-8B Model**: als, have been revealed. Cryptic Games have collaborated with BBC Studios to bring a life-like game based on the BBC's popular Doctor Who. Doctor Who: The Eternals is a massively multiplayer online role-playing game based in the Doctor Who universe and is being developed by Cryptic Games, the makers of Neverwinter and Lord of the Rings Online. At the heart of the game are the Eternals, extra-dimensional beings that played a pivotal role in several key stories from Doctor Who. Among the Eternals are creatures known as the Oncoming Storm, whose mysterious plans for the universe have resulted in alien invasions and apocalyptic destruction . . .

(b) Example watermarked and unwatermarked text.

*Figure 1.* Demonstration of the efficacy of GaussMark on C4 prompts (a) and a of snippet watermarked text (b). The full text completions are given in Appendix J.1

or *distortion-freeness* (Kuditipudi et al., 2023; Christ et al., 2024b; Golowich and Moitra, 2024), requiring the watermarked text to closely resemble unwatermarked text in Total Variation (TV) distance. However, this approach overlooks the inherent structure of language, treating text as merely a sequence of tokens generated by an unstructured autoregressive process. While this focus on distortion-freeness helps safeguard text quality, it often comes at the cost of other crucial properties, such as efficiency; for example Kuditipudi et al. (2023), exhibits extremely slow generation and detection. Fortunately, such stringent distortion-freeness is not necessary in practice, as the primary goal is to generate text that appears indistinguishable from unwatermarked text to humans. Recent empirical evidence suggests that humans have limited ability to differentiate between high-quality generated text and subtle variations thereof (Dugan et al., 2023; Clark et al., 2021; Ippolito et al., 2019; Elangovan et al., 2024; Lee et al., 2022), suggesting that instead of enforcing approximation in TV, weaker metrics like Maximum Mean Discrepancy (MMD) (Gretton et al., 2012), evaluated using the implicit criteria humans employ to assess text quality, may be sufficient. Furthermore, natural language exhibits intricate implicit structures that modern language models learn and encode within their weights. While some recent work has introduced distortionary watermarks (Kirchenbauer et al., 2023a;b), they tend to underperform

empirically when assessed on the quality of the generated text. Perhaps most similar to the current paper is the concurrent work of Christ et al. (2024a), which proposes adding a Gaussian vector to the bias in the final layer of an LM in an effort to ensure the watermark is difficult to remove even after open-sourcing the model. While the focus of that work is not primarily on formal statistical guarantees of the resulting watermark, they demonstrate that under natural assumptions on the access an adversary has, it is challenging to remove the embedded watermark without degrading the quality of the model; such results likely also apply to our proposed approach.

In this work, we propose GaussMark, a novel approach that takes advantage of the structure inherent to text by modifying the model weights through a Gaussian perturbation at generation time and tests for statistical independence at detection time. Our approach is motivated by empirical observations that language models are robust to small perturbations of their weights, as evidenced by phenomena such as linear interpolation through model merging and model fine-tuning by thresholding or modifying weights along low-rank components, which does not degrade output quality (Sharma et al., 2024; Sun et al., 2023; Hu et al., 2021). Our method is simple to implement and efficient, providing both significant detection ability and essentially no loss in model performance. Our key contributions are:

- In Section 3, we introduce GaussMark, a novel watermarking scheme that is practical, efficient, and has no effect on generation latency. Therein, we motivate GaussMark using classical statistical theory, and provide rigorous guarantees on the statistical validity of GaussMark under essentially no assumptions. We then theoretically analyze the power of GaussMark to be detected under modeling assumptions approximating modern language models.

- In Section 4, we describe key empirical results providing a comprehensive evaluation of GaussMark on a variety of modern LMs, demonstrating its detectability and lack of effect on model quality.

- We supplement our empirical results in the main body with an extensive exploration of (i) ablations investigating the effects that different parameter choices have on both detectability and model quality (Appendix D and Appendix E); (ii) demonstrations of the relative robustness of GaussMark to token- and sequence-level corruptions (Appendix F); (iii) an investigation of a low-rank variant of GaussMark that further reduces the watermark's impact on model quality (Appendix G); and (iv) a comparison of GaussMark to the scheme of Kirchenbauer et al. (2023a;b), demonstrating that while GaussMark can be less detectable and robust to corruptions, it has a much smaller impact on the quality of the generated text and

allows for significantly faster generation (Appendix H).

# 2. Problem Setup and Prerequisites

## 2.1. Hypothesis Testing

We begin by recalling the formalism of hypothesis testing from classical statistics (see Casella and Berger (2001); Lehmann et al. (1986) for a thorough introduction).

**Definition 2.1.** Given a domain $\mathcal{Z}$, hypotheses $\mathbf{H_0}$ and $\mathbf{H_A}$ are nonempty collections of probability measures on $\mathcal{Z}$, with $\mathbf{H_0}$ called the *null hypothesis* and $\mathbf{H_A}$ called the *alternative hypothesis*. If $\mathbf{H_0}$ (or $\mathbf{H_A}$) is a singleton, we call it *simple*; otherwise, we call it *composite*. A *test* is any measurable function $\rho : \mathcal{Z} \to \{0, 1\}$.

We assume there exists a ground-truth distribution $p \in \mathbf{H_0} \cup \mathbf{H_A}$ and, given a sample $Z \sim p$, we wish to use the test $\rho$ to inform us as to whether we can confidently say that $p \in \mathbf{H_A}$ (i.e. $\rho(Z) = 1$) or if we, by default, do not have sufficient evidence to discount the hypothesis that $p \in \mathbf{H_0}$ (i.e. $\rho(Z) = 0$). As such, there are two relevant notions of error: the probability that we *falsely reject* the null hypothesis (Type-I error) and the chance that we *falsely accept* the null hypothesis (Type-II error). These errors are controlled by the *level* and the *power* of test $\rho$.

**Definition 2.2.** Let $\alpha, \beta \in (0, 1)$. Given hypotheses $\mathbf{H_0}$ and $\mathbf{H_A}$, and a test $\rho : \mathcal{Z} \to \{0, 1\}$, we say that $\rho$ has *level* $\alpha$ if $\sup_{p \in \mathbf{H_0}} \Pr_{Z \sim p}(\rho(Z) = 1) \le \alpha$, i.e., for any distribution in $\mathbf{H_0}$, the probability we falsely classify a sample from that distribution as coming from a distribution in $\mathbf{H_A}$ is at most $\alpha$. We say that $\rho$ has *power* $1 - \beta$ if $\sup_{p \in \mathbf{H_A}} \Pr_{Z \sim p}(\rho(Z) = 0) \le \beta$, i.e., the probability that we accept the null hypothesis despite the fact that the true distribution is in $\mathbf{H_A}$ is bounded above by $\beta$. Finally, a function $p : \mathcal{Z} \to [0, 1]$ is called a *p-value* if for all $\alpha \in [0, 1]$ it holds that $\sup_{q \in \mathbf{H_0}} \Pr_{Z \sim q}(p(Z) \le \alpha) \le \alpha$.

We aim to design tests $\rho$ that maintain a low probability of rejecting the null hypothesis when it is true (i.e. $\alpha \approx 0$) while achieving a high probability of rejecting the null hypothesis under the alternative hypothesis (i.e. $\beta \approx 0$). The p-value is a crucial tool in hypothesis testing, as it provides a measure of evidence against the null hypothesis. Specifically, it represents the probability of obtaining a result at least as extreme as the one observed, assuming the null hypothesis is true. Thus, smaller p-values imply that $\mathbf{H_0}$ is less likely.

## 2.2. Language Models

A Language Model (LM), parameterized by weights $\theta \in \Theta$ and a token space $\mathcal{V}$, is represented by a function $p_\theta : \mathcal{V}^\star \to \Delta(\mathcal{V})$ that maps a sequence of past tokens to a distribution over the next-token, where $\mathcal{V}^\star$ is the set of all strings of tokens in $\mathcal{V}$. Given a prompt $x \in \mathcal{V}^\star$ and a generation length

$T \ge 1$, the response text $y = (v_1, \dots, v_T) \in \mathcal{V}^T$ is generated by autoregressively sampling from the language model for $T$ tokens using the process $v_t \sim p_\theta(\cdot \mid x \circ v_{1:t-1})$ for $t \in [T]$. We often use the notation $y = v_{1:T} \sim p_\theta(\cdot \mid x)$ to represent the above sampling process.

## 2.3. Watermarking Text Generated by LM

We next formalize watermarking as a hypothesis testing problem, a framework that has also been used in prior rigorous watermarking works (Huang et al., 2023; Kuditipudi et al., 2023; Kirchenbauer et al., 2023a). Formally, a watermarking scheme consists of Generate and Detect procedures. Given a prompt space $\mathcal{X} \subseteq \mathcal{V}^\star$, a key space $\Xi \in \mathbb{R}^d$, and a sample space $\mathcal{Y} \subseteq \mathcal{V}^T$ (consisting of strings from the token space), watermarking takes place in two steps:

1. Generate is parameterized by a language model $p_\theta : \mathcal{X} \to \Delta(\mathcal{Y})$ and a distribution $\nu \in \Delta(\Xi)$. When given a prompt $x \in \mathcal{X}$, Generate samples a key $\xi \sim \nu$ and returns the watermarked text $y \sim p_{\theta(\xi)}(\cdot \mid x)$, where $\theta(\xi)$ denotes the model watermarked using the key $\xi$.

2. Detect takes a watermarking key $\xi' \in \Xi$, model $p_\theta$, and some candidate text $y' \in \mathcal{Y}$ then tests the following:

$\mathbf{H_0}$ : Given $x$, the key and the sample are independent, i.e., $(\xi', y') \sim \nu \otimes q$ with $q \in \Delta(\mathcal{Y})$.

$\mathbf{H_A}$ : The sample $y'$ is produced by Generate using the key $\xi'$, i.e., $y' \sim p_{\theta(\xi')}(\cdot \mid x)$.

Our goal in designing a watermarking scheme is to choose $\nu$ such that for any $x$, the modified language model $p_{\theta(\xi)}$ generates text that is sufficiently distinct from that of the original language model $p_\theta$ when the key $\xi$ is known, but at the same time has good quality and is relatively indistinguishable when $\xi$ is unknown. We emphasize that the null hypothesis considered above does *not* assume that $y' \sim p_\theta(\cdot \mid x)$, as this would allow for human-generated text to be classified as machine-generated as long as the human's distribution is sufficiently different from the model's. Instead, $\mathbf{H_0}$ is a *composite* hypothesis that places no structural assumption on the distribution $Q$ of responses and thus encompasses text generated by humans or even other LMs. We note that this formulation is similar to that of Huang et al. (2023), except that we drop the additional requirement in that work that the marginal of $y$ in $\mathbf{H_A}$ is $\epsilon$-close in TV to the original LM's distribution; as we stated above, we believe TV to be an unnecessarily strong metric for bounding degradation in model quality.

# 3. Watermarking Scheme: GaussMark

Our scheme, GaussMark, applies to an arbitrary parameterized model, where $p_\theta : \mathcal{V}^\star \to \Delta(\mathcal{V}^T)$ is a distribution with parameters $\theta \in \Theta \subset \mathbb{R}^d$. In GaussMark.Generate

**Algorithm 1** GaussMark.Generate

1: **Input** Language Model $p_\theta : \mathcal{V}^\star \to \Delta(\mathcal{Y})$ with parameters $\theta \in \Theta$, prompt $x$, variance $\sigma^2 > 0$.
2: **Sample** watermark key $\xi \sim \mathcal{N}(0, \sigma^2 \mathbf{I})$.
3: **Watermark** model by setting $\theta(\xi) = \theta + \xi$.
4: **Generate** using the perturbed model: $y \sim p_{\theta(\xi)}(\cdot \mid x)$.

(Algorithm 1), we generate the watermarking key $\xi$ by sampling from a centered multivariate Gaussian distribution with covariance $\sigma^2 \mathbf{I}$ for some small $\sigma > 0$, i.e. $\xi \sim \nu$ with $\nu = \mathcal{N}(0, \sigma^2 \mathbb{I}_d)$. We then produce the watermarked model $\theta(\xi)$ by additively perturbing $\theta$ with the key $\xi$, i.e. $\theta(\xi) = \theta + \xi$. On the given input prompt $x$, we then generate the watermarked response by sampling $y \sim p_{\theta(\xi)}(\cdot \mid x)$.

**Algorithm 2** GaussMark.Detect

1: **Input** Language Model $p_\theta : \mathcal{V}^\star \to \Delta(\mathcal{Y})$ with parameters $\theta \in \Theta$, prompt $x$, variance $\sigma^2 > 0$, watermark key $\xi$, candidate text $y \in \mathcal{Y}$, level $\alpha \in (0, 1)$.
2: **Evaluate** the test statistic

$$\psi(y, \xi \mid x) = \frac{\langle \xi, \nabla_\theta \log p_\theta(y \mid x) \rangle}{\sigma \|\nabla_\theta \log p_\theta(y \mid x)\|}. \qquad (1)$$

3: **Return** 'Watermarked' if $\psi(y, \xi \mid x) \geq \Phi^{-1}(1 - \alpha)$; **else** fail to reject null.

In GaussMark.Detect (Algorithm 2), we test the composite hypothesis $\mathbf{H_0}$, that $\xi \sim \mathcal{N}(0, \sigma^2 \mathbf{I})$ and $y \sim q \in \Delta(\mathcal{Y})$ are independent of each other, against the simple alternative hypothesis that $y \sim p_{\theta+\xi}(\cdot \mid x)$ using the test statistic $\psi(y, \xi \mid x)$ given in (1). Letting $\Phi$ denote the *cumulative distribution function (CDF) of a standard Gaussian random variable*, we reject the null hypothesis (determine the text is watermarked) if $\psi(y, \xi \mid x) \geq \Phi(1 - \alpha)$, where $\alpha$ denotes the acceptable false positive rate (Type-I error). Although GaussMark.Detect is formulated as a statistical test, in our experiments we report the $p$-value, $1 - \Phi(\psi(y, \xi \mid x))$.

It is critical that our test is valid for the composite null hypothesis $\mathbf{H_0}$, allowing any $q \in \Delta(\mathcal{Y})$ without imposing assumptions on the distribution of $y$ under the null. A test that is valid only for a simple null, such as $y \sim p_\theta(\cdot \mid x)$, against $\mathbf{H_A}$ may fail to have a small level when applied to arbitrary human-generated text that does not originate from the model $p_\theta$, which would undermine the key benefit of a watermarking scheme with formal statistical guarantees. As explained in the sequel, the choice of Gaussian $\xi$ is motivated by the fact that Gaussians (a) are rotationally invariant and (b) have independent directions and norms. In principle, any distribution with these properties could replace the Gaussian, and with minor modifications, the test would still maintain statistical validity. We provide an illustrative example in Appendix I.1.

**Practical Implementation.** Modern language models are parameterized by billions of parameters (Kaplan et al., 2020; Touvron et al., 2023) and, while GaussMark could be implemented across the entire model, we find that effective watermarking can be achieved by modifying a single MLP weight (a matrix) within a single layer. This approach offers two key advantages: First, perturbing parameters in a single feedforward layer minimizes potential degradation in model quality. Specifically, GaussMark relies on the statistical distinguishability of the distributions $p_{\theta+\xi}(\cdot \mid x)$ and $p_\theta(\cdot \mid x)$, which we observe can be achieved without compromising model performance. Recent empirical studies demonstrate that much of the signal in the feedforward layers of language models resides in a low-dimensional manifold. Consequently, any added noise $\xi$ to a single layer is likely almost orthogonal to the underlying signal, preserving model quality. Second, GaussMark.Detect incurs minimal memory and computational overhead as it requires only a single forward pass through the model and a backward pass through the watermarked parameter $\theta$. When $\theta$ is a single MLP layer, the memory cost of taking a gradient is negligible compared to the memory demands of a forward pass. In any case, simplicity of our approach allows seamless integration into generation pipelines such as vLLM (Kwon et al., 2023) and HuggingFace (Wolf et al., 2020b).

### 3.1. Motivation

Recall from Definition 2.2 that there are two sources of error in hypothesis testing: false positives (controlled by the level of the test) and false negatives (controlled by the test's power). Because these two desiderata are at odds, statisticians either wish to find tests with maximal power subject to a constraint on the level (Neyman and Pearson, 1933) or wish to control the *risk* of a test, defined for weights $a, b \in \mathbb{R}_{\geq 0}$ as $a\alpha + b\beta$. In either case, as long as the null hypothesis is convex, under general measure theoretic conditions (cf Lehmann et al. (1986, Theorem 3.8.1) for the Neyman-Pearson approach and Le Cam (2012); Ghosal and van der Vaart (2017) for the case of minimax risk), the optimal test is given by a likelihood ratio test with respect to the 'worst-case' distribution in $\mathbf{H_0}$, which can be shown in special cases to be that in $\mathbf{H_0}$ which is closest in TV to $\mathbf{H_A}$. Thus finding the optimal test reduces to finding the 'worst' $p \in \mathbf{H_0}$ and rejecting $\mathbf{H_0}$ if $\log p_{\mathbf{H_A}}/p$ is large. Unfortunately, in many settings, it is not clear how to characterize this $p$ in a way that is amenable to computing $p$-values or the thresholds for test statistics. On the other hand, if we relax projection in TV to projection in KL divergence, a standard calculation shows that the the closest distribution in $\mathbf{H_0}$ to $\mathbf{H_A}$ is given by marginalizing over $\xi$, i.e. $\nu \otimes \mathbb{E}_{\xi' \sim \nu}[p_{\theta+\xi'}(\cdot \mid x)]$ (cf. Proposition I.2 in Appendix I.1). Heuristically, then, a reasonable candidate for the optimal

test between $\mathbf{H_0}$ and $\mathbf{H_A}$ would be to reject the null when the log-likelihood ratio of the data under the null and the alternative is larger than some threshold $\tau_\alpha$ satisfying

$$\sup_{p \in \mathbf{H_0}} \mathbb{P}_p \left( \log \left( \frac{p_{\theta+\xi}(y \mid x)}{\mathbb{E}_{\xi \sim \nu}[p_{\theta+\xi}(y \mid x)]} \right) > \tau_\alpha \right) \le \alpha. \quad (2)$$

While (2) would yield a valid (although possibly suoptimal) test, it is not obvious how to easily compute the threshold $\tau_\alpha$, nor is it clear that computing the second term in the test statistics is feasible due to the scales of the probabilities involved. In order to circumvent these problems, we approximate the log-likelihood test to first order, specialize to the case that $\nu = \mathcal{N}(0, \sigma^2 \mathbf{I})$, and further suppose that $\sigma \ll 1$ so as to ignore terms scaling with $\sigma^2$ (see Appendix I.1 for details); we then have

$$\log\big(p_{\theta+\xi}(y|x) \big/ \mathbb{E}_{\xi' \sim \nu}[p_{\theta+\xi'}(y|x)]\big) \approx \langle \xi, \nabla_\theta \log p_\theta(y \mid x) \rangle. \quad (3)$$

Thus, under the above linearization approximation, the statistic $\langle \xi, \nabla_\theta \log p_\theta(y \mid x) \rangle$ yields a test approximately as powerful as that in (2) that is easily computed, solving the second problem above. The key insight in solving the first problem and determining $\tau_\alpha$ is, due to the rotation invariance of the Gaussian and the fact that the direction and norm of Gaussian vectors are independent, the distribution of $\psi(y, \xi \mid x)$, given in (1), is *always a standard normal under* $\mathbf{H_0}$ because $y$ is independent of $\xi$; thus with appropriate normalization, $\tau_\alpha$ can be read off from the Gaussian cumulative distribution function. This insight is formalized in the following proposition wherein we show that GaussMark, delivers statistically valid p-values, and is even an *exact test*, ensuring provable control of the false positive rate under virtually no assumptions.

**Proposition 3.1.** *Let $\alpha \in (0, 1)$, and $\tau_\alpha := \Phi^{-1}(1 - \alpha)$ where $\Phi$ is the CDF of the standard normal distribution. Then, for any $x \in \mathcal{X}$, the test $\mathbb{I}\left\{ \frac{\langle \xi, \nabla \log p_\theta(y|x) \rangle}{\sigma \|\nabla \log p_\theta(y|x)\|} \ge \tau_\alpha \right\}$ in Algorithm 2 has level $\alpha$. Furthermore, $1 - \Phi(\psi(y, \xi \mid x))$ is a valid p-value for the test.*

### 3.2. Bounding the Power of GaussMark

While *statistical validity holds unconditionally, we must make some assumptions on the model in order to guarantee any nontrivial power*. Indeed, in the trivial case where $p_\theta$ is constant in $\theta$ and $\nabla \log p_\theta(\cdot \mid x) = 0$ identically, we clearly cannot hope to achieve any watermarking detection with GaussMark as all models are identical. Motivated by our application to LMs, we consider the following setting.

**Definition 3.2** (Linear softmax model). Let $\mathcal{X}$ and $\mathcal{Y}$ be discrete spaces, $\theta \in \mathbb{R}^d$ be the underlying model's parameters, and $\varphi : \mathcal{Y} \times \mathcal{X} \to \mathbb{R}^d$ a known feature map. We say $p_\theta(\cdot \mid x)$ is a *linear softmax model* if, for all $y \in \mathcal{Y}$, $p_\theta(y \mid x) = e^{\langle \theta, \varphi(y|x) \rangle} \big/ \sum_{y' \in \mathcal{Y}} e^{\langle \theta, \varphi(y'|x) \rangle}$.

In practice, we watermark by adding the noise $\xi$ to a feedforward layer in a single transformer block (see Section 4 for exact implementation details). The above linear softmax model is motivated by the empirical observation that LM's are well-behaved in a small neighborhood around a trained model's parameters, and can thus be locally approximated via a first-order Taylor's expansion (cf. Appendix I.3 for further discussion). Recent empirical findings on model merging via linear interpolation of weights provide further support to the assumption that the model behaves linearly in a small neighborhood around the trained parameters (Wortsman et al., 2022; Chronopoulou et al., 2023; Yang et al., 2024; Goddard et al., 2024; Ilharco et al., 2022; Yu et al., 2024). The preceding reasoning suggests that at least *next-token prediction* can be approximated by a linear softmax model, but a key benefit of the same is that the property of being such a model is preserved under autoregressive generation (cf. Appendix I.3.1). This intuition suggests that our theoretical results hold only for sufficiently small $\sigma$. Indeed, as shown in our experiments, when $\sigma$ becomes too large, GaussMark starts to lose its power (cf. Appendix D). We next bound the power of our test:

**Proposition 3.3.** *Let $\alpha \in (0, 1)$ and $\tau_\alpha = \Phi^{-1}(1 - \alpha)$. Let $p_\theta$ be given by a linear softmax model w.r.t. some underlying features $\varphi$. Then, for any $x \in \mathcal{X}$, the hypothesis test $\mathbb{I}\left\{ \frac{\langle \xi, \nabla \log p_\theta(y|x) \rangle}{\sigma \|\nabla \log p_\theta(y|x)\|} \ge \tau_\alpha \right\}$ from Algorithm 2, is level $\alpha$ and has power $1 - \beta$, where*

$$\beta = \mathbb{E}_{\substack{y \sim p_\theta \\ \xi \sim \mathcal{N}(0, \sigma^2 I)}} \left[ \gamma(y, \xi \mid x) \cdot \mathbb{I}\{\psi(y, \xi \mid x) \le \tau_\alpha\} \right], \quad (4)$$

*with $\gamma(y, \xi \mid x) := e^{\langle \xi, \nabla \log p_\theta(y|x) \rangle} \big/ \mathbb{E}_{y \sim p_\theta}\left[e^{\langle \xi, \nabla \log p_\theta(y|x) \rangle}\right]$.*

We remark that our computations for the bounds on the level and the power of our test are valid for any fixed $x \in \mathcal{X}$, but are marginalized over the key $\xi$, as is common in other statistically valid watermarks (Christ et al., 2024b; Kuditipudi et al., 2023; Golowich and Moitra, 2024). The multiplicative factor of $\gamma(y, \xi \mid x)$ in (4) is a reweighing that tilts the underlying distribution $y \sim p_\theta(\cdot \mid x)$ to prefer those $y$ for which $\langle \xi, \nabla \log p_\theta(y \mid x) \rangle$ is large. On the other hand, the indicator term is non-zero only when $\psi(y, \xi \mid x)$, which depends on $\langle \xi, \nabla \log p_\theta(y \mid x) \rangle$, is smaller than $\tau_\alpha$; thus, in general, we should expect the power of the test to increase (i.e. $\beta$ to be smaller) as we place more mass on those $y$ for which $\langle \xi, \nabla \log p_\theta(y \mid x) \rangle$ is large. While intuitive, this logic suffers from the fact that if $\|\nabla \log p_\theta(y \mid x)\|$ can vary widely depending on $y$, those responses maximizing $\langle \xi, \nabla \log p_\theta(y \mid x) \rangle$ may not be the same as those maximizing $\psi(y, \xi \mid x)$ due to the invariance of the latter, but not the former, to the norm; this is precisely the price we pay in moving from (3) to (1) by normalizing.

To better understand the bound on the power of the test in Proposition 3.3, consider the simplified setting where

$\sigma \to \infty$, and suppose that there exists $R, r > 0$ such that $r \le \|\nabla \log p_\theta(y \mid x)\| \le R$ for all $y \in \mathcal{Y}$ and $x \in \mathcal{X}$. In this case, for any fixed $\xi$, the exponential tilting places all of the mass on $y^\star(\xi) := \operatorname{argmax}_y \langle \xi, \nabla \log p_\theta(y \mid x) \rangle$, and a few simple calculations imply that

$$\beta = \Pr_{Z \sim \mathcal{N}(0,\mathbf{I})} \left( \sup_{y \in \mathcal{Y}} \langle Z, \nabla \log p_\theta(y \mid x) \rangle \le \tau_\alpha \cdot R/r \right).$$

Thus, the power of the test depends on the distribution of $\sup_y \langle Z, \nabla \log p_\theta(y \mid x) \rangle$ for $Z \sim \mathcal{N}(0, \mathbb{I}_d)$, whose expectation is the Gaussian width of the set $\{\nabla \log p_\theta(y \mid x)\}$, a well-known measure of the complexity of sets in learning theory (Vershynin, 2018). Were the feature set $\{\varphi(y)\}_{y \in \mathcal{Y}}$ sufficiently rich, the vectors $\{\nabla \log p_\theta(y \mid x)\}_{y \in \mathcal{Y}}$ would cover the unit sphere, making this supremum large with high probability (See Appendix I.3 for a discussion on the role of entropy in ensuring such diversity). In this case, we expect $\sup_y \langle Z, \nabla \log p_\theta(y \mid x) \rangle \approx \|Z\|$. Under these simplifications, GaussMark achieves power at least $1 - \beta$ at level $\alpha$, provided $d \gtrsim \log \left( 1/\min\{\alpha, \beta\} \right) \cdot R/r$, which captures the intuition that bigger models are easier to watermark since a larger dimension makes it easier to hide the watermarking signal without hurting the model performance. We observe that across a number of recent LMs, the actual conditioning is less than 50 (cf. Figure 6), suggesting that the above is a reasonable approximation of what is observed in practice. This intuition also aligns with our empirical observations that larger noise generally makes watermarking easier (cf. Appendix D) and longer sequences are easier to watermark: $\|\nabla \log p_\theta(y \mid x)\|$ grows with sequence length (Figure 2) and more choices of $y$ increase the likelihood that $\{\nabla \log p_\theta(y \mid x)\}$. Moreover, this framework is consistent with earlier work that shows that higher-entropy sequences are fundamentally easier to watermark (Kuditipudi et al., 2023; Fairoze et al., 2023; Christ et al., 2024b; Golowich and Moitra, 2024; Huang et al., 2023), since $\|\nabla \log p_\theta(y \mid x)\|$ tends to increase with entropy. We formalize this intuition in Corollary I.5 (in Appendix I.3.1) under mild assumptions.

In addition to linear softmax models, we can also control the power of GaussMark in significantly greater generality. In Appendix I.3.2, we provide a bound on the power of the test when the distribution $p_\theta$ is strongly log-concave. Our primary detection method in Algorithm 2 empirically exhibits some robustness against modifications, including insertions, deletions, substitutions, and roundtrip translations (cf. Figures 13 and 14). In Appendix F, we provide a theoretical analysis (cf. Proposition I.9 of the robustness of a variant of GaussMark (Algorithm 4) against random token-level corruptions under mild independence assumptions; this approach can also arbitrarily enhance the statistical power of GaussMark as the length of generated text increases.

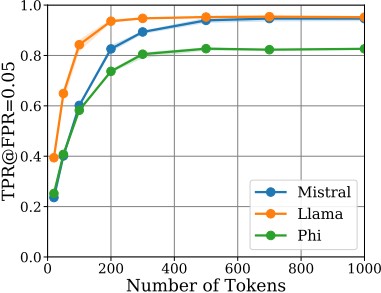

(a) Fraction of detected watermarked responses at $p = 0.05$ (averaged over 3 seeds)

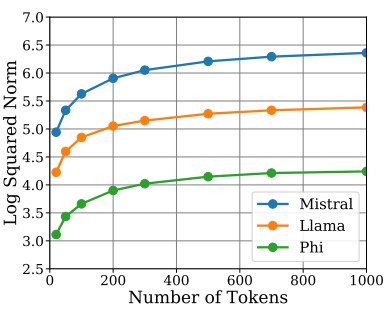

(b) $\|\nabla \log p_\theta(y)\|^2$ vs. number of tokens

*Figure 2.* Effect of length of generated token sequence on p-values and gradient norm for GaussMark.

## 4. Experiments

In this section, we empirically validate our watermarking scheme. In our experiments, we use the HuggingFace repository (Wolf et al., 2020a) to load our models, vLLM (Kwon et al., 2023) for generation, and PyTorch (Paszke et al., 2019) for watermark detection. For examining the quality of our watermarked models, we make use of EleutherAI's lm-evaluation-harness (Gao et al., 2024) as well as the AlpacaEval framework of Li et al. (2023b). All of our experiments are run on 40GB NVIDIA A100 GPUs, with each model chosen sufficiently small that a single GPU suffices for both generation and detection.

**Key Experimental Details.** The GaussMark procedure described in Algorithms 1 and 2 can be used essentially out of the box, requiring careful selection of two key hyperparameters: the variance of the Gaussian noise, $\sigma$, and the specific parameter $\theta$. Statistical validity (Proposition 3.1) holds independent of $\theta$ and $\sigma$, whereas Proposition 3.3 (our power bound) is mostly agnostic to the precise $\theta$ and suggests that $\sigma$ should be large subject to the linear approximation remaining sound. Selecting these hyperparameters requires careful consideration to ensure $p_{\theta+\xi}$ is sufficiently distinct from $p_\theta$ for GaussMark to detect while avoiding adverse impact on generated text quality. As such, the choice of these parameters is a careful empirical balancing act. For each of

the language models under consideration, we choose $\theta$ to be a single MLP layer (cf. Appendix C). The choice of both $\theta$ (which can vary over layer and MLP parameter within the architecture) and $\sigma$ is made by maximizing detectability subject to the evaluation metrics of the model remaining on par with those of the unwatermarked counterpart.

### 4.1. Can GaussMark be Detected?

As has become standard in recent empirical evaluations of watermarking (Kirchenbauer et al., 2023a; Kuditipudi et al., 2023; Lau et al., 2024; Pan et al., 2024), we use the `realnewslike` split of the C4 dataset (Raffel et al., 2020) as prompts for generation. We use the same 1K prompts for all models and all watermarking keys in order to make the comparison fair. We consider three recent LMs: Llama3.1–8B (Dubey et al., 2024), Mistral–7B (Jiang et al., 2023), and Phi3.5–Mini (Abdin et al., 2024). The last, Phi3.5–Mini was instruction fine-tuned and thus performs significantly better on a number of reasoning, math, and coding tasks (Abdin et al., 2024) as well as generating substantially lower entropy sequences, making watermarking more challenging (Kirchenbauer et al., 2023a; Aaronson and Kirchner, 2022; Christ et al., 2024b; Kuditipudi et al., 2023). Thus, we see Phi3.5–Mini as a particularly challenging model to watermark, which shows up empirically in Figures 1 and 2. In Figure 2(b), we see that as the number of generated tokens increases, $\|\nabla \log p_\theta(y \mid x)\|^2$ increases as well; Moreover, comparing plots (a) and (b) shows the strong relationship between gradient norm and $p$-values predicted by our theory.

A notable feature of GaussMark is its remarkable speed in both generation and detection. As shown in Figure 3, generating 1K tokens takes only milliseconds, while detection requires less than a second even for the same token count. Comparing these times to those of Kuditipudi et al. (2023), we see that GaussMark is significantly faster and even has lower latency than reported in Dathathri et al. (2024), which does not have statistical guarantees. We defer further investigation and ablation experiments to Appendix D.

### 4.2. Does GaussMark Degrade Model Quality?

To evaluate the effect of GaussMark on text quality, we employ three benchmarks: SuperGLUE (Wang et al., 2019), GSM–8K (Cobbe et al., 2021), and AlpacaEval–2.0 (Dubois et al., 2024; Li et al., 2023b); in Appendix E, we report the effect that watermarking has on model perplexity of real text. The first benchmark, SuperGLUE, is a collection of eight challenging language understanding and reasoning tasks. The second, GSM–8K is a collection of approximately 8K grade school math questions. For both tasks, we use the LM Evaluation Harness (Gao et al., 2024) to evaluate performance using the Chain of Thought (CoT)

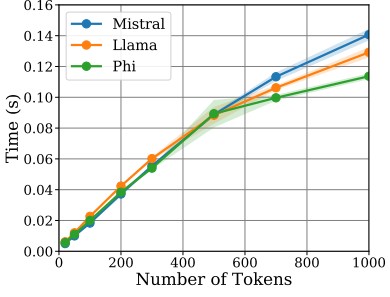

(a) Generation time

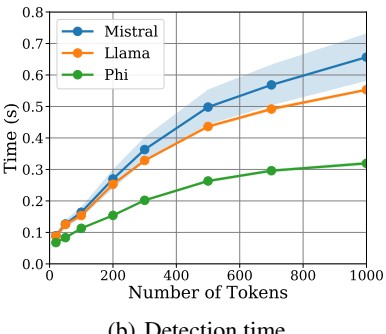

(b) Detection time

*Figure 3.* Latency of GaussMark in seconds for generation (a) and detection (b) as number of generated tokens increases.

and multi-shot prompting provided by Gao et al. (2024) in order to standardize our results. Finally, AlpacaEval–2.0, is a significantly more challenging task, where the candidate and benchmark language models generate responses to approximately 800 questions and a larger 'evaluator' language model (used as a proxy for human preferences) decides which response is better. Instead of comparing win rates against responses generated by humans or a much stronger model, we directly evaluate the win rate of watermarked models relative to un-watermarked models to enhance the signal. For cost efficiency, we use GPT-4o, the current flagship model of OpenAI, as the evaluator, as opposed to the older GPT-4-Turbo model used in Li et al. (2023b).

We report our results in Table 1. For SuperGLUE, we average the scores over the 8 tasks (cf. Appendix E for scores on individual tasks, and further discussion and experiments) and, along with the GSM–8K task, we report mean accuracy and standard error. For AlpacaEval–2.0, we report the win rate of the watermarked model against the un-watermarked model; in order to handle stochasticity, we also consider the evaluator's preferences when comparing the unwatermarked model to itself with different seeds used for generation and report this performance as an interval. We observe that each of the models discussed in Section 4.1 can be watermarked with GaussMark with essentially no loss in performance.

*Table 1.* Performance of GaussMark on various models.

| Model | SuperGLUE (**Avg**) | | GSM–8K (**Acc**) | | AlpacaEval–2.0 (**win rate**) | |
|---|---|---|---|---|---|---|
| | Unwatermarked | GaussMark | Unwatermarked | GaussMark | Unwatermarked | GaussMark |
| Llama3.1–8B | $0.7012 \pm 0.0336$ | $0.7094 \pm 0.0327$ | $0.6300 \pm 0.0133$ | $0.6111 \pm 0.0134$ | (0.46, 0.54) | 0.45 |
| Mistral–7B | $0.6836 \pm 0.0335$ | $0.7033 \pm 0.0332$ | $0.4291 \pm 0.0136$ | $0.4321 \pm 0.0136$ | (0.49, 0.51) | 0.50 |
| Phi3.5–Mini | $0.6451 \pm 0.0254$ | $0.6489 \pm 0.0258$ | $0.8423 \pm 0.0100$ | $0.8552 \pm 0.0097$ | (0.46, 0.54) | 0.49 |

## 4.3. Additional Experiments

We defer full discussion of our other empirical results to the appendix for the sake of space and summarize them here.

**Robustness.** One important characteristic of any deployable watermark is corruption-robustness; in Appendix F we investigate the extent to which GaussMark is robust to token-level additions, deletions, and substitutions as well as a roundtrip translation attack through French. In all cases, we find that GaussMark is somewhat robust to these attacks and is especially robust to corruptions that occur *contiguously*. This latter property is especially important as it demonstrates that, despite the seeming need of GaussMark to know the prompt, which is unrealistic in practical scenarios, this requirement can be dropped with minimal loss of performance.

**Rank-Reduction.** Motivated by recent empirical work suggesting that lower principal components of the weight matrices can be treated as 'noise' and their removal can improve model performance on some tasks (Hu et al., 2021; Sharma et al., 2024; Dettmers et al., 2024; Guo et al., 2023; Han et al., 2024; Wang et al., 2024), in Appendix G we propose a modification of GaussMark, where noise is only added to the bottom $k$ principal components of the watermarked weight. Extensive exploration of the effects of this intervention demonstrates that rank-reduction allows GaussMark to tolerate greater $\sigma$ (and associated detectability) without compromising model quality, although this can occur at the cost of decreased robustness.

**Comparison with Kirchenbauer et al. (2023a).** In Appendix H, we compare GaussMark with the only other practical watermarking scheme with *statistical guarantees on detectability* currently implemented of which we are aware: that of Kirchenbauer et al. (2023a;b). We find that setting the parameters of this other scheme in such a way as to match the generation quality of GaussMark leads to diminished detectability and robustness relative to our approach. Furthermore, we find that their generation latency is several orders of magnitude greater than GaussMark (Figure 27).

**Miscellaneous.** We conduct many additional ablations to better understand the effect of different choices of watermarking hyperparameters on GaussMark both w.r.t. detectability (Appendix D) and model quality (Appendix E).

## 5. Discussion

In this work, we introduced a new watermarking scheme, GaussMark, with theoretical guarantees on statistical validity and power, and empirically demonstrated its efficacy.

**Advantages of GaussMark.** The primary advantage of GaussMark is that it is the first simple, efficient, and statistically valid watermarking scheme that can be immediately integrated into existing inference pipelines with essentially no modification and absolutely no impact on generation latency. In addition, detection speed is extremely fast relative to alternative schemes and has memory requirements essentially on par with those of inference. Furthermore, GaussMark is somewhat robust to corruption and, in practice, does not need to know the prompt or any other information about the text to detect the watermark (cf. Figure 14). In contradistinction to many alternative watermarking schemes, which focus on the next-token sampling process and whose robustness is often limited to edit-distance corruptions, GaussMark changes the distribution of the text itself and thus can be viewed as a form of 'structural' watermarking, which has the potential to be more robust to a wider variety of corruptions such as roundtrip translations. Finally, unlike prior works, GaussMark is not restricted to the next-token prediction paradigm in autoregressive language models; by design, it can also be readily implemented in emerging hybrid paradigms such as Language Diffusion Models (Nie et al., 2025), a direction we leave for future exploration.

**Limitations of GaussMark.** While there are a number of benefits of GaussMark relative to prior approaches, there are four key limitations. (1) Unlike many alternatives, GaussMark has no formal guarantees of distortion freeness; while we have empirically demonstrated that GaussMark does not hurt model quality on a range of understanding, reasoning, math, and conversational benchmarks, further investigation on the effects on text quality is required before GaussMark can be safely deployed. (2) In general, GaussMark requires more tokens to reliably detect a watermark than alternative approaches (Kuditipudi et al., 2023; Kirchenbauer et al., 2023b; Dathathri et al., 2024); that said, the speed of GaussMark allows our approach to remain competitive, as we can detect even 1K tokens much more quickly than some alternative approaches require to detect 200 tokens (Kirchenbauer et al., 2023b). (3) Unlike some al-

ternative schemes, GaussMark assumes *white-box* access to model weights in order for detection to be possible; we view this as natural, motivated by a setting where the LM provider offers watermarking detection as an additional service on top of the LM itself, but this is a stronger access model than is considered in some other works (Golowich and Moitra, 2024; Christ et al., 2024b; Aaronson and Kirchner, 2022; Kirchenbauer et al., 2023b; Kuditipudi et al., 2023). (4) GaussMark is less robust to token-level corruptions than Kuditipudi et al. (2023), although is potentially more robust than instantiations of the scheme of Kirchenbauer et al. (2023a) that preserve text quality. Finally, in addition to the above limitations, we did not explore the extent to which aggressive quantization can lead to a reduction in the ability to detect GaussMark, which may pose a challenge in practical deployment. We leave the interesting question of how to make our approach quantization-aware to future work.

**Future Directions.** GaussMark is a first step toward generating *structural watermarks* in LMs, but potential improvements abound. In particular, while we propose adding noise to low-rank components of the model weights as some way to mitigate the trade-off between model quality and watermark detectability, we did not explore alternative methods, such as training the model slightly after adding noise, adding noise only to a sub-network identified by mechanistic interpretability (Templeton et al., 2024; Bereska and Gavves, 2024), or noising multiple weights at once. As well as potential modifications of the intervention itself, additional probing on the effect GaussMark has on text quality is likely necessary before widespread deployment. In particular, we did not investigate the effect that our approach has on safety or alignment benchmarks, an important component of production-ready LMs.

## Acknowledgements

AS thanks Martin Pawelczyk for useful discussions. AB thanks Jordan Ash and Cyril Zhang for helpful discussions. AS and SR acknowledge support from the Simons Foundation and NSF through award DMS-2031883, as well as from ARO through award W911NF-21-1-0328.

## Impact Statement

This paper presents work whose goal is to advance the field of Machine Learning. There are many potential societal consequences of our work, none of which we feel must be specifically highlighted here.

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

# Contents

## A. Additional Notation

For any $T \geq 0$, $[T]$ denotes the set $\{1, \ldots, T\}$. For a vector $v$, $\|v\|_2$ denotes its $\ell_2$ norms. We denote by $\mathbf{I}_d$ the identity matrix in $\mathbb{R}^{d \times d}$. For any set $\mathcal{Y}$, $\Delta(\mathcal{Y})$ denotes the set of all distributions over $\mathcal{Y}$. $\mathcal{N}(0, \sigma^2 \mathbf{I}_d)$ represents an isotropic Gaussian distribution in $d$ with variance $\sigma^2$. The symbol $\circ$ is used to denote the concatenation of strings of tokens, and $\otimes$ denotes the product of two distributions (i.e. samples are independent).

## B. Related Works

In this section, we discuss the most relevant works on LLM watermarking below, and refer the reader to Liang et al. (2024); Liu et al. (2024b) for a thorough discussion on other watermarking schemes for text, image and audio generation.

**Watermarking Approaches with Statistical Guarantees.** Watermarking schemes with rigorous statistical guarantees can be broadly categorized as either *distorionary* or *non-distortionary*. In the former, the scheme modifies the distribution of the LM's next-token predictions; for example, Kirchenbauer et al. (2023a); Zhao et al. (2023a;b); Aaronson and Kirchner (2022) introduced schemes that partition the vocabulary into a 'red set' and a 'green set,' and adjust the distribution to favor green-set tokens in a manner detectable by examining empirical frequencies of tokens in generated text. While such schemes offer theoretical guarantees, they often lead to significant distortion in the generated text due to token biasing; moreover, they can be quite vulnerable to paraphrasing attacks (Rastogi and Pruthi, 2024; Hou et al., 2023; Chang et al., 2024; Jovanović et al., 2024). We empirically compare GaussMark to the scheme introduced by Kirchenbauer et al. (2023a) in Appendix H.

The second category, non-distortionary schemes, involves concealing the watermarking signal by influencing the pseudo-random number generator (PRNG) involved in sampling individual tokens. Kuditipudi et al. (2023) provide an example of an extremely robust such scheme; unfortunately, it suffers from poor scalability due to the requirement that the length of the watermarking key must be proportional to the number of queries submitted to the model in order to maintain diversity of responses; with the popularity of contemporary LMs, this would lead to impractically large keys and slow detection times. In order to avoid the limitations of Kuditipudi et al. (2023), Christ et al. (2024b) introduce a stronger, cryptographically-inspired notion of *undetectability* and they, along with (Golowich and Moitra, 2024; Zamir, 2024; Fairoze et al., 2023), provide undetectable watermarking schemes that pseudorandomly select the next token using a hash function of the preceding $k$ tokens, based on a secret key with provable guarantees both on statistical validity and robustness to token-level corruptions. While important theoretical contributions, these schemes are based on cryptographical primitives that are not yet ready for large-scale deployment in LLM generation due to the resulting increase in generation latency. For example, the scheme of Fairoze et al. (2023) requires hundreds of seconds for generation and detection in contrast to our scheme, which achieves orders-of-magnitude faster generation and detection speeds for longer token sequences (Figure 3). In contradistinction to the above schemes, GaussMark is *white-box*, i.e. it requires access to the weights of the LM and is distortionary; however, we empirically demonstrate that the perturbations we introduce do not result in a statistically significant decline in model quality.

**Other Practical Watermarking Methods.** While the above schemes provide strong theoretical guarantees, with the exception of the schemes of Kirchenbauer et al. (2023a;b); Kuditipudi et al. (2023), they have generally not been seriously evaluated empirically. Due to the aforementioned drawbacks of both of these schemes, Dathathri et al. (2024) proposed SynthID-Text, which relies on an efficient implementation of ideas similar to the approach of Christ et al. (2024b); Zamir (2024); Golowich and Moitra (2024), with an additional layer of sophistication with a more involved sampling scheme. While the bespoke implementation on several select models (Gemma 2B, Gemma 7B (Team et al., 2024), and Mistral 7B-IT (Jiang et al., 2023)) is efficient and has relatively low latency, the approach considered empirically unfortunately requires storage costs scaling linearly with the number of tokens generated in order to maintain high quality in generated text. Indeed, similar to the approach of Kirchenbauer et al. (2023a), SynthID-Text hashes previous tokens in order to affect the distribution of generation going forward; thus repeated strings of tokens lead to significantly distorted text. While this issue is not as much of a problem as in the approach of Kirchenbauer et al. (2023a) due to the fact that sampling remains stochastic in Dathathri et al. (2024), the empirical degree of distortion necessitates storing previous strings of tokens in order to not implement the watermark on such repeated strings; for production-level systems that require maintaining quality and diversity while generating millions of tokens per day, this approach is likely not entirely feasible. Most similar to our

approach is that of the concurrent work Christ et al. (2024a), discussed in the introduction. Fu et al. (2024) provides a more practical implementation of Kirchenbauer et al. (2023a) that improves the generation quality by incorporating the semantic structure of the text to partition the token set into corresponding red-lists / green-lists.

**Post-hoc Detection of AI-generated text.**   An alternative to developing rigorous watermarking schemes is to design algorithms that can post-hoc distinguish whether a given piece of text is more likely to be AI or Human-generated, e.g. by relying on statistical properties of the text under a given language model (Gehrmann et al., 2019; Mitchell et al., 2023), or by using specifically trained classifiers to discriminate human-text from AI-text (OpenAI, 2023; Zellers et al., 2019). Given the rapid rate of progress in natural language processing, however, the development of better and larger language models will only make post-hoc detection harder (Schütz et al., 2021; Gambini et al., 2022; Sadasivan et al., 2023); furthermore, for a number of applications, such as detecting cheating in scholastic environments, a well-calibrated procedure is essential, which relies on formal statistical guarantees.

**Watermarking the model by perturbing weights.**   Finally, we note that earlier works have considered perturbing model weights to prevent IP theft (Li et al., 2023a; Fernandez et al., 2024; Boroujeny et al., 2024). For example, Aaronson and Kirchner (2022); Li et al. (2023a); Boroujeny et al. (2024) use the gap between the quantized model and the full-precision model to plant the watermarks and Fernandez et al. (2024) use model symmetries to generate an invariant model copy. While our approach also alters the weights of the network, our objective is different from theirs as we want to watermark the generated text instead of the model itself.

**Attacks on Watermarked Models.**   A series of works have demonstrated that adversaries with substantial resources can compromise watermarking schemes. Gu et al. (2023) demonstrate that watermarking schemes can be learned using a student-teacher framework. Kirchenbauer et al. (2023b); Christ et al. (2024b); Pang et al. (2024); Zhang et al. (2023) developed further attacks on watermarking schemes. These attacks are generally resource-intensive, often requiring repeated access to an oracle capable of generating high-quality perturbations of input text or evaluating the quality of a given text. Unfortunately, with sufficient resources, an adversary can bypass these schemes entirely by training their own "unwatermarked" AI model to produce content.

**Relation to Pawelczyk et al. (2024)**   The structure of our test statistic in Equation (1) resembles the Gaussian Unlearning Score (GUS) from Pawelczyk et al. (2024) for evaluating the efficacy of machine unlearning algorithms. However, we emphasize a key methodological distinction: while the GUS perturbs data and computes gradients with respect to inputs, our method perturbs model parameters and computes gradients with respect to those parameters, reflecting the distinct goals of unlearning and watermarking. Pawelczyk et al. (2024) considers unlearning by aggregating an inner product-based test statistic over multiple data points on models altered by a stochastic unlearning process. In contrast, we focus on a fixed data point and evaluate the likelihood that it was generated by a watermarked model. Furthermore, Section 3.1 shows that our test can be viewed as a computationally efficient approximation to the minimax optimal test, providing theoretical grounding for our approach. The resemblance between GUS and GaussMark suggests that the former likely admits a theoretical justification, although this was beyond the scope of that work.

# C. Implementation Details

In this section, we provide the details of the implementation of our experiments. As discussed earlier, we use the `realnewslike` split of the C4 dataset (Raffel et al., 2020); for our experiments, we took the first 1000 samples from this corpus and created prompts by truncating each text at 200 tokens. We considered three models, Mistral−7B, Llama3.1−8B, and Phi3.5−Mini. Each model is a transformer and we apply our watermark to one of the MLP layers in one of the transformer blocks. For Mistral−7B and Llama3.1−8B, which share an architecture, there are three MLP weights for each layer: down_proj, up_proj, and gate_proj. For Phi3.5−Mini, there are two choices of weights per layer: gate_up_proj and down_proj. In all cases, there is a final linear layer to which the softmax function is applied to get the next token probabilities; we found that directly perturbing this layer led to either negligable detectability of the watermark or an unacceptable decrement of model quality and thus only consider perturbing the MLP weights that are part of the transformer blocks.

In all of our experiments, we used 3 seeds for watermarking the model and generating tokens. The watermarking parameters used to construct the plots in the main body of the papers are summarized in Table 2. These parameters were selected by

| Model | Watermarked Layer | Watermarked Weight | Variance ($\sigma^2$) | Dimension ($d$) |
|-------|-------------------|--------------------|-----------------------|-----------------|
| Mistral–7B | 20 | up_proj | 1e-05 | 58M |
| Llama3.1–8B | 28 | up_proj | 3e-04 | 58M |
| Phi3.5–Mini | 20 | down_proj | 1e-03 | 25M |

*Table 2.* Selected watermarking hyperparameters.

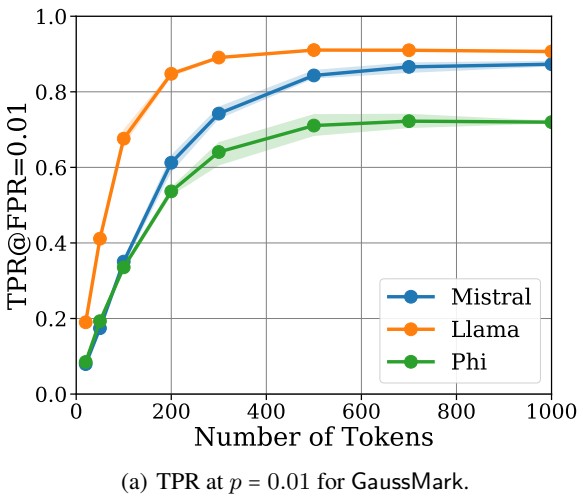

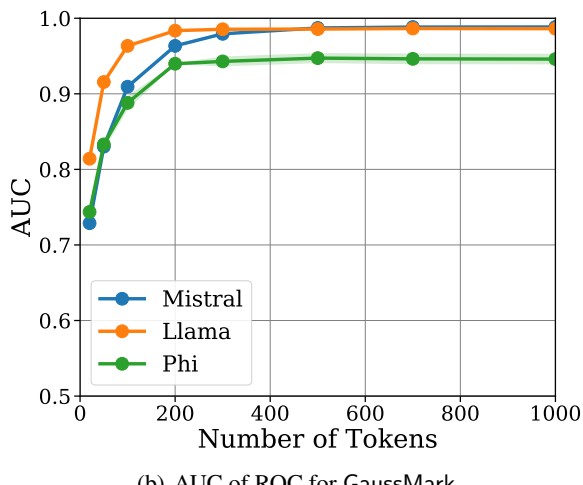

(a) TPR at $p = 0.01$ for GaussMark.

(b) AUC of ROC for GaussMark.

*Figure 4.* Effect of number of tokens on the detectability of GaussMark as measured by (a) the True Positive Rate at $p = 0.01$ and (b) the AUC of the ROC. As the number of tokens increases, the detectability of the watermark improves, as predicted by the theory.

sweeping over many choices of layer, weight, and variance and for each watermarking parameter evaluating the following: (1) finding the p-values of watermark detection on completions of the 1K prompts and (2) evaluating the quality of the watermarked model on SuperGLUE and GSM–8K. For the subset of those models that had acceptably small p-values and had essentially no statistically significantly worse performance on GSM–8K and SuperGLUE, we then (3) ran the watermarked model through AlpacaEval–2.0 with the unwatermarked model's responses as the comparator. We then chose the watermark parameters for each model as those that maximized detectability subject to not hurting performance on steps (2) and (3).

## D. Empirical Results on Detectability of GaussMark

In this section, we present further results on the detectability of the watermarked models. In Figure 4, we display two different measures of watermark detectability: the true positive rate (TPR) at a false positive rate (FPR) of $p = 0.01$ and the area under the curve (AUC) of the Receiver Operating Characteristic (ROC) (Carlini et al., 2022; Casella and Berger, 2001). We also plot in Figure 5 the ROC curves themselves for each model on different generation lengths of watermarked text. Along with Figure 1 and Figure 2(a), these plots conclusively demonstrate the benefits of longer token sequences for detectability of GaussMark. The trends are consistent across all metrics and models and align well with the theoretical predictions discussed in Section 3.2. Notably, the detectability of the watermark asymptotes as the number of tokens increases and is predicted by the dimension of $\Theta$.

To further evaluate the validity of the simplifying assumptions discussed following Proposition 3.3, we analyze the distribution of gradient norms, $\|\nabla \log p_\theta(y \mid x)\|$, for 150 sampled responses across five different prompts for each of the considered models. The distributions are visualized in Figure 6. We see that the gradient norms are relatively tightly concentrated, for Mistral–7B and Llama3.1–8B in particular, and empirically justify the hypothesis that the gradients lie within a relatively well-conditioned annulus, which is a sufficient condition for the alignment of gradient norms. Indeed the multiplicative range of the gradients (i.e. the maximum norm divided by the minimum norm of the responses for each prompt) is less than 25 for both Mistral–7B and Llama3.1–8B, and less than 40 for Phi3.5–Mini.

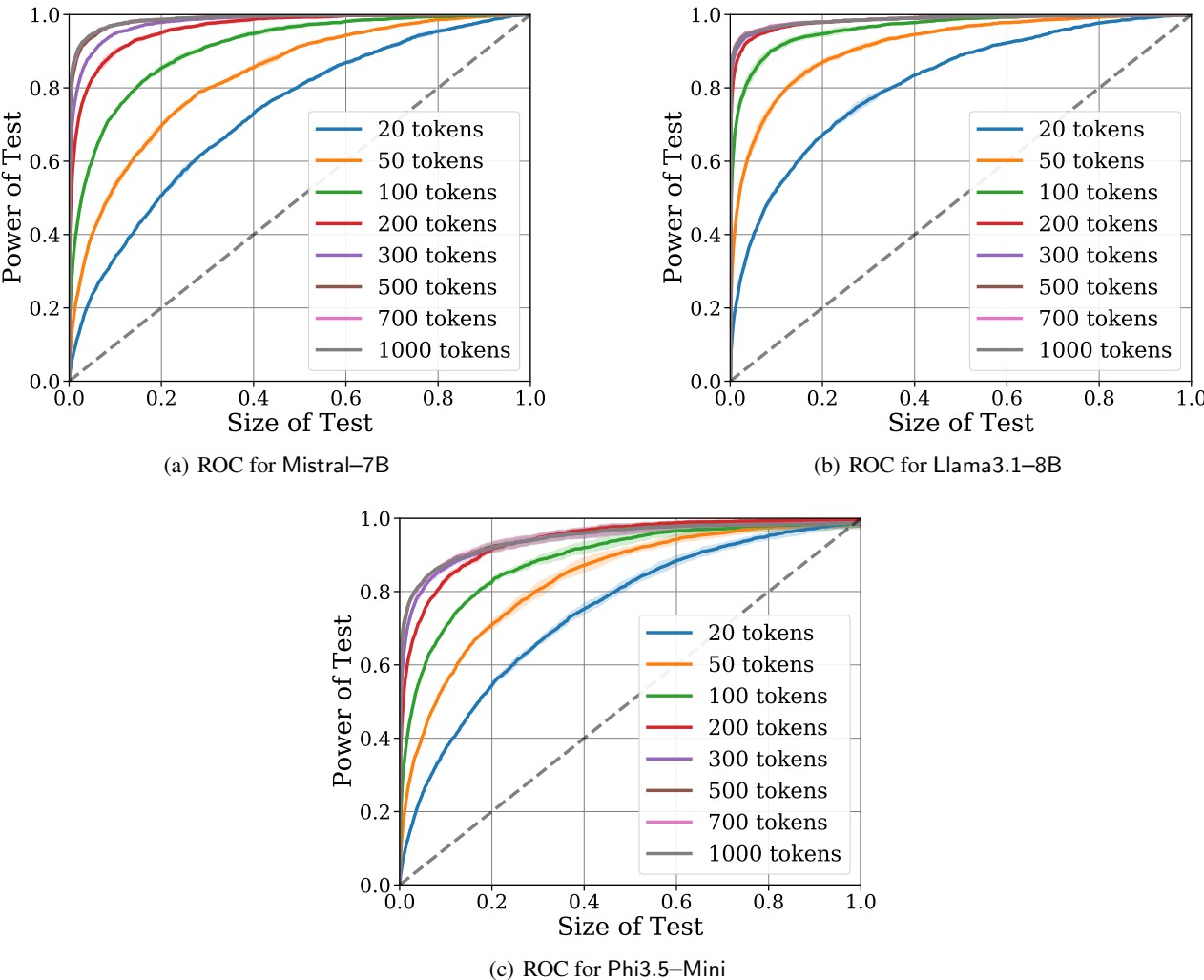

(a) ROC for Mistral–7B

(b) ROC for Llama3.1–8B

(c) ROC for Phi3.5–Mini

*Figure 5.* Effect of number of tokens on the detectability of GaussMark as measured by the ROC curves for (a) Mistral–7B, (b) Llama3.1–8B, and (c) Phi3.5–Mini. As the number of tokens increases, the ROC curve shifts to the left, indicating better detectability. The trivially achievable ROC curve of a random test is shown in gray for comparison.

Finally, we report the median p-values of watermarked text generated from the 1K prompts for many different choices of layer, variance, and weight matrix for each of the three considered models in Figure 7, Figure 8, and Figure 9. As expected, when the variance is too small, the p-values are large, indicating that the watermark signal is insufficiently strong. Perhaps surprisingly, when the variance is too large, there is a similar increase in p-values. We conjecture that this latter effect is caused by the fact that the linear softmax approximation is *local* and the quality of this approximation degrades as the norm of $\xi$ increases; thus the extent to which the conclusion of Proposition 3.3 applies attenuates when the variance is too large.

## E. Empirical Results on the Quality of Watermarked Models

We here provide additional results on the effect that GaussMark has on the downstream model performance. We begin by reporting the SuperGLUE scores of the both the watermarked and unwatermarked models broken down by the 8 individual tasks that make up the benchmark. We report the average score and the standard error as computed by `lm-evaluation-harness` (Gao et al., 2024) for each model in Table 4, Table 5, and Table 6. We see that the watermark has a negligible effect on the performance of the models on the SuperGLUE benchmark, which is precisely why these particular watermarking parameters were chosen.

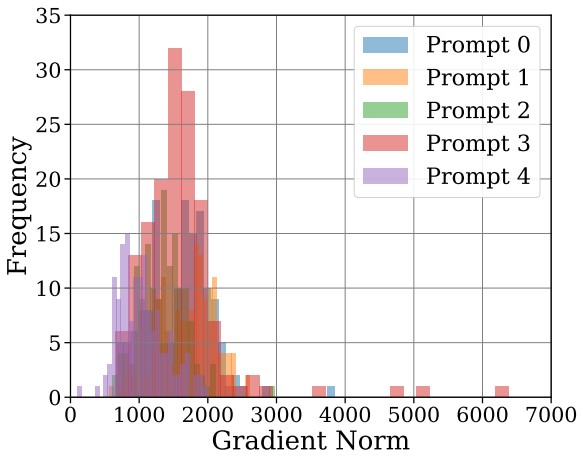

(a) Gradient Norm Distribution for Mistral–7B.

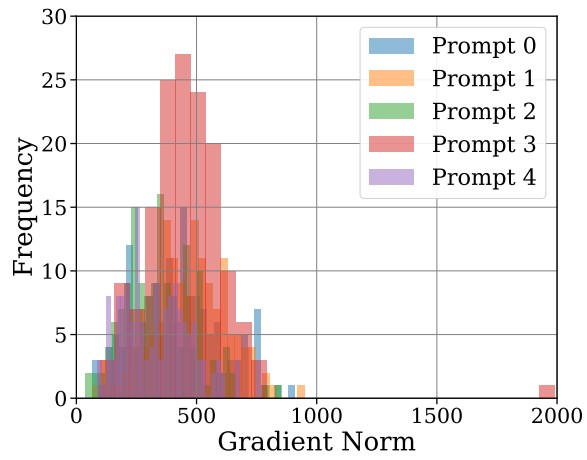

(b) Gradient Norm Distribution for Llama3.1–8B.

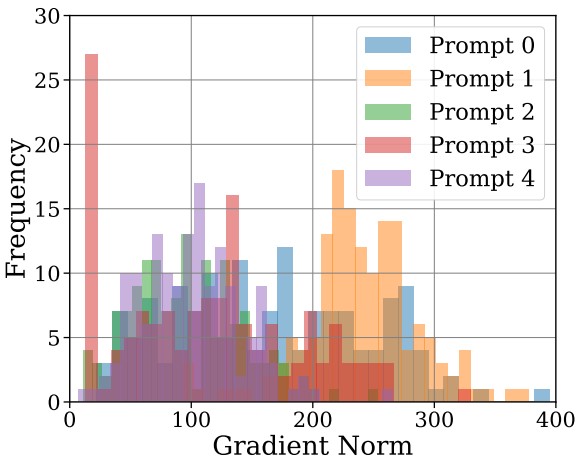

(c) Gradient Norm Distribution for Phi3.5–Mini.

*Figure 6.* Distribution of $\|\nabla \log p_\theta(y \mid x)\|$ for 5 different prompts for (a) Mistral–7B, (b) Llama3.1–8B, and (c) Phi3.5–Mini. The relatively tight concentration of the gradient norms serves to justify the linear softmax approximations as per the discussion following Proposition 3.3.

In addition to SuperGLUE, GSM–8K, and AlpacaEval–2.0, we also measure the decline in model quality through crossentropy on actual text. In particular, for each of the 1K samples in our corpus, we compared the per-token crossentropy of the unwatermarked models with those of the watermarked models (as well as the Rank-reduced version, cf. Appendix G). We see in Table 3 that there is essentially no difference in next-token prediction abilities between watermarked and unwatermarked models, providing further evidence that GaussMark does not negatively affect the performance of the model.

Finally, analogous to the ablation in Appendix D where we demonstrate the effect that different choices of watermarking parameters have on the detectability of the watermark, in Figures 10 to 12 we show the effect that these identical variations have on the performance of the models on GSM–8K benchmark. The plots are colored so as to clearly display which variations are statistically indistinguishable from the base model (yellow) and which are not (purple). For the sake of brevity, we restrict our attention to the math benchmark as opposed to the individual SuperGLUE tasks, although the general trend persists across tasks. Unsurprisingly, as the variance parameter $\sigma$ increases and the watermarked model becomes more different from the unwatermarked model, the performance decreases. Interestingly, the middle layers appear to be the most noise-robust in Mistral–7B and Llama3.1–8B, with layers near the top and bottom being able to handle substantially less perturbation. In addition, Phi3.5–Mini appears to be substantially more noise-robust and can tolerate significantly higher variance than either of the other two models.

| Var | 1e-05 | 3e-05 | 1e-04 | 3e-04 | 1e-03 |
|---|---|---|---|---|---|
| Layer 31 | 0.0000 | 0.0000 | 0.0000 | 0.0000 | 0.0000 |
| Layer 30 | 0.0008 | 0.0000 | 0.0000 | 0.0000 | 0.0000 |
| Layer 29 | 0.0001 | 0.0000 | 0.0000 | 0.0000 | 0.0000 |
| Layer 28 | 0.0002 | 0.0000 | 0.0000 | 0.0000 | 0.0000 |
| Layer 25 | 0.0000 | 0.0000 | 0.0000 | 0.0000 | 0.0000 |
| Layer 20 | 0.0000 | 0.0000 | 0.0000 | 0.0000 | 0.0000 |
| Layer 10 | 0.0011 | 0.0006 | 0.0016 | 0.0048 | 0.0193 |

(a) Mistral–7B median $p$-values (down_proj)

| Var | 1e-05 | 3e-05 | 1e-04 | 3e-04 | 1e-03 |
|---|---|---|---|---|---|
| Layer 31 | 0.0000 | 0.0000 | 0.0000 | 0.0000 | 0.0000 |
| Layer 30 | 0.0000 | 0.0000 | 0.0000 | 0.0000 | 0.0000 |
| Layer 29 | 0.0000 | 0.0000 | 0.0000 | 0.0000 | 0.0000 |
| Layer 28 | 0.0001 | 0.0000 | 0.0000 | 0.0000 | 0.0000 |
| Layer 25 | 0.0000 | 0.0000 | 0.0000 | 0.0000 | 0.0000 |
| Layer 20 | 0.0000 | 0.0000 | 0.0000 | 0.0000 | 0.0000 |
| Layer 10 | 0.0017 | 0.0003 | 0.0016 | 0.0009 | 0.0237 |

(b) Mistral–7B median $p$-values (up_proj)

| Var | 1e-05 | 3e-05 | 1e-04 | 3e-04 | 1e-03 |
|---|---|---|---|---|---|
| Layer 31 | 0.0000 | 0.0000 | 0.0000 | 0.0000 | 0.0000 |
| Layer 30 | 0.0002 | 0.0000 | 0.0000 | 0.0000 | 0.0067 |
| Layer 29 | 0.0007 | 0.0000 | 0.0000 | 0.0000 | 0.0146 |
| Layer 28 | 0.0008 | 0.0000 | 0.0000 | 0.0000 | 0.0010 |
| Layer 25 | 0.0005 | 0.0000 | 0.0000 | 0.0000 | 0.0000 |
| Layer 20 | 0.0005 | 0.0000 | 0.0000 | 0.0000 | 0.0000 |
| Layer 10 | 0.0078 | 0.0010 | 0.0105 | 0.0287 | 0.1523 |

(c) Mistral–7B median $p$-values (gate_proj)

*Figure 7.* Detectability of watermarked Mistral–7B models with GaussMark for different layers and variances as measured by the median p-value over 1K prompts for (a) down_proj, (b) up_proj, and (c) gate_proj, colored by value.

## F. Empirical Results on the Robustness of GaussMark

In addition to statistical validity, detectability, and lack of model degradation, an important property of a watermarking scheme is its robustness to corruption. In many applications, the watermarked text may be modified in some ways, either intentionally (e.g. to evade plagiarism accusations) or unintentionally (e.g. when only a portion of a model's output is included in the final draft of a paper). Consequently, for a watermark to be practical, it must demonstrate at least some degree of robustness to such changes.

We consider four kinds of corruptions commonly discussed in the watermarking literature (Kuditipudi et al., 2023; Christ et al., 2024b; Liu et al., 2024a; Zhao et al., 2023a; Golowich and Moitra, 2024): three types of token-level corruptions (random insertions, deletions, and substitutions) and a more challenging paraphrasing attack—namely, 'roundtrip translation,' where watermarked text is translated into another language and then back to English. Details of the token-level corruptions are provided below:

- **Random insertions.** We choose random tokens from the vocabulary and insert them into the text. We either insert these tokens at random locations or as a prefix, with the latter simulating some degree of ignorance of the prompt itself.

- **Random deletions.** We choose random tokens from the text and delete them. We also consider deleting the first few tokens of the text to understand whether or not knowledge of the prompt is necessary for GaussMark to detect the watermark.

| Var | 1e-05 | 3e-05 | 1e-04 | 3e-04 | 1e-03 |
|---|---|---|---|---|---|
| Layer 31 | 0.0000 | 0.0000 | 0.0000 | 0.0000 | 0.0000 |
| Layer 30 | 0.0379 | 0.0011 | 0.0000 | 0.0000 | 0.0000 |
| Layer 29 | 0.0488 | 0.0054 | 0.0000 | 0.0000 | 0.0000 |
| Layer 28 | 0.1403 | 0.0292 | 0.0003 | 0.0000 | 0.0000 |
| Layer 25 | 0.0830 | 0.0154 | 0.0002 | 0.0000 | 0.0000 |
| Layer 20 | 0.0867 | 0.0099 | 0.0000 | 0.0000 | 0.0000 |
| Layer 10 | 0.0873 | 0.0156 | 0.0003 | 0.0000 | 0.0004 |

(a) Llama3.1–8B median $p$-values (down_proj).

| Var | 1e-05 | 3e-05 | 1e-04 | 3e-04 | 1e-03 |
|---|---|---|---|---|---|
| Layer 31 | 0.0019 | 0.0000 | 0.0000 | 0.0000 | 0.0000 |
| Layer 30 | 0.0966 | 0.0072 | 0.0000 | 0.0000 | 0.0000 |
| Layer 29 | 0.2075 | 0.0482 | 0.0008 | 0.0000 | 0.0000 |
| Layer 28 | 0.1238 | 0.0263 | 0.0003 | 0.0000 | 0.0000 |
| Layer 25 | 0.1275 | 0.0253 | 0.0006 | 0.0000 | 0.0000 |
| Layer 20 | 0.0993 | 0.0126 | 0.0001 | 0.0000 | 0.0000 |
| Layer 10 | 0.1160 | 0.0195 | 0.0007 | 0.0001 | 0.0017 |

(b) Llama3.1–8B median $p$-values (up_proj).

| Var | 1e-05 | 3e-05 | 1e-04 | 3e-04 | 1e-03 |
|---|---|---|---|---|---|
| Layer 31 | 0.1166 | 0.0216 | 0.0001 | 0.0000 | 0.0000 |
| Layer 30 | 0.1713 | 0.0624 | 0.0037 | 0.0000 | 0.0000 |
| Layer 29 | 0.1989 | 0.0959 | 0.0059 | 0.0000 | 0.0000 |
| Layer 28 | 0.1976 | 0.0802 | 0.0051 | 0.0001 | 0.0000 |
| Layer 25 | 0.1817 | 0.0710 | 0.0066 | 0.0001 | 0.0000 |
| Layer 20 | 0.2244 | 0.0729 | 0.0054 | 0.0001 | 0.0000 |
| Layer 10 | 0.1900 | 0.0798 | 0.0093 | 0.0007 | 0.0017 |

(c) Llama3.1–8B median $p$-values (gate_proj).

*Figure 8.* Detectability of watermarked Llama3.1–8B models with GaussMark for different layers and variances as measured by the median p-value over 1K prompts for (a) down_proj, (b) up_proj, and (c) gate_proj, colored by value.

- **Random substitutions.** We choose random tokens from the vocabulary and substitute them for tokens in the text. We consider both random substitutions and substitutions of the first few tokens of the text.

For these token-level attacks, Figure 13 (a)-(c) illustrates how the percentage of watermarked text detected at the $p = 0.05$ level by GaussMark.Detect decreases as progressively larger fractions of the text are corrupted. The results indicate that the degree of robustness varies significantly depending on both the model and the type of corruption, with Llama3.1–8B being sufficiently robust so as to retain significant detection power with even half of the tokens corrupted. However, we emphasize that these token-level attacks are relatively crude and do not reflect realistic threat models, as the quality of the corrupted text degrades considerably. Thus, following Kuditipudi et al. (2023), we consider a more realistic paraphrasing attack, where we use Helsinki-NLP/opus-mt-tc-big-en-fr and Helsinki-NLP/opus-mt-tc-big-fr-en (Tiedemann et al., 2022) to translate watermarked text from English to French and back. We then evaluate the impact of this roundtrip translation on the detectability of GaussMark. Figure 13-(d) presents the ROC curves for each model following roundtrip translation through French. Despite the increased difficulty of this attack, all models demonstrate nontrivial detection power, demonstrating some robustness in this more realistic corruption scenario.

Finally, we note that in its most basic form, GaussMark.Detect requires knowledge of the prompt. Indeed, the detection procedure is predicated on the assumption that we can take the gradient of the log probabilities of the *generated* tokens conditional on the input. Because such knowledge is rarely available in practice, in Figure 14 we investigate the effect that inserting, deleting, and substituting tokens at the beginning of the concatenation of the input prompt and the generated text has on detectability. We see that GaussMark is extremely robust to ignorance of the prompt, with the median p-values of the

| Var | 1e-05 | 3e-05 | 1e-04 | 3e-04 | 1e-03 |
|---|---|---|---|---|---|
| Layer 31 | 0.0130 | 0.0001 | 0.0000 | 0.0000 | 0.0000 |
| Layer 30 | 0.3106 | 0.2389 | 0.0844 | 0.0111 | 0.0001 |
| Layer 29 | 0.1966 | 0.0265 | 0.0122 | 0.0077 | 0.0047 |
| Layer 28 | 0.4543 | 0.3561 | 0.2060 | 0.0567 | 0.0008 |
| Layer 25 | 0.2985 | 0.2208 | 0.0616 | 0.0072 | 0.0055 |
| Layer 20 | 0.3211 | 0.2141 | 0.1000 | 0.0158 | 0.0002 |
| Layer 10 | 0.3109 | 0.2481 | 0.1259 | 0.0337 | 0.0015 |

(a) Phi3.5–Mini median $p$-values (down_proj).

| Var | 1e-05 | 3e-05 | 1e-04 | 3e-04 | 1e-03 |
|---|---|---|---|---|---|
| Layer 31 | 0.2620 | 0.1290 | 0.0188 | 0.0001 | 0.0000 |
| Layer 30 | 0.2737 | 0.1526 | 0.0412 | 0.0015 | 0.0000 |
| Layer 29 | 0.2948 | 0.1757 | 0.0607 | 0.0036 | 0.0009 |
| Layer 28 | 0.3174 | 0.2133 | 0.0742 | 0.0099 | 0.0000 |
| Layer 25 | 0.3503 | 0.2336 | 0.1041 | 0.0145 | 0.0001 |
| Layer 20 | 0.3408 | 0.2286 | 0.0947 | 0.0101 | 0.0000 |
| Layer 10 | 0.3442 | 0.2300 | 0.0777 | 0.0101 | 0.0004 |

(b) Phi3.5–Mini median $p$-values (gate_up_proj).

*Figure 9.* Detectability of watermarked Phi3.5–Mini models with GaussMark for different layers and variances as measured by the median p-value over 1K prompts for (a) down_proj and (b) gate_up_proj, colored by value.

| Model | Unwatermarked | **Vanilla** GaussMark | **Rank-Reduced** GaussMark |
|---|---|---|---|
| Mistral | 1.9199 ± 0.0135 | 1.9385 ± 0.0134 | 1.9410 ± 0.0135 |
| Llama | 2.1709 ± 0.0149 | 2.2331 ± 0.0145 | 2.1901 ± 0.0148 |
| Phi | 2.1396 ± 0.0133 | 2.1617 ± 0.0133 | 2.1620 ± 0.0133 |

*Table 3.* Comparison of crossentropies of GaussMark to unwatermarked models on C4.

watermarked text remaining relatively stable across different prompt corruptions. Indeed, because removing tokens from the concatenation of the prompt and LM's response results in significantly shorter sequences, we see by comparing Figure 14(b) to Figure 2(a) that much of the attenuation in detection power is due to the shorter length of the sequences rather than the corruption itself.

We conclude this section by summarizing our experiments on the robustness of GaussMark and the key takeaways. In Figure 15 and 16, along with Figure 13 and 14 we display the results of these token-level corruptions. In particular, we replace increasing fractions of the watermarked text with the corrupted text and measure the effect on the TPR at FPR of 0.05 as well as the median p-values and the AUCs of the resulting tests. We find that all of the measures generally agree in terms of the effect that such corruption has on the detectability of GaussMark. First, when substantial fractions of the text are substituted or removed, we find that the detectability of GaussMark is negatively affected. On the other hand, corruptions to the prompt itself, including removing half of the concatenated prompt and generated response, have very little effect, demonstrating the robustness of GaussMark to ignorance of the prompt. We emphasize that these token-level corruptions should only be seen as a preliminary analysis. On one hand, they are too simplistic to encompass the range of techniques a motivated attacker might employ. On the other hand, they are overly aggressive, as the resulting corrupted text is of extremely low quality.

## G. Can GaussMark be Improved with Rank-reduction?

As discussed previously, the key challenge in applying GaussMark is identifying the appropriate parameters $\theta$ to which to add noise, taking into account the careful balance between detectability and model quality. Recent empirical work has observed that the highest principal components of LMs' weight matrices are maximally significant in determining model predictions (Hu et al., 2021; Sharma et al., 2024; Dettmers et al., 2024; Guo et al., 2023; Han et al., 2024; Wang et al., 2024), with Sharma et al. (2024) in particular suggesting that the lower principal components of the weight matrices can be treated as 'noise' and their removal can even improve model performance on some tasks. Motivated by these observations, we propose a rank-reduced version of GaussMark wherein we let $\Theta$ be the subspace spanned by the lower principal components

| Task | Llama | Llama (Unwatermarked) |
|------|-------|----------------------|
| **BoolQ** | 0.8223 ± 0.0067 | 0.8217 ± 0.0067 |
| **CB** | 0.6607 ± 0.0638 | 0.6250 ± 0.0653 |
| **COPA** | 0.9000 ± 0.0302 | 0.8700 ± 0.0338 |
| **MultiRC** | 0.5718 ± 0.0071 | 0.5720 ± 0.0071 |
| **ReCoRD** | 0.9168 ± 0.0027 | 0.9222 ± 0.0026 |
| **RTE** | 0.6931 ± 0.0278 | 0.7040 ± 0.0275 |
| **WiC** | 0.5047 ± 0.0198 | 0.5157 ± 0.0198 |
| **Winograd** | 0.6058 ± 0.0482 | 0.5865 ± 0.0485 |

*Table 4.* Performance of GaussMark on SuperGLUE(Llama3.1−8B).

| Task | Mistral | Mistral (Unwatermarked) |
|------|---------|------------------------|
| **BoolQ** | 0.8162 ± 0.0068 | 0.8205 ± 0.0067 |
| **CB** | 0.6071 ± 0.0659 | 0.5357 ± 0.0672 |
| **COPA** | 0.9100 ± 0.0288 | 0.9200 ± 0.0273 |
| **MultiRC** | 0.5590 ± 0.0071 | 0.5672 ± 0.0071 |
| **ReCoRD** | 0.9203 ± 0.0027 | 0.9219 ± 0.0026 |
| **RTE** | 0.7112 ± 0.0273 | 0.6823 ± 0.0280 |
| **WiC** | 0.5831 ± 0.0195 | 0.5674 ± 0.0196 |
| **Winograd** | 0.5192 ± 0.0492 | 0.4615 ± 0.0491 |

*Table 5.* Performance of GaussMark on SuperGLUE (Mistral−7B).

of a given parameter matrix.[1].

The rank-reduced instantiation of GaussMark is motivated by the empirical observation that the top Principal Components (PCs) of the MLP weights in transformers are often sufficient to obtain strong performance in modern LMs. Thus, for a given weight matrix with rank $m$, we compute the projection onto the bottom $m − k$ PCs, where $k$ is a hyperparameter we select. Thus, we only add the perturbations $\xi$ to the bottom $m − k$ principle components. Increasing $k$ better preserves the model's performance, as the effect of the watermark perturbation is restricted to a less influential and lower dimensional space; at the same time, choosing $k$ too large leads to a significantly less detectable watermark. We then compute the gradient by projecting the gradient of the weight matrices onto this lower dimensional space as well for use in GaussMark.Detect. The specific watermarking hyperparameters we employ are detailed in Table 7.

We find that this rank-reduced version of GaussMark allows for significantly more noise to be added without compromising model quality (as measured by the metrics discussed in Section 4.2). In Figure 17(a), we plot the fraction of detected watermarked responses at $p = 0.05$ for the rank-reduced version of GaussMark averaged over 3 seeds for each model for a variety of response lengths, and observe a similar trend as in the vanilla instantiation of GaussMark (cf. Figure 2(a)). In Figure 17(b), we demonstrate that by further reducing the rank of the watermarking space, we are able to tolerate significantly greater variance without affecting model performance on GSM−8K. This trend persists on all our models and metrics and is discussed at greater length, along with many more empirical observations. Finally, while the rank-reduction approach serves to occasionally improve the quality of generations with no sacrifice in watermark detectability, we observe that the rank-reduced watermarks investigated here are less robust to corruption than the vanilla instantiation of GaussMark (e.g., compare Figure 13 to Figure 22). Whether this is an inherent feature of the rank-reduction or a result of the specific parameters chosen is an interesting question for future research.

---

[1]An alternative approach more directly inspired by the results of Sharma et al. (2024) would be to first project the weight matrix onto its top principal components and then add a small amount of noise to the bottom components. While we did experiment with this technique, and observed that it was beneficial from the perspective of detectability and on SuperGLUE, we found that it performed extremely poorly on AlpacaEval−2.0 and thus do not report it here.

| Task | Phi | Phi (Unwatermarked) |
|------|-----|---------------------|
| **BoolQ** | 0.8514 ± 0.0062 | 0.8532 ± 0.0062 |
| **CB** | 0.0893 ± 0.0385 | 0.0893 ± 0.0385 |
| **COPA** | 0.8800 ± 0.0327 | 0.8700 ± 0.0338 |
| **MultiRC** | 0.3855 ± 0.0070 | 0.3296 ± 0.0068 |
| **ReCoRD** | 0.8689 ± 0.0033 | 0.8725 ± 0.0033 |
| **RTE** | 0.7509 ± 0.0260 | 0.7509 ± 0.0260 |
| **WiC** | 0.5768 ± 0.0196 | 0.5752 ± 0.0196 |
| **Winograd** | 0.7885 ± 0.0402 | 0.8269 ± 0.0373 |

*Table 6.* Performance of GaussMark on SuperGLUE(Phi3.5–Mini).

| Model | Watermarked Layer | Watermarked Weight | Variance ($\sigma^2$) | # PCs Dropped |
|-------|-------------------|--------------------|-----------------------|---------------|
| Mistral–7B | 30 | up_proj | 1e-05 | 1024 |
| Llama3.1–8B | 29 | down_proj | 1e-04 | 512 |
| Phi3.5–Mini | 31 | gate_up_proj | 3e-04 | 1024 |

*Table 7.* Selected watermarking hyperparameters.

### G.1. Detectability

While in Figure 17(a) we displayed the TPR at an FPR of 0.05, we here present the same at an FPR of 0.01 (Figure 18(a)) and the median p-values (Figure 18(b)) as the number of tokens increases. As in the case of the vanilla instantiation of GaussMark (cf. Appendix D), all detectability metrics broadly agree and demonstrate that the watermark remains detectable even with the rank-reduction.

Analogous to the exhaustive search over watermarking hyperparameters that we presented in Figures 7 to 9, we present in Figure 18 an abbreviated version of a similar search for the rank-reduced models, with a special focus on the effect that projecting away different ranks of PCs has on the median p-value. As predicted by our theory, projecting away more PCs leads to a less detectable watermark, both due to a decrement in the gradient norm and a reduction in the "effective dimension" of $\theta$. As in the vanilla instantiation, we see that increasing variance generally leads to a more detectable watermark, with the caveat that too large variances reduce the verisimilitude of the linear softmax approximation, and can thus lead GaussMark to be less detectable.

### G.2. Model Quality

We now report the effect that the rank-reduced instantiation of GaussMark has on model quality in each of our measurements thereof: SuperGLUE, AlpacaEval–2.0, GSM–8K, and token-level cross-entropy. As we did in Appendix E, we begin by demonstrating that our selected models, whose watermarking parameters are given in Table 7, do not suffer in comparison to the unwatermarked versions. The cross-entropy is reported in Table 3, whereas the other metrics are reported in Table 8, with detailed SuperGLUE results broken down by task in Tables 5, 9 and 11. In all cases, we see that we can find watermarking hyperparameters that do not significantly affect the model's performance while retaining detectability.

In addition to our model quality checks for the selected models, we report in Figure 20 the effects that rank reduction and variance have on GSM–8K performance for fixed weights and layers for each model. Along with Appendix G.2, these results demonstrate that rank reduction is an effective technique in allowing the watermarked models to tolerate greater variance $\sigma^2$ while maintaining quality.

### G.3. Robustness

Finally, we present the effect that our three token-level corruptions and one semantic level corruption have on the rank-reduced instantiation of GaussMark. The latter of these is displayed in the ROC curves in Figure 21, whereas the former are displayed as measured by the TPR at FPR $p = 0.05$ (Figure 22), median p-values (Figure 23), and as AUC under the ROC (Figure 24).

| Var | 1e-05 | 3e-05 | 1e-04 | 3e-04 | 1e-03 |
|---|---|---|---|---|---|
| Layer 31 | 0.3707 | 0.3306 | 0.2183 | 0.0000 | 0.0000 |
| Layer 30 | 0.4367 | 0.4405 | 0.4185 | 0.2494 | 0.0000 |
| Layer 29 | 0.4405 | 0.4443 | 0.4215 | 0.2904 | 0.0000 |
| Layer 28 | 0.4253 | 0.4139 | 0.3867 | 0.2669 | 0.0000 |
| Layer 25 | 0.4276 | 0.4139 | 0.3639 | 0.1706 | 0.0000 |
| Layer 20 | 0.4215 | 0.3571 | 0.1471 | 0.0053 | 0.0000 |
| Layer 10 | 0.3571 | 0.1903 | 0.0000 | 0.0000 | 0.0000 |

(a) GSM–8K performance of Mistral–7B (down_proj).

| Var | 1e-05 | 3e-05 | 1e-04 | 3e-04 | 1e-03 |
|---|---|---|---|---|---|
| Layer 31 | 0.3942 | 0.3237 | 0.1410 | 0.0182 | 0.0000 |
| Layer 30 | 0.4344 | 0.4397 | 0.3995 | 0.0462 | 0.0000 |
| Layer 29 | 0.4094 | 0.3958 | 0.3662 | 0.2411 | 0.0000 |
| Layer 28 | 0.4359 | 0.4230 | 0.3935 | 0.2381 | 0.0000 |
| Layer 25 | 0.4284 | 0.4291 | 0.3874 | 0.1797 | 0.0000 |
| Layer 20 | 0.4321 | 0.4102 | 0.3048 | 0.0409 | 0.0000 |
| Layer 10 | 0.3851 | 0.2760 | 0.0182 | 0.0053 | 0.0000 |

(b) GSM–8K performance of Mistral–7B (up_proj).

| Var | 1e-05 | 3e-05 | 1e-04 | 3e-04 | 1e-03 |
|---|---|---|---|---|---|
| Layer 31 | 0.4337 | 0.4139 | 0.3829 | 0.0015 | 0.0000 |
| Layer 30 | 0.4223 | 0.4086 | 0.3078 | 0.0000 | 0.0000 |
| Layer 29 | 0.4026 | 0.3533 | 0.2805 | 0.0000 | 0.0000 |
| Layer 28 | 0.4367 | 0.4337 | 0.0000 | 0.0000 | 0.0000 |
| Layer 25 | 0.4344 | 0.4230 | 0.3632 | 0.0129 | 0.0000 |
| Layer 20 | 0.4428 | 0.4344 | 0.3275 | 0.0008 | 0.0000 |
| Layer 10 | 0.4261 | 0.3692 | 0.0781 | 0.0068 | 0.0000 |

(c) GSM–8K performance of Mistral-7B (gate_proj).

*Figure 10.* Effect that GaussMark has on Mistral–7B's performance on GSM–8K for different layers and variances for (a) down_proj, (b) up_proj, and (c) gate_proj, colored by whether the watermarked model's performance is statistically indistinguishable from that of the base model (yellow) or not (purple).

Finally, we present the impact of our three token-level corruptions and one semantic-level corruption on the rank-reduced instantiation of GaussMark. The effect of semantic-level corruption is illustrated in the ROC curves shown in Figure 21, while the token-level corruptions are evaluated using the TPR at FPR $p = 0.05$ (Figure 22), median p-values (Figure 23), and AUC under the ROC (Figure 24). In all cases, we continue to see some robustness to both roundtrip translation and token-level corruptions, although if we compare these results to our earlier results for the vanilla instantiation of GaussMark, we see that the vanilla instantiation is generally more robust.

## H. Empirical Comparison with Other Watermarking Schemes

In this section, we compare the Soft Red List approach of Kirchenbauer et al. (2023a;b) with our GaussMark. As we discussed in the main text, the approach of Dathathri et al. (2024) does not provide a statistically valid watermark as implemented and thus we do not compare with it. While the approach of Kuditipudi et al. (2023) *does* provide statistically valid watermarks, the detection times as reported in the text are orders of magnitude greater than those of GaussMark; furthermore, for realistic scenarios, wherein a provider may wish to service many clients with similar prompts and diverse responses, the corresponding growth in detection time of Kuditipudi et al. (2023) is prohibitive (quadratic in terms of number of queries) and thus we restrict our empirical comparison to the unique pre-existing *practical* watermarking approach of

| Var | 1e-05 | 3e-05 | 1e-04 | 3e-04 | 1e-03 |
|---|---|---|---|---|---|
| Layer 31 | 0.5269 | 0.4587 | 0.3707 | 0.1205 | 0.0000 |
| Layer 30 | 0.6217 | 0.6224 | 0.6126 | 0.6126 | 0.5641 |
| Layer 29 | 0.6217 | 0.6285 | 0.6156 | 0.6058 | 0.5747 |
| Layer 28 | 0.6187 | 0.6156 | 0.6035 | 0.5997 | 0.5959 |
| Layer 25 | 0.6194 | 0.6141 | 0.6126 | 0.6042 | 0.5777 |
| Layer 20 | 0.6171 | 0.6270 | 0.6338 | 0.6171 | 0.5057 |
| Layer 10 | 0.6270 | 0.6065 | 0.5254 | 0.2775 | 0.0053 |

(a) GSM–8K performance of Llama3.1–8B (down_proj).

| Var | 1e-05 | 3e-05 | 1e-04 | 3e-04 | 1e-03 |
|---|---|---|---|---|---|
| Layer 31 | 0.6103 | 0.6133 | 0.5944 | 0.4829 | 0.0440 |
| Layer 30 | 0.6293 | 0.6361 | 0.6255 | 0.6133 | 0.3434 |
| Layer 29 | 0.6346 | 0.6323 | 0.6406 | 0.6338 | 0.6232 |
| Layer 28 | 0.6331 | 0.6285 | 0.6187 | 0.6111 | 0.6012 |
| Layer 25 | 0.6262 | 0.6293 | 0.6323 | 0.6118 | 0.5891 |
| Layer 20 | 0.6270 | 0.6331 | 0.6331 | 0.6111 | 0.4898 |
| Layer 10 | 0.6217 | 0.6035 | 0.5262 | 0.2775 | 0.0129 |

(b) GSM–8K performance of Llama3.1–8B (up_proj).

| Var | 1e-05 | 3e-05 | 1e-04 | 3e-04 | 1e-03 |
|---|---|---|---|---|---|
| Layer 31 | 0.6202 | 0.6133 | 0.6020 | 0.5898 | 0.5603 |
| Layer 30 | 0.6240 | 0.6247 | 0.6240 | 0.6232 | 0.6027 |
| Layer 29 | 0.6277 | 0.6277 | 0.6262 | 0.6187 | 0.6164 |
| Layer 28 | 0.6224 | 0.6217 | 0.6262 | 0.6187 | 0.6133 |
| Layer 25 | 0.6270 | 0.6331 | 0.6293 | 0.6202 | 0.5959 |
| Layer 20 | 0.6285 | 0.6179 | 0.6088 | 0.6149 | 0.5982 |
| Layer 10 | 0.6224 | 0.6050 | 0.5792 | 0.4587 | 0.1350 |

(c) GSM–8K performance of Llama3.1–8B (gate_proj).

*Figure 11.* Effect that GaussMark has on Llama3.1–8B's performance on GSM–8K for different layers and variances for (a) down_proj, (b) up_proj, and (c) gate_proj, colored by whether the watermarked model's performance is statistically indistinguishable from that of the base model (yellow) or not (purple).

which we are aware that has formal statistical guarantees.[2]

The Soft Red List approach described by Kirchenbauer et al. (2023a;b) selects a pseudo-random hash function based on the previous $m$ tokens to identify a random $\gamma$-fraction of the token space. A constant $\delta$ is then added to the log probabilities of these selected tokens before sampling. Under this scheme, the unwatermarked text is expected to exhibit, on average, $\gamma T$ many "upweighted" tokens, so if significantly more favored tokens appear in a given text it is likely that the text is watermarked; Kirchenbauer et al. (2023a) proposes a $z$-test to detect this signal in a statistically quantifiable way. Following prior empirical studies (Dathathri et al., 2024; Kuditipudi et al., 2023), we set $\gamma = 0.25$ and evaluate the approach with $\delta = 1.0$ and $\delta = 2.0$. Note that there is a tradeoff in increasing $\delta$; A higher $\delta$ strengthens the detectability of the watermark, though may lead to greater distortion of the generated text.

As in Kuditipudi et al. (2023), we refer to this scheme as KGW-1 and KGW-2 depending on whether $\delta = 1.0$ or $2.0$, respectively. In our timing plots (Figure 27), we refer to this approach simply as KGW. As the performance of KGW is not the main focus of this work, for the sake of saving expensive computational resources, we evaluated KGW only on 100 prompts as opposed to the 1K prompts we used to evaluate GaussMark. In Figures 25 and 26, we compare the detectability of GaussMark to KGW-1 and KGW-2 on Llama3.1–8B, Mistral–7B, and Phi3.5–Mini. These comparisons are based on metrics such as the median $p$-value and AUC of the ROC curve (Figure 25), as well as the fraction of significant detections at

---

[2]We observe that technically the guarantees of Kirchenbauer et al. (2023a;b) only give approximate $p$-values; we elide over this point and treat them as formal for the purposes of this comparison.

| Var | 1e-05 | 3e-05 | 1e-04 | 3e-04 | 1e-03 |
|---|---|---|---|---|---|
| Layer 31 | 0.8499 | 0.8469 | 0.8332 | 0.7695 | 0.6558 |
| Layer 30 | 0.8453 | 0.8438 | 0.8469 | 0.8415 | 0.7930 |
| Layer 29 | 0.8484 | 0.8400 | 0.7309 | 0.7316 | 0.7491 |
| Layer 28 | 0.8514 | 0.8499 | 0.8506 | 0.8560 | 0.8529 |
| Layer 25 | 0.8461 | 0.8453 | 0.8438 | 0.8453 | 0.8150 |
| Layer 20 | 0.8544 | 0.8552 | 0.8529 | 0.8567 | 0.8552 |
| Layer 10 | 0.8552 | 0.8544 | 0.8506 | 0.8537 | 0.8385 |

(a) GSM–8K performance of Phi3.5–Mini (down_proj).

| Var | 1e-05 | 3e-05 | 1e-04 | 3e-04 | 1e-03 |
|---|---|---|---|---|---|
| Layer 31 | 0.8506 | 0.8537 | 0.8544 | 0.8605 | 0.8408 |
| Layer 30 | 0.8537 | 0.8476 | 0.8506 | 0.8438 | 0.8415 |
| Layer 29 | 0.8469 | 0.8415 | 0.8385 | 0.8393 | 0.8173 |
| Layer 28 | 0.8499 | 0.8537 | 0.8499 | 0.8575 | 0.8529 |
| Layer 25 | 0.8400 | 0.8506 | 0.8484 | 0.8446 | 0.8218 |
| Layer 20 | 0.8476 | 0.8415 | 0.8423 | 0.8370 | 0.8180 |
| Layer 10 | 0.8461 | 0.8431 | 0.8309 | 0.8287 | 0.7544 |

(b) GSM–8K performance of Phi3.5–Mini (gate_up_proj).

*Figure 12.* Effect that GaussMark has on Phi3.5–Mini's performance on GSM–8K for different layers and variances for (a) down_proj and (b) gate_up_proj, a, colored by whether the watermarked model's performance is statistically indistinguishable from that of the base model (yellow) or not (purple).

| Model | SuperGLUE(Avg) | GSM8K (Acc) | Alpaca Eval (win rate) |
|---|---|---|---|
| Llama3.1–8B (LowRank) | 0.6992 ± 0.0337 | 0.6133 ± 0.0134 | 0.47 |
| Llama3.1–8B (Unwatermarked) | 0.7012 ± 0.0336 | 0.6300 ± 0.0133 | (0.46, 0.54) |
| Mistral–7B (LowRank) | 0.6796 ± 0.0336 | 0.4253 ± 0.0136 | 0.50 |
| Mistral–7B (Unwatermarked) | 0.6836 ± 0.0335 | 0.4291 ± 0.0136 | (0.49, 0.51) |
| Phi3.5–Mini (LowRank) | 0.6384 ± 0.0258 | 0.8537 ± 0.0097 | 0.47 |
| Phi3.5–Mini (Unwatermarked) | 0.6451 ± 0.0254 | 0.8423 ± 0.0100 | (0.46, 0.54) |

*Table 8.* Performance of GaussMark on various models.

FPR 0.05 and 0.01 (Figure 26), averaged across three seeds. Consistent with our experiments on detectability in Appendix D and Appendix G.1, we see that each of these metrics broadly agrees with the others.

The weaker of the soft Red list implementations (KGW-1) expresses broadly similar performance as GaussMark with some variation in the model and the choice of measurement. The stronger implementation (KGW-2) evinces superior detectability. In both cases, however, this detectability comes at a serious computational cost in generation time, as shown in Figure 27. While the detection times are broadly similar to those of GaussMark, the generation times are orders of magnitude greater. While in all cases we use 40GB NVIDIA A100s in our experiments, one reason for this vast disparity is the use of Huggingface's generation code (Wolf et al., 2020b), as opposed to the accelerated vLLM codebase (Kwon et al., 2023). While this may not seem like a fair comparison at first glance, one of the key advantages of GaussMark is that LM decoding acceleration code can be used out of the box to generate high-quality watermarked text as opposed to requiring bespoke implementations for each watermarking scheme and model. This is a significant advantage in practice, as it allows for the rapid deployment of GaussMark across a wide variety of models and use cases.

We also compare the quality of text generated by KGW-1 and KGW-2 to that of GaussMark using AlpacaEval–2.0, with the results summarized in Table 12. We see that in all cases either GaussMark or its rank-reduced version improves on the performance of KGW, sometimes significantly. The more distortionary scheme, KGW-2, fairs particularly poorly in this comparison. We reiterate that we did not conduct a fully exhaustive search over the possible parameter combinations to apply GaussMark and we suspect that improved performance could be had with further search.

Finally, we compare the robustness of KGW-1 and KGW-2 to GaussMark in Figures 28 and 29. In Figures 28(a), 28(c) and 28(e), we present the effects of adding random tokens to each watermark. Similarly, Figures 28(b), 28(d) and 28(f) illustrates the impact of removing random tokens, while Figures 29(a), 29(c) and 29(e) examines the effects of substituting random tokens. Across all cases, we evaluate the effect on the true positive rate (TPR) at a false positive rate (FPR) of 0.05, as our earlier experiments have shown that all measurements of detectability generally agree. We see that KGW-1 and

| Task | Llama (LowRank) | Llama (Unwatermarked) |
|---|---|---|
| BoolQ | 0.8205 ± 0.0067 | 0.8217 ± 0.0067 |
| CB | 0.5893 ± 0.0663 | 0.6250 ± 0.0653 |
| COPA | 0.8700 ± 0.0338 | 0.8700 ± 0.0338 |
| MultiRC | 0.5720 ± 0.0071 | 0.5720 ± 0.0071 |
| ReCoRD | 0.9221 ± 0.0026 | 0.9222 ± 0.0026 |
| RTE | 0.7076 ± 0.0274 | 0.7040 ± 0.0275 |
| WiC | 0.5063 ± 0.0198 | 0.5157 ± 0.0198 |
| Winograd | 0.6058 ± 0.0482 | 0.5865 ± 0.0485 |

*Table 9.* Performance of GaussMark on SuperGLUE(Llama3.1−8B LowRank).

| Task | Mistral (LowRank) | Mistral (Unwatermarked) |
|---|---|---|
| BoolQ | 0.8199 ± 0.0067 | 0.8205 ± 0.0067 |
| CB | 0.4821 ± 0.0674 | 0.5357 ± 0.0672 |
| COPA | 0.9200 ± 0.0273 | 0.9200 ± 0.0273 |
| MultiRC | 0.5693 ± 0.0071 | 0.5672 ± 0.0071 |
| ReCoRD | 0.9198 ± 0.0027 | 0.9219 ± 0.0026 |
| RTE | 0.6570 ± 0.0286 | 0.6823 ± 0.0280 |
| WiC | 0.5690 ± 0.0196 | 0.5674 ± 0.0196 |
| Winograd | 0.5000 ± 0.0493 | 0.4615 ± 0.0491 |

*Table 10.* Performance of GaussMark on SuperGLUE(Mistral−7B LowRank).

KGW-2 are significantly more robust to token-level corruptions than GaussMark. This result is not altogether surprising in light of the fact that we used a context window of a single token for the KGW experiments, which allows for significant textual distortion; indeed, Kuditipudi et al. (2023) report a considerable increase in perplexity when using KGW as well as a concurrent decline in the quality of the generated text, especially when prompted to follow instructions. Furthermore, as we described in Appendix F, these token-level attacks are not very realistic given the strong negative effects they have on text quality. A more realistic attack is the roundtrip translation attack, whose results we display in Figures 29(b), 29(d) and 29(f) as ROC curves, with corresponding AUC values summarized in Table 13. Here we see that GaussMark performs better than KGW-1 on Llama3.1−8B and Mistral−7B and comparably on Phi3.5−Mini. Although KGW-2 demonstrates improved robustness, we again note that the amount of distortion introduced by KGW-2 is unacceptably high, as reported by Kuditipudi et al. (2023).

## I. Theoretical Analysis

In Appendix I.1, we provide the missing details from Section 3, including a formal proof of the bound on the expected deviation under the null hypothesis for the Gaussian example. We then present the proof of Proposition I.2, which highlights the form of the hypothesis in $\mathbf{H_0}$ that is closest to $\mathbf{H_A}$. In Appendix I.2, we establish the statistical guarantees on the level of the test (Proposition 3.1). Next, Appendix I.3 contains proofs for the claims regarding the power of the test, with results on the linear softmax distribution detailed in Appendix I.3.1. Additionally, we discuss the role of entropy in ensuring the diversity of $\{\nabla \log p_\theta(y)\}_{y \in \mathcal{Y}}$. In Appendix I.3.2, we further relax the linear softmax assumption and consider strongly log-concave distributions. Finally, in Appendix F, we introduce a variant of GaussMark.Detect and GaussMark.Generate that is theoretically robust to random corruptions.

### I.1. Missing Details from Section 3

We start with giving an illustrative example using Gaussian distributions to explain why GaussMark works.

**Illustrative Example.** Suppose $\mathcal{X} = \mathcal{Y} = \mathbb{R}^d$ and let $p_\theta(y \mid x) = \mathcal{N}(\theta, \mathbb{I}_d)$. Because under $\mathbf{H_0}$, the watermarking key $\xi \sim \mathcal{N}(0, \sigma^2 \mathbb{I}_d)$ is independent of $y$, it is easy to see that $\psi(y, \xi \mid x) \sim \mathcal{N}(0, 1)$ for any fixed $x, y$. In particular,

| Task | Phi (LowRank) | Phi (Unwatermarked) |
|---|---|---|
| **BoolQ** | $0.8502 \pm 0.0062$ | $0.8532 \pm 0.0062$ |
| **CB** | $0.0893 \pm 0.0385$ | $0.0893 \pm 0.0385$ |
| **COPA** | $0.8700 \pm 0.0338$ | $0.8700 \pm 0.0338$ |
| **MultiRC** | $0.3195 \pm 0.0067$ | $0.3296 \pm 0.0068$ |
| **ReCoRD** | $0.8705 \pm 0.0033$ | $0.8725 \pm 0.0033$ |
| **RTE** | $0.7545 \pm 0.0259$ | $0.7509 \pm 0.0260$ |
| **WiC** | $0.5549 \pm 0.0197$ | $0.5752 \pm 0.0196$ |
| **Winograd** | $0.7981 \pm 0.0396$ | $0.8269 \pm 0.0373$ |

*Table 11.* Performance of GaussMark on SuperGLUE(Phi3.5–Mini LowRank).

| **Model** | KGW-1 | KGW-2 | GaussMark | GaussMark (Rank Reduced) |
|---|---|---|---|---|
| Llama3.1–8B | **.47** | .42 | .45 | **.47** |
| Mistral–7B | .47 | .43 | **.50** | **.50** |
| Phi3.5–Mini | .48 | .44 | **.49** | .47 |

*Table 12.* Comparison of win-rate in AlpacaEval–2.0 between KGW and GaussMark. Higher is better.

$\mathbb{E}_{\mathbf{H_0}}[\psi(y, \xi \mid x)] = 0$. On the other hand, under $\mathbf{H_A}$, $y \sim p_{\theta + \xi}(\cdot \mid x) = \mathcal{N}(\theta + \xi, \mathbb{I}_d)$ and Lemma I.1 implies that

$$\mathbb{E}_{\mathbf{H_A}}[\psi(y, \xi \mid x)] \gtrsim \frac{\sigma^2 \sqrt{d}}{\sqrt{1 + \sigma^2}}$$

for $\sigma \geq 1/d$. Applying a standard concentration of Gaussian random variables yields bounds on the power of the test, with higher dimensions leading to an improved test.

**Lemma I.1.** *Let $\xi \sim \mathcal{N}(0, \sigma^2 \mathbb{I}_d)$ and $p_\theta(y \mid x) \sim \mathcal{N}(\theta, \mathbb{I}_d)$. Then, with probability at least $1 - \delta$,*

$$\psi(y, \xi \mid x) \gtrsim \frac{\sigma^2}{\sqrt{1 + \sigma^2}} \cdot \sqrt{d}$$

*as long as $\sigma\sqrt{d} \gtrsim \sqrt{\log(1/\delta)}$. In particular, if $\sigma\sqrt{d} \gtrsim \sqrt{\log(1/\min(\alpha, \beta))}$, then $\psi$ yields a test with level $\alpha$ and power $1 - \beta$.*

**Proof.** Note that $\nabla \log p_\theta(y \mid x) = y - \theta$. Letting $z \sim \mathcal{N}(0, \mathbb{I}_d)$, we see that under $\mathbf{H_A}$, we may take $y = \theta + \xi + z$. Thus,

$$\langle \xi, \nabla \log p_\theta(y \mid x) \rangle = \|\xi\|^2 + \langle \xi, z \rangle.$$

By classical Gaussian concentration (e.g. Vershynin (2018, Theorem 3.1.1)), it holds that with probability at least $1 - \delta$

$$\|\xi\|^2 \geq \sigma^2 d - C\sqrt{\sigma^2 d \log(1/\delta)} \qquad \text{and} \qquad \|\xi\| \leq \sqrt{\sigma^2 d} + C\sqrt{\log(1/\delta)}.$$

Noting that $\langle \xi, z \rangle$ is distributed as a centred one-dimensional Gaussian with variance $\|\xi\|^2$, we see that

$$\langle \xi, \nabla \log p_\theta(y \mid x) \rangle \geq \sigma^2 d - C\sqrt{\sigma^2 d} \cdot \log(1/\delta),$$

where we allow the universal constant $C$ to change from line to line. On the other hand, it holds by the same concentration inequality that with probability at least $1 - \delta$,

$$\|\xi + z\| \leq \|\xi\| + \|z\| \leq \sqrt{C(1 + \sigma^2) d \log(1/\delta)}.$$

Combining these events and taking a union bound, we see that with probability at least $1 - \delta$,

$$\psi(y, \xi \mid x) \geq \frac{\sigma^2 d - C\sqrt{\sigma^2 d} \cdot \log(1/\delta)}{\sqrt{C(1 + \sigma^2) d \log(1/\delta)}} \geq c \frac{\sigma^2}{\sqrt{1 + \sigma^2}} \cdot \sqrt{d} - C\sqrt{\frac{\sigma^2 \log(1/\delta)}{1 + \sigma^2}}.$$

| Model | GaussMark | KGW-1 | KGW-2 |
|-------|-----------|-------|-------|
| Llama3.1–8B | 0.7983 | 0.7221 | 0.8519 |
| Mistral–7B | 0.7284 | 0.7034 | 0.8618 |
| Phi3.5–Mini | 0.7198 | 0.7349 | 0.8587 |

*Table 13.* Comparison of AUC after roundtrip translation through French for GaussMark, KGW-1, and KGW-2 on various models. Higher numbers are better.

Note that if $\sigma d \gtrsim \sqrt{\log(1/\delta)}$, the first term dominates the second concludes the proof of the first statement.

For the second statement, note that under the null hypothesis, by Proposition 3.1, $\psi$ is normally distributed and thus we may set $\tau_\alpha \lesssim \sqrt{\log(1/\alpha)}$. The result follows. $\qquad\square$

Next, we outline the proof of Proposition I.2, which characterizes the hypothesis in $\mathbf{H_0}$ that is nearest to $\mathbf{H_A}$.

**Proposition I.2.** *Let $\{p_{\theta'}\}_{\theta' \in \mathbb{R}^d}$ be a family of measures on $\mathcal{Y}$, $x \in \mathcal{X}$, $\theta \in \mathbb{R}^d$, and $\nu \in \Delta(\mathbb{R}^d)$ be fixed. Suppose $\mathbf{H_0} \coloneqq \{\nu \otimes \mu \mid \mu \in \Delta(\mathcal{Y})\}$ is composite and $\mathbf{H_A}$ is simple, such that $\xi \sim \nu$ and $y \sim p_{\theta+\xi}$. Then the projection of $\mathbf{H_A}$ onto $\mathbf{H_0}$ in KL divergence is given by $\nu \otimes \mathbb{E}_{\xi' \sim \nu}[p_{\theta+\xi'}(\cdot \mid x)]$, i.e.,*

$$\inf_{p \in \mathbf{H_0}} \mathrm{D_{KL}}\left(\mathbf{H_A} \parallel p\right) = \mathrm{D_{KL}}\left(p_{\theta+\xi}(\cdot \mid x) \parallel \mathbb{E}_{\xi' \sim \nu}[p_{\theta+\xi'}(\cdot \mid x)]\right).$$

**Proof of Proposition I.2.** Let $x \in \mathcal{X}$ be fixed. Further, let $\nu \in \Delta(\mathbb{R}^d)$ and $\mu \in \Delta(\mathcal{Y})$ be arbitrary. For any $\xi \in \mathbb{R}^d$, we have

$$\mathrm{D_{KL}}\left(p_{\theta+\xi}(\cdot \mid x) \parallel \mu\right) - \mathrm{D_{KL}}\left(p_{\theta+\xi}(\cdot \mid x) \parallel \mathbb{E}_{\xi' \sim \nu}[p_{\theta+\xi'}(\cdot \mid x)]\right)$$

$$= \mathbb{E}_{y \sim p_{\theta+\xi}}\left[\log\left(\frac{\mathbb{E}_{\xi' \sim \nu}[p_{\theta+\xi'}(y \mid x)]}{\mu(y)}\right)\right]$$

$$= \mathbb{E}_{p_{\theta+\xi}}\left[\frac{p_{\theta+\xi}(y \mid x)}{\mathbb{E}_{\xi' \sim \nu}[p_{\theta+\xi'}(y \mid x)]} \log\left(\frac{\mathbb{E}_{\xi' \sim \nu}[p_{\theta+\xi'}(y \mid x)]}{\mu(y)}\right)\right].$$

Taking the expectation with respect to $\xi \sim \nu$ and noting that $\xi \sim \nu$ for both $p \in \mathbf{H_0}$ and $\mathbf{H_A}$, if $p = \nu \otimes \mu$ and $p^\star = \nu \otimes \mathbb{E}_{\xi' \sim \nu}[p_{\theta+\xi'}(\cdot \mid x)]$, then

$$\mathrm{D_{KL}}\left(\mathbf{H_A} \parallel p\right) - \mathrm{D_{KL}}\left(\mathbf{H_A} \parallel p^\star\right) = \mathbb{E}_{\xi \sim \nu}\left[\mathrm{D_{KL}}\left(p_{\theta+\xi}(\cdot \mid x) \parallel \mu\right) - \mathrm{D_{KL}}\left(p_{\theta+\xi}(\cdot \mid x) \parallel \mathbb{E}_{\xi' \sim \nu}[p_{\theta+\xi'}]\right)\right]$$

$$= \mathbb{E}_{\xi \sim \nu}\left[\mathbb{E}_{p_{\theta+\xi}}\left[\frac{p_{\theta+\xi}(y \mid x)}{\mathbb{E}_{\xi' \sim \nu}[p_{\theta+\xi'}(y \mid x)]} \log\left(\frac{\mathbb{E}_{\xi' \sim \nu}[p_{\theta+\xi'}(y \mid x)]}{\mu(y)}\right)\right]\right]$$

$$= \mathrm{D_{KL}}\left(\mathbb{E}_{\xi' \sim \nu}[p_{\theta+\xi'}(\cdot \mid x)] \parallel \mu\right) \geq 0,$$

where the first equality follows from the chain rule. $\qquad\square$

We next provide the series of approximations leading to (3).

**Proof Sketch for** (3) Note that using the first-order Taylor's approximation for $\log p_{\theta+\xi}(y \mid x)$ around $\log p_\theta(y \mid x)$ in , we have

$$\log p_{\theta+\xi}(y \mid x) - \log \mathbb{E}_{\xi' \sim \nu}[p_{\theta+\xi'}(y \mid x)] = \log p_{\theta+\xi}(y \mid x) - \log p_\theta(y \mid x) - \log \mathbb{E}_{\xi' \sim \nu}\left[\frac{p_{\theta+\xi'}(y \mid x)}{p_\theta(y \mid x)}\right]$$

$$\approx \langle \xi, \nabla_\theta \log p_\theta(y \mid x) \rangle - \log \mathbb{E}_{\xi' \sim \nu}\left[\exp\left(\langle \xi', \nabla_\theta \log p_\theta(y \mid x) \rangle\right)\right]$$

$$= \langle \xi, \nabla_\theta \log p_\theta(y \mid x) \rangle - \frac{\sigma^2 \|\nabla_\theta \log p_\theta(y \mid x)\|^2}{2}$$

$$\approx \langle \xi, \nabla_\theta \log p_\theta(y \mid x) \rangle,$$

where the third line uses the MGF for Gaussian random variables, and the approximation in the last line holds as $\sigma \to 0$.

## I.2. Level of GaussMark

In this section, we provide the proof of Proposition 3.1 that demonstrates that GaussMark produces statistically valid p-values and qualifies as an *exact test*, with almost no underlying assumptions.

**Proof of Proposition 3.1.** Recall that under the null hypothesis $\mathbf{H_0}$, for any $x \in \mathcal{X}$, the key and the generated text $y$ are independent of each other, i.e., $(\xi, y) \sim \mathcal{N}(0, \sigma^2 \mathbb{I}_d) \otimes q$ for some $q \in \Delta(\mathcal{Y})$. Thus, the level of the test is

$$
\begin{aligned}
\mathrm{Pr}_{\mathbf{H_0}}(\psi(y, \xi \mid x) = 1) &= \mathbb{E}_{\xi \sim \mathcal{N}(0, \sigma^2 \mathbb{I}_d),\ y \sim q}\left[ \mathbb{I}\left\{ \frac{\langle \xi, \nabla \log p_\theta(y \mid x) \rangle}{\sigma \| \nabla \log p_\theta(y \mid x) \|} > \tau_\alpha \right\} \right] \\
&= \mathbb{E}_{y \sim q}\left[ \mathrm{Pr}_\xi\left( \frac{\langle \xi, \nabla \log p_\theta(y \mid x) \rangle}{\sigma \| \nabla \log p_\theta(y \mid x) \|} > \tau_\alpha \right) \right] \\
&= \mathbb{E}_{y \sim q}[\mathrm{Pr}_\xi(\psi(y, \xi \mid x) \geq \tau_\alpha)] \\
&= 1 - \Phi(\tau_\alpha) = \alpha,
\end{aligned}
$$

where in the last line, we used the fact that $\psi(y, \xi \mid x) = \frac{\langle \xi, \nabla \log p_\theta(y|x) \rangle}{\sigma \| \nabla \log p_\theta(y|x) \|} \sim \mathcal{N}(0, 1)$ for any vector $\nabla \log p_\theta(y \mid x)$. The last equality simply plugs in $\tau_\alpha = \Phi^{-1}(1 - \alpha)$, concluding the proof of the first statement. For the second statement, because $\psi(y, \xi \mid x) \sim \mathcal{N}(0, 1)$, it is immediate that $\Phi(\psi(y, \xi \mid x)) \sim \mathrm{Uniform}([0, 1])$, and thus $1 - \Phi(\psi(y, \xi \mid x))$ is a valid $p$-value. □

## I.3. Power of GaussMark

We first note the following supporting technical lemma bounding the expected value of Gaussian density under a halfspace constraint.

**Lemma I.3.** *Let $a, \gamma \in \mathbb{R}$ with $\gamma \geq 0$, and $u \in \mathbb{R}^d$. Let $Z \sim \mathcal{N}(0, \mathbf{I}_d)$. Then,*

$$
\mathbb{E}\left[ \mathbf{I}[\langle Z, u \rangle \leq a] e^{-\frac{\gamma}{2} \|Z\|^2} \right] = (1 + \gamma)^{-d/2} \cdot \Phi\left( \frac{\sqrt{1 + \gamma} \cdot a}{\|u\|} \right).
$$

**Proof.** Note that

$$
\begin{aligned}
\mathbb{E}\left[ \mathbf{I}[\langle Z, u \rangle \leq a] e^{-\frac{\gamma}{2} \|Z\|^2} \right] &= (2\pi)^{-d/2} \int_{\{z\ :\ \langle z, u \rangle \leq a\}} e^{-\frac{\gamma + 1}{2} \|z\|^2} \mathrm{d}z \\
&= (1 + \gamma)^{-d/2} \cdot \int_{\{z\ :\ \langle z, u \rangle \leq a\}} \left( \frac{2\pi}{1 + \gamma} \right)^{-d/2} e^{-\frac{1}{2(1+\gamma)^{-1}} \|z\|^2} \mathrm{d}z \\
&= (1 + \gamma)^{-d/2} \cdot \mathbb{P}\left( \langle Z/\sqrt{1 + \gamma}, u \rangle \leq a \right) \\
&= (1 + \gamma)^{-d/2} \cdot \mathbb{P}\left( \langle Z, u \rangle \leq a\sqrt{1 + \gamma} \right) \\
&= (1 + \gamma)^{-d/2} \cdot \Phi\left( \frac{a \cdot \sqrt{1 + \gamma}}{\|u\|} \right),
\end{aligned}
$$

where the equality in the third line above is by noticing that the term inside the integral corresponds to the density function of $\frac{Z}{\sqrt{1+\gamma}}$. The last line follows by observing that $\langle Z, u \rangle \sim \mathcal{N}(0, \|u\|^2)$ for $Z \sim \mathcal{N}(0, \mathbf{I}_d)$. □

### I.3.1. Linear Softmax Distributions

Throughout this subsection, we use the notation $\mathbb{E}_\theta[\cdot]$ to denote $\mathbb{E}_{y \sim p_\theta(\cdot | x)}[\cdot]$.

**Compatibility of softmax model with generation process of language models** In particular, for any sequence of tokens $y = v_{1:T}$, if the conditional probability $p_\theta(v_t \mid x \circ v_{<t})$ is a linear softmax model for each $t \leq T$, then the overall model

$$
p_\theta(y \mid x) = \prod_{t=1}^T p_\theta(v_t \mid x \circ v_{<t}) = \prod_{t=1}^T \frac{e^{\langle \theta, \varphi_t(v_t | x \circ v_{<t}) \rangle}}{\sum_{v' \in \mathcal{V}} e^{\langle \theta, \varphi_t(v' | x \circ v_{<t}) \rangle}}
$$

$$= \frac{e^{\langle\theta,\Sigma_t\,\varphi_t(v_t|x\circ v_{<t})\rangle}}{\sum_{y'\in\mathcal{Y}}e^{\langle\theta,\Sigma_t\,\varphi_t(v'_t|x\circ v'_{<t})\rangle}}$$

is also a linear softmax model. However, note that throughout the paper, we focus on softmax modeling at the level of the sequence $y$.

**Lemma I.4.** *Let $p_\theta(y\mid x)$ denote a linear softmax policy in dimension $d$ (Definition 3.2) w.r.t. feature mapping $\varphi$. Then,*

$$\frac{p_{\theta+\xi}(y\mid x)}{p_\theta(y\mid x)} = \frac{e^{\langle\xi,\nabla\log p_\theta(y|x)\rangle}}{\mathbb{E}\left[e^{\langle\xi,\nabla\log p_\theta(y|x)\rangle}\right]}$$

**Proof.** Fix any $x\in\mathcal{X}$, and define the normalizing constant

$$Z(\theta\mid x) := \sum_{y\in\mathcal{Y}}e^{\langle\theta,\varphi(y|x)\rangle}.$$

Thus, $\log(p_\theta(y\mid x)) = \langle\theta,\varphi(y\mid x)\rangle - \log Z(\theta\mid x)$. A few simple calculations show that for any $y\in\mathcal{Y}$,

$$\nabla\log p_\theta(y\mid x) = \varphi(y\mid x) - \mathbb{E}_\theta[\varphi(y\mid x)] \tag{5}$$

This implies that

$$\begin{aligned}
\log p_{\theta+\xi}(y\mid x) - \log p_\theta(y\mid x) &= \langle\theta+\xi,\varphi(y\mid x)\rangle - \log Z(\theta+\xi\mid x) - \langle\theta,\varphi(y\mid x)\rangle + \log Z(\theta\mid x)\\
&= \log Z(\theta\mid x) - \log Z(\theta+\xi\mid x) + \langle\xi,\varphi(y\mid x)\rangle\\
&= \log Z(\theta\mid x) - \log Z(\theta+\xi\mid x) + \langle\xi,\mathbb{E}_\theta[\varphi(y\mid x)]\rangle + \langle\xi,\nabla\log p_\theta(y\mid x)\rangle,
\end{aligned}$$

where the last line plugs in (5) for $\nabla\log p_\theta(y\mid x)$. Next, note that

$$\log Z(\theta\mid x) - \log Z(\theta+\xi\mid x) = -\log\left(\frac{\sum_{y\in\mathcal{Y}}e^{\langle\xi,\varphi(y|x)\rangle}\cdot e^{\langle\theta,\varphi(y|x)\rangle}}{Z(\theta\mid x)}\right) = -\log\left(\mathbb{E}_\theta\left[e^{\langle\xi,\varphi(y|x)\rangle}\right]\right).$$

Combining the above two bounds, we get

$$\begin{aligned}
\log p_{\theta+\xi}(y\mid x) - \log p_\theta(y\mid x) &= \langle\xi,\nabla\log p_\theta(y\mid x)\rangle + \langle\xi,\mathbb{E}_\theta[\varphi(y\mid x)]\rangle - \log\left(\mathbb{E}_\theta\left[e^{\langle\xi,\varphi(y|x)\rangle}\right]\right)\\
&= \langle\xi,\nabla\log p_\theta(y\mid x)\rangle - \log\left(\mathbb{E}_\theta\left[e^{\langle\xi,\varphi(y|x)-\mathbb{E}_\theta[\varphi(y|x)]\rangle}\right]\right)\\
&= \langle\xi,\nabla\log p_\theta(y\mid x)\rangle - \log\left(\mathbb{E}_\theta\left[e^{\langle\xi,\nabla\log p_\theta(y|x)\rangle}\right]\right),
\end{aligned}$$

where the last equality again uses (5). This concludes the proof. $\quad\square$

We next prove Proposition 3.3, which establishes bounds on the power of our test in GaussMark.Detect under the linear softmax modeling assumption.

**Proof of Proposition 3.3.** Proposition 3.1 shows that $\psi$ yields a test at level $\alpha$. Note that the power of the test

$$\begin{aligned}
\mathrm{Pr}_{\mathbf{H_A}}\left(\psi(y,\xi\mid x)\le\tau_\alpha\right) &= \mathbb{E}_\xi\left[\mathbb{E}_{y\sim p_{\theta+\xi}}[\mathbb{I}\{\psi(y,\xi\mid x)\le\tau_\alpha\}]\right]\\
&= \mathbb{E}_\xi\left[\mathbb{E}_{y\sim p_\theta}\left[\frac{p_{\theta+\xi}(y\mid x)}{p_\theta(y\mid x)}\cdot\mathbb{I}\{\psi(y,\xi\mid x)\le\tau_\alpha\}\right]\right]\\
&= \mathbb{E}_\xi\left[\mathbb{E}_{y\sim p_\theta}\left[\frac{e^{\langle\xi,\nabla\log p_\theta(y|x)\rangle}}{\mathbb{E}_{y\sim p_\theta}\left[e^{\langle\xi,\nabla\log p_\theta(y|x)\rangle}\right]}\cdot\mathbb{I}\{\psi(y,\xi\mid x)\le\tau_\alpha\}\right]\right],
\end{aligned}$$

where the last line plugs in the corresponding form for the density ratio $p_{\theta+\xi}(y|x)/p_\theta(y|x)$, obtained by simply utilizing the properties of linear-softmax distributions (Lemma I.4 in the appendix). $\quad\square$

As mentioned in the main body of the paper in the discussion proceeding Proposition 3.3, for the linear softmax model, entropy plays a key role in ensuing diversity of the vectors $\nabla \log p_\theta(y \mid x)_{y \in \mathcal{Y}}$ on the unit sphere, when the feature set $\{\varphi(y)\}_{y \in \mathcal{Y}}$ is sufficiently rich. In the following, we provide intuition for this claim. Note that for any $y$, under the linear softmax modeling,

$$\nabla \log p_\theta(y \mid x) = \varphi(y \mid x) - \mathbb{E}_{y \sim p_\theta}\left[\varphi(y \mid x)\right].$$

Thus, the spread of the set of vectors $\{\nabla \log p_\theta(y \mid x)\}_{y \in \mathcal{Y}}$ is controlled by the spread of the vectors $\{\varphi(y \mid x)\}_{y \in \mathcal{Y}}$. Recall that $\|\mathcal{Y}\| \le c$ for some $c > 0$. Consequently, $\|\nabla \log p_\theta(y \mid x)\| \le 2c$. On the other hand, the lower bound on $\|\nabla \log p_\theta(y \mid x)\|$ is influenced by the entropy of $p_\theta$.

If the set $\{\varphi(y \mid x)\}$ is well-spread on the sphere and $p_\theta$ has large entropy, then we expect $\mathbb{E}_{y \sim p_\theta}\left[\varphi(y \mid x)\right] \approx \vec{0}$. As a result, $\langle \xi, \nabla \log p_\theta(y \mid x) \rangle \approx \langle \xi, \varphi(y \mid x) \rangle \approx \|\xi\|$ when $\{\varphi(y \mid x)\}$ covers the sphere. Furthermore, this implies that $\|\nabla \log p_\theta(y \mid x)\| \gtrsim c$. These approximations, however, break down when the entropy is small —for instance if $p_\theta(\cdot \mid x)$ is an atom on some $y' \in \mathcal{Y}$—then $\nabla \log p_\theta(y' \mid x) = 0$, and we do not have a lower bound on $\sup_y \|\nabla \log p_\theta(y \mid x)\|$.

The above discussion gives intuition for why we can expect $r \le \|\nabla \log p_\theta(y \mid x)\| \le R$ when the underlying distribution $p_\theta$ exhibits high entropy. Corollary I.5 quantifies the degree to which these idealized conditions hold and establishes bounds on the test's power under certain mild assumptions. We now proceed with its proof.

**Corollary I.5.** *Let $\alpha \in (0,1)$, input prompt $x \in \mathcal{X}$, and suppose that all the conditions of Proposition 3.3 hold. For any $\xi \in \mathbb{R}^d$ and $\delta \in (0,1)$, let $\widetilde{\Gamma}_{\widetilde{\alpha}}(\xi \mid \theta, x)$ denote the $(1 - \delta)$th quantile of the random variable $\langle \xi, \nabla \log p_\theta(y \mid x) \rangle$, where $y \sim p_\theta(\cdot \mid x)$. Suppose there exists $\widetilde{\alpha} \in (0,1)$ such that*

$$\Lambda(\theta, x) := -\sup_\sigma \frac{1}{\sigma} \log\left(\mathbb{E}_{\xi \sim \mathcal{N}(0, \sigma^2 \mathbf{I}_d), y \sim p_\theta}\left[\exp\left(\sigma\|\nabla \log p_\theta(y \mid x)\| - \widetilde{\Gamma}_{\widetilde{\alpha}}(\xi \mid \theta, x)\right)\right]\right), \tag{6}$$

*is non-negative. Then, for any $\beta \in (0,1)$, the test in Algorithm 2 has power $1 - \beta$, for*

$$\sigma \ge \frac{1}{\Lambda(\theta, x)} \log\left(\frac{1}{\widetilde{\alpha}\beta}\right).$$

**Proof of Corollary I.5.** Starting from the result of Proposition 3.3, note that under softmax parameterization, the power of the test, denoted by $\widehat{\beta}$, is given by

$$\widehat{\beta} = \mathbb{E}_{\xi \sim \mathcal{N}(0, \sigma^2 \mathbb{I}_d)}\left[\frac{\mathbb{E}_\theta\left[e^{\langle \xi, \nabla \log p_\theta(y|x) \rangle} \cdot \mathbb{I}\left[\langle \xi, \nabla \log p_\theta(y \mid x) \rangle \le \sigma\tau_\alpha \|\nabla \log p_\theta(y \mid x)\|\right]\right]}{\mathbb{E}_\theta\left[e^{\langle \xi, \nabla \log p_\theta(y|x) \rangle}\right]}\right], \tag{7}$$

We can upper bound the numerator as:

$$\mathbb{E}_\theta\left[e^{\langle \xi, \nabla \log p_\theta(y|x) \rangle} \cdot \mathbb{I}\left[\langle \xi, \nabla \log p_\theta(y \mid x) \rangle \le \sigma\tau_\alpha \|\nabla \log p_\theta(y \mid x)\|\right]\right] \le \mathbb{E}_\theta\left[e^{\sigma\tau_\alpha \|\nabla \log p_\theta(y|x)\|}\right].$$

Next, for the denominator, note that

$$\begin{aligned}
\mathbb{E}_\theta\left[e^{\langle \xi, \nabla \log p_\theta(y|x) \rangle}\right] &\ge \mathbb{E}_\theta\left[e^{\langle \xi, \nabla \log p_\theta(y|x) \rangle} \cdot \mathbb{I}\left[\langle \xi, \nabla \log p_\theta(y \mid x) \rangle > \widetilde{\Gamma}_{\widetilde{\alpha}}(\xi \mid \theta, x)\right]\right] \\
&\ge e^{\widetilde{\Gamma}_{\widetilde{\alpha}}(\xi|\theta, x)} \cdot \Pr\left(\langle \xi, \nabla \log p_\theta(y \mid x) \rangle > \widetilde{\Gamma}_{\widetilde{\alpha}}(\xi \mid \theta, x)\right) \\
&= e^{\widetilde{\Gamma}_{\widetilde{\alpha}}(\xi|\theta, x)} \cdot \widetilde{\alpha},
\end{aligned}$$

where the first inequality simply ignores the events where the indicator is $0$. The second inequality follows by using the condition inside the indicator in the exponent. Finally, the last inequality simply uses the fact that $\widetilde{\Gamma}_{\widetilde{\alpha}}(\xi \mid \theta, x)$ is the $(1 - \widetilde{\alpha})$th quantile of the random variable $\langle \xi, \nabla \log p_\theta(y \mid x) \rangle$, and thus

$$\Pr\left(\langle \xi, \nabla \log p_\theta(y \mid x) \rangle > \widetilde{\Gamma}_{\widetilde{\alpha}}(\xi \mid \theta, x)\right) = \widetilde{\alpha}.$$

Using the above bounds in (7), we get that

$$\Pr_{\mathbf{H_A}}\left(\psi(y, \xi \mid x) \le \sigma\tau_\alpha\right) \le \frac{1}{\widetilde{\alpha}} \cdot \mathbb{E}_\xi\left[\mathbb{E}_\theta\left[e^{\sigma\|\nabla \log p_\theta(y|x) - \widetilde{\Gamma}_{\widetilde{\alpha}}(\xi|\theta, x)\|}\right]\right]$$

$$\leq \frac{e^{-\sigma \Lambda(\theta,x)}}{\widetilde{\alpha}},$$

where the last line uses (6).

We conclude the proof by observing that the term on the right-hand side above is smaller than $\beta$ when $\sigma \geq \frac{1}{\Lambda(\theta,x)} \log\left(\frac{1}{\widetilde{\alpha}\beta}\right)$

$\square$

The following lemma demonstrates that the final probability distribution over the token space $\mathcal{V}$ can be expressed as a non-linear softmax distribution, where $\theta$ represents the weights of any feedforward layer in the model architecture. This result provides justification for the discussion following 3.2 and clarifies why, when $\|\xi\|$ is small, a linear softmax model serves as a good approximation.

**Lemma I.6.** *Let $\mathcal{V}$ be a token space, and suppose $\theta$ denote the parameters in a feedforward layer of a language model to be watermarked. Then, $p_\theta(y \mid x) \propto \exp(\chi(\theta,x), \varphi(v))$ for some fixed feature maps $\chi$ and $\varphi$.*

**Proof.** Consider an input prompt $x$, and suppose we watermark the parameters $\theta$ at a feedforward layer with index $\ell$ that uses the ReLU activation function. Suppose the input to this layer is given by $g_{\text{in}}(x)$ where the non-convex function $g_{\text{in}}$ hides all the parameters from layers $1,\ldots,\ell-1$ which are hidden since they are unaffected by the watermarking procedure. Thus, the output of the $\ell$th layer is given by $\text{ReLU}(\theta^\top g_{\text{in}}(x))$. The future laters (including feedforward and attention layers) would simply process this further into the logits $g_{\text{fin}}(\text{ReLU}(\theta^\top g_{\text{in}}(x)))$, where the function $g_{\text{fin}}$ is non-convex and hides all the parameters in the layers $\ell+1,\ldots,L$ that are again hidden since the watermarking procedure does not affect them. Finally, the output token is generated by sampling from a softmax distribution given by $p_\theta(v \mid x) \propto \exp(g_{\text{fin}}(\text{ReLU}(\theta^\top g_{\text{in}}(x)))^\top \varphi(v))$ for $v \in \mathcal{V}$, where $\varphi$ denotes the one-hot feature vectors corresponding to token $v$. Defining $\chi(\theta,x) = g_{\text{fin}}(\text{ReLU}(\theta^\top g_{\text{in}}(x)))$ concludes the proof. A similar proof can be given for other non-linearities such as GELU, Swish, or SwiGLU. $\square$

### I.3.2. STRONGLY LOG-CONCAVE DISTRIBUTIONS

While the linear softmax model is useful for modeling the effect of noise added to a feedforward layer in a single transformer block in our experiments, we can also bound the power of our method under significantly more general assumptions. In the following, we provide a bound on the power of the test when the distribution $p_\theta$ is strongly log-concave:

**Definition I.7** (Strong log-concavity). *For any $x \in \mathcal{X}$ and $y \in \mathcal{Y}$, the map $\theta \mapsto \log p_\theta(y \mid x)$ is $\eta$-strongly log-concave if for any $\xi \in \mathbb{R}^d$,*

$$\log p_{\theta+\xi}(y \mid x) \leq \log p_\theta(y \mid x) + \langle \xi, \nabla \log p_\theta(y \mid x)\rangle - \frac{\eta}{2}\|\xi\|^2.$$

Various probability distributions satisfy the strong log-concavity assumption above. For example, the $d$-dimensional multivariate Gaussian $p_\theta = \mathcal{N}(\theta, \Sigma)$ is strongly log-concave with $\eta = \lambda_{\min}(\Sigma)$. Furthermore, the linear softmax model with feature map $\varphi$ is (locally) $\eta$-strongly concave with $\eta = \lambda_{\min}(\text{Cov}_{y\sim p_\theta}[\varphi(y \mid x)])$. The following theorem lower bounds the power of our test when $p_\theta$ is strongly log-concave. Again, for ease of notation, we will use $\mathbb{E}_\theta[\cdot]$ to denote $\mathbb{E}_{y\sim p_\theta(\cdot|x)}[\cdot]$.

**Theorem I.8.** *Let $\alpha, \beta \in (0,1)$ and $\tau_\alpha = \Phi^{-1}(1-\alpha)$, where $\Phi$ is the CDF of the standard normal random variable, and suppose for all $x \in \mathcal{X}$ and $y \in \mathcal{Y}$, $p_\theta(y \mid x)$ is $\eta$-strongly concave w.r.t. $\theta$. Then, for any $x \in \mathcal{X}$, the test $\mathbb{I}\left[\frac{\langle \xi, \nabla \log p_\theta(y|x)\rangle}{\sigma\|\nabla \log p_\theta(y|x)\|} > \tau_\alpha\right]$, utilized in Algorithm 2, has power $1 - \beta$ whenever*

$$d \geq \frac{2\log\left(\mathbb{E}_\theta\left[e^{\tau_\alpha \sigma\|\nabla \log p_\theta(y|x)\|}\right]\right) + 2\log\left(\frac{1}{\beta}\right)}{\log(1+\eta\sigma^2)}.$$

In particular, let the function $\Lambda : \Theta \times \mathcal{X} \to \mathbb{R}$ de defined such that

$$\Lambda(\theta,x) := \sup_\lambda \frac{1}{\lambda}\exp(\lambda\|\nabla \log p_\theta(y \mid x)\|),$$

then the power of the level $\alpha$ test $\varphi$ is at least $1 - \beta$, whenever

$$d \geq \frac{2\sigma\tau_\alpha\Lambda(\theta,x) + 2\log\left(\frac{1}{\beta}\right)}{\log(1+\eta\sigma^2)}.$$

The key idea in the proof of Theorem I.8 is to use strong log-concavity once we have changed the measure to $y \sim p_\theta$ followed by Lemma I.3 that measures the expected value of a gaussian density function, under halfspace constraints.

**Proof of Theorem I.8.** Due to Proposition 3.1, it is clear that $\psi$ yields a test at level $\alpha$. In the following, we compute the power of the test. Let $x \in \mathcal{X}$ be fixed. Under the alternate hypothesis $\mathbf{H_A}$, we have that $\xi \sim \mathcal{N}(0, \sigma^2 \mathbb{I}_d)$ and $y \sim p_{\theta + \xi}(\cdot \mid x)$. The power of the test is given by:

$$
\begin{aligned}
\mathrm{Pr}_{\mathbf{H_A}}(\psi(y, \xi \mid x) = 0) &= \mathrm{Pr}_{\mathbf{H_A}}(\langle \xi, \nabla \log p_\theta(y \mid x) \rangle \le \tau_\alpha \sigma \| \nabla \log p_\theta(y \mid x) \|) \\
&= \mathbb{E}_{\xi \sim \mathcal{N}(0, \sigma^2 \mathbb{I}_d)} \big[ \mathbb{E}_{y \sim p_{\theta + \xi}} [\mathbb{I}[\langle \xi, \nabla \log p_\theta(y \mid x) \rangle \le \tau_\alpha \sigma \| \nabla \log p_\theta(y \mid x) \|]] \big] \\
&= \mathbb{E}_{\xi \sim \mathcal{N}(0, \sigma^2 \mathbb{I}_d)} \left[ \mathbb{E}_{y \sim p_\theta} \left[ \frac{p_{\theta + \xi}(y \mid x)}{p_\theta(y \mid x)} \cdot \mathbb{I}[\langle \xi, \nabla \log p_\theta(y \mid x) \rangle \le \tau_\alpha \sigma \| \nabla \log p_\theta(y \mid x) \|] \right] \right],
\end{aligned}
$$

where in the last line, we changed the measure from $p_{\theta + \xi}$ to $p_\theta$ by paying for the density ratio $p_{\theta + \xi}(y|x)/p_\theta(y|x)$. Using the $\eta$-strong concavity of $\log p_\theta(y \mid x)$, for every $y$, we have:

$$
\log \left( \frac{p_{\theta + \xi}(y \mid x)}{p_\theta(y \mid x)} \right) \le \langle \xi, \nabla \log p_\theta(y \mid x) \rangle - \frac{\eta}{2} \| \xi \|^2,
$$

which implies that

$$
\begin{aligned}
\mathrm{Pr}_{\mathbf{H_A}}&(\psi(y, \xi \mid x) = 0) \\
&\le \mathbb{E}_{\xi \sim \mathcal{N}(0, \sigma^2 \mathbb{I}_d)} \left[ \mathbb{E}_{y \sim p_\theta} \left[ e^{\langle \xi, \nabla \log p_\theta(y|x) \rangle - \frac{\eta}{2} \| \xi \|^2} \cdot \mathbb{I}[\langle \xi, \nabla \log p_\theta(y \mid x) \rangle \le \tau_\alpha \sigma \| \nabla \log p_\theta(y \mid x) \|] \right] \right] \\
&\le \mathbb{E}_{y \sim p_\theta} \left[ e^{\tau_\alpha \sigma \| \nabla \log p_\theta(y|x) \|} \mathbb{E}_{\xi \sim \mathcal{N}(0, \sigma^2 \mathbb{I}_d)} \left[ \mathbb{I}[\langle \xi, \nabla \log p_\theta(y \mid x) \rangle \le \tau_\alpha \sigma \| \nabla \log p_\theta(y \mid x) \|] \cdot e^{-\frac{\eta}{2} \| \xi \|^2} \right] \right],
\end{aligned}
$$

where the second inequality follows by utilizing the constraints inside the indicator in the exponent, and by using the fact that $p_\theta$ is independent of $\xi$. Using the bound in Lemma I.3 for the expected value of Gaussian density function under the halfspace constraint, we get that for any $y$,

$$
\mathbb{E}_{\xi \sim \mathcal{N}(0, \sigma^2 \mathbb{I}_d)} \left[ \mathbb{I}[\langle \xi, \nabla \log p_\theta(y \mid x) \rangle \le \tau_\alpha \sigma \| \nabla \log p_\theta(y \mid x) \|] \cdot e^{-\frac{\eta}{2} \| \xi \|^2} \right] \le (1 + \eta \sigma^2)^{-d/2} \cdot \Phi \left( \tau_\alpha \cdot \sqrt{1 + \eta \sigma^2} \right),
$$

which implies that

$$
\begin{aligned}
\mathrm{Pr}_{\mathbf{H_A}}(\psi(y, \xi \mid x) = 0) &\le (1 + \eta \sigma^2)^{-d/2} \cdot \Phi \left( \tau_\alpha \cdot \sqrt{1 + \eta \sigma^2} \right) \mathbb{E}_\theta \left[ e^{\tau_\alpha \sigma \| \nabla \log p_\theta(y|x) \|} \right] \\
&\le (1 + \eta \sigma^2)^{-d/2} \mathbb{E}_\theta \left[ e^{\tau_\alpha \sigma \| \nabla \log p_\theta(y|x) \|} \right],
\end{aligned}
$$

where the last inequality follows since $\Phi(\cdot) \le 1$. The above bound implies that

$$
\mathrm{Pr}_{\mathbf{H_A}}(\psi(y, \xi \mid x) = 0) \le \beta,
$$

whenever

$$
d \ge \frac{2 \log \left( \mathbb{E}_\theta \left[ e^{\tau_\alpha \sigma \| \nabla \log p_\theta(y|x) \|} \right] \right) + 2 \log \left( \frac{1}{\beta} \right)}{(1 + \eta \sigma^2)}.
$$

$\square$

## I.4. Robustness to Corruptions of the Candidate Text

We have thus far demonstrated that GaussMark.Detect (Algorithm 2) enjoys high power under some mild assumptions (cf. Proposition 3.3 and I.8), but in practice, an adversarial user may modify the generated text to avoid detection of the watermark; for example, the user may first generate text $y$ from the watermarked model, and then modify it to $\widehat{y}$, by inserting, deleting or substituting tokens, or even by paraphrasing $y$. Thus, a practical watermarking scheme must be at least somewhat robust to corruption of the generated test. Indeed, as we show in the following theorem, a slight variant

of GaussMark.Generate and GaussMark.Detect, given in Algorithm 3 and Algorithm 4 respectively, enjoys theoretical robustness guarantees against a constant fraction of corruptions under the mild assumption that a token-level test on the uncorrupted tokens has sufficiently small level and high power. While we do not empirically investigate this alternative variant, we do observe that the original GaussMark enjoys some degree of robustness against corruptions in our experiments (cf. Appendix F).

Formally, we devise a robust detection test that takes in an input prompt $x'$, a watermarking key $\xi' = (\xi'_1, \ldots, \xi'_K) \in \mathbb{R}^{d \times K}$, and a candidate text $y' = (v'_1, \ldots, v'_K) \in \mathcal{V}^{\times K}$ consisting of $K$ tokens, and tests the following null and alternative hypothesis:

$\mathbf{H_0}$ : Given $x$, the key and the sample are independent, i.e., $(\xi', y') \sim \mathcal{N}(0, \sigma^2 \mathbb{I}_d)^{\otimes K} \otimes q$ with $q \in \Delta(\mathcal{Y})$.

$\mathbf{H_A}$ : The sample $y'$ is produced by Algorithm 3, using the key $\xi'$ and input prompt $x'$.

Additionally, in Algorithm 4, the notation $\text{Quantile}_{\lambda'}(\{\Gamma_1, \ldots, \Gamma_K\})$ is used to denote the maximal (in a sorted list) element $\Gamma' \in \{\Gamma_{(k)}\}_{k \leq K}$ that satisfies the relation $\frac{1}{K} \sum_{k=1}^{K} \mathbb{I}\{\Gamma_{(k)} \leq \Gamma'\} \leq \lambda'$, where $\lambda'$ is the quantile parameter. Then, the following result holds.

**Proposition I.9** (Robustness). *Let $\alpha_0, \beta_0 \in (0, 1)$ and $K \geq 1$ denote the length of the candidate text. Suppose that for every $k \in [K]$, input prompt $x' \in \mathcal{X}$, the prefix of tokens $v'_{<k} \in \mathcal{V}^{\times (k-1)}$, watermarking key $\xi_k \in \mathbb{R}^d$, and the candidate next token $v_k$, the token-level test given by $\mathbb{I}\left\{\frac{\langle \xi_k, \nabla \log p_\theta(v'_k | x' \circ v'_{<k}) \rangle}{\sigma \|\nabla \log p_\theta(v'_k | x' \circ x_{<k})\|} \geq \tau_{\alpha_0}\right\}$ has level $\alpha_0$ and power $1 - \beta_0$ in distinguishing between the following (token-level) null and alternative hypothesis:*

$\mathbf{H_0^k}$ : *The key $\xi'_k$ and the token $v'_k$ are independent, i.e., $(\xi'_k, v'_k) \sim \mathcal{N}(0, \sigma^2 \mathbb{I}_d) \otimes \mu_k$ with $\mu_k \in \Delta(\mathcal{V})$.*

$\mathbf{H_A^k}$ : *The token $v'_k$ is generated in (round $k$ of) Algorithm 3 using the input prompt $x'$, prefix of tokens $v'_{<k}$, and watermarking key $\xi'_k$.*

*Then, for any $\alpha, \beta, \lambda_0 \in (0, 1)$, input prompt $x$, candidate text $y = (v_1, \ldots, v_K)$, and quantile parameter $\lambda'$, the test in Algorithm 4 has:*

- *level $\alpha$, whenever $\lambda' \leq 1 - \alpha_0$ and $K \geq \frac{\log(1/\alpha)}{2(1-\lambda'-\alpha_0)^2}$,*

- *power $1 - \beta$, whenever $\lambda' \geq \lambda_0 + \beta_0$ and $K \geq \frac{2}{(1-\lambda_0)(\lambda'-\lambda_0-\beta_0)^2} \log\left(\frac{1}{\beta}\right)$,*

*while being robust under random independent substitutions affecting up to a $\lambda_0$-fraction of the tokens in $y$.*

We observe that Proposition I.9 requires the token-level test to have the respective guarantees on its power and level for any prompt $x'$ and prefix sequence $v'_{<k}$. This aligns with the style of guarantees that we provided for the vanilla test in GaussMark.Detect. In fact, the results of Proposition 3.1 and Proposition 3.3 already imply such a guarantee by setting $y = v'_k$ and $x = x' \circ v_{<k}$. Furthermore, while Proposition I.9 considers random substitutions of candidate text $y$, our analysis can be easily extended to robustness guarantees against random additions and deletions. The quantile-based test in Algorithm 4 can also boost the power of our test; notice that $\beta$ can be made arbitrarily smaller than $\beta_0$ by appropriately setting $K$. However, since we find that Algorithm 2 is already quite powerful in our experiments, we leave empirical validation of utilizing Algorithm 4 for future work.

---

**Algorithm 3** GaussMark.Generate: Robustly generating watermarked text

---

1: **Input:** Language Model $p_\theta : \mathcal{V}^* \to \Delta(\mathcal{V})$ with parameters $\theta \in \Theta$, prompt $x$, variance $\sigma^2 > 0$, generation length $K$.
2: Sample watermarking key $\xi = \{\xi_1, \ldots, \xi_K\}$ where $\xi_k \overset{\text{iid}}{\sim} \mathcal{N}(0, \sigma^2 \mathbf{I})$.
3: For $k = \{1, \ldots, K\}$, generate next token via $v_k \sim p_{\theta+\xi_k}(\cdot \mid x \circ v_{<k})$.
4: Return $y = (v_1, \ldots, v_K)$.

---

**Proof of Proposition I.9.** For $k \in [K]$, define the random variable $Z_k = \mathbb{I}\{\Gamma_k \geq \Phi^{-1}(1 - \alpha_0)\}$, where $\Gamma_k$ is defined in Algorithm 4.

---

**Algorithm 4** GaussMark.Detect: Robustly detecting watermarked text

---

1: **Input**: Language Model $p_\theta : \mathcal{V}^\star \to \Delta(\mathcal{Y})$ with parameters $\theta \in \Theta$, prompt $x$, candidate text $y = \{v_1, \ldots, v_K\}$, variance $\sigma^2 > 0$, watermark key $\xi = (\xi_1, \ldots, \xi_K)$, robustness parameter $\lambda_0 \in (0, 1)$, token-level test's level $\alpha_0 \in (0, 1)$, quantile parameter $\lambda'$.
2: For $k = \{1, \ldots, K\}$, evaluate the test statistic $\Gamma_k = \frac{\langle \xi_k, \nabla_\theta \log p_\theta(v_k | x \circ v_{1:k-1}) \rangle}{\sigma \| \nabla_\theta \log p_\theta(v_k | x \circ v_{1:k-1}) \|}$.
3: Define $\psi(y, \xi \mid x) := \text{Quantile}_{\lambda'}(\{\Gamma_1, \ldots, \Gamma_k\})$.
4: **if** $\psi(y, \xi \mid x) \geq \Phi^{-1}(1 - \alpha_0)$ **then** $\quad\quad\quad\quad$ # $\Phi$ is the CDF of standard Normal distribution
5: $\quad$ **Reject Null** and output "watermarked".
6: **else**
7: $\quad$ **Fail to reject Null**

---

**We first bound the level of the test in Algorithm 4.** Let $q \in \Delta(\mathcal{Y})$ be the distribution, independent of $\xi$, from which the sample $y$ is drawn under the null hypothesis $\mathbf{H_0}$. Since, the token-level test has level $\alpha_0$ for all $k \in [K]$, we have

$$\mathbb{E}_{\xi_k \sim \mathcal{N}(0, \sigma^2 \mathbb{I}_d), v_k \sim q}[Z_k \mid x \circ v_{<k}] \leq \alpha_0. \tag{8}$$

For the ease of notation, we use $\mathbb{E}_{k,0}[Z_k]$ to denote the expectation $\mathbb{E}_{\xi_k \sim \mathcal{N}(0, \sigma^2 \mathbb{I}_d), v_k \sim q}[Z_k \mid x \circ v_{<k}]$ for $k \in [K]$. The level of the sequence level test is given by:

$$\Pr_{\mathbf{H_0}}(\psi(y, \xi \mid x) \geq \Phi^{-1}(1 - \alpha_0)) = \Pr_{\mathbf{H_0}}\left(\text{Quantile}_{\lambda'}\{\Gamma_1, \ldots, \Gamma_k\} \geq \Phi^{-1}(1 - \alpha_0)\right)$$

$$= \Pr_{\mathbf{H_0}}\left(\sum_{k \in [K]} Z_k \geq K(1 - \lambda')\right)$$

$$= \Pr_{\mathbf{H_0}}\left(\sum_{k \in \mathcal{K}} Z_k - \mathbb{E}_{k,0}[Z_k] \geq K(1 - \lambda') - \sum_{k \in \mathcal{K}} \mathbb{E}_{k,0}[Z_k]\right)$$

$$\leq \Pr_{\mathbf{H_0}}\left(\frac{1}{K}\left(\sum_{k \in \mathcal{K}} Z_k - \mathbb{E}_{k,0}[Z_k]\right) \geq 1 - \lambda' - \alpha_0\right),$$

where the equality in the second line follows from the definition of Quantile, and the inequality in the last line is due to (8). Next, note that under the null hypothesis $\mathbf{H_0}$, $\xi = \{\xi_1, \ldots, \xi_K\} \sim \mathcal{N}(0, \mathbb{I}_d)^{\otimes K}$, and $y \sim q$ where $q$ is independent of $\xi$. Additionally, because the substitutions are done randomly, and independently of $y$, the random variables $\{Z_1, \ldots, Z_k\}$ are independent (under the null hypothesis). Thus, using Hoeffding's inequality, we get

$$\Pr_{\mathbf{H_0}}(\psi(y, \xi \mid x) \geq \Phi^{-1}(1 - \alpha_0)) \leq e^{-2K(1 - \lambda' - \alpha_0)^2},$$

whenever $\lambda' \leq 1 - \alpha_0$. The above implies that the test is at level $\alpha$ for $K \geq \frac{1}{2(1 - \lambda' - \alpha_0)^2} \log\left(\frac{1}{\alpha}\right)$.

**We next bound the power of the test in Algorithm 4.** Let $\mathcal{K}$ denote the set of those indices $k \in [K]$ for which the block $v_k$ is not corrupted. Thus, $|\mathcal{K}| \geq K(1 - \lambda_0)$. Since, the token-level test has power $1 - \beta_0$ for any uncorrupted token, we have

$$\mathbb{E}_{\substack{\xi_k \sim \mathcal{N}(0, \sigma^2 \mathbb{I}_d), \\ v_k \sim p_{\theta + \xi_k}(\cdot | x \circ v_{<k})}}[Z_k \mid x \circ v_{<k}] \geq 1 - \beta_0. \tag{9}$$

for any $k \in \mathcal{K}$. We note that

$$\Pr_{\mathbf{H_A}}(\psi(y, \xi \mid x) < \Phi^{-1}(1 - \alpha_0)) = \Pr_{\mathbf{H_A}}\left(\text{Quantile}_{\lambda'}\{\Gamma_1, \ldots, \Gamma_k\} < \Phi^{-1}(1 - \alpha_0)\right)$$

$$= \Pr_{\mathbf{H_A}}\left(\sum_{k \in [K]} Z_k < K(1 - \lambda')\right)$$

$$\leq \Pr_{\mathbf{H_A}}\left(\sum_{k \in \mathcal{K}} Z_k < K(1 - \lambda')\right),$$

where the second line follows by plugging in the definition of Quantile, and the inequality in the third line follows by ignoring the corrupted tokens. Next, for $k \in [K]$, let $\mathbb{E}_{k,A}[Z_k]$ denote the expectation $\mathbb{E}_{\xi_k \sim \mathcal{N}(0,\sigma^2 \mathbb{I}_d), v_k \sim p_{\theta + \xi_k}(\cdot | x \circ v_{<k})}[Z_k \mid x \circ v_{<k}]$. Adding and subtracting $\mathbb{E}_{k,A}[Z_k]$ on both sides, we get

$$\mathrm{Pr}_{\mathbf{H_A}}(\psi(y, \xi \mid x) < \Phi^{-1}(1 - \alpha_0)) \le \mathrm{Pr}_{\mathbf{H_A}}\left(\sum_{k \in \mathcal{K}} Z_k - \mathbb{E}_{k,A}[Z_k] < K(1 - \lambda') - \sum_{k \in \mathcal{K}} \mathbb{E}_{k,A}[Z_k]\right)$$

$$\le \mathrm{Pr}_{\mathbf{H_A}}\left(\frac{1}{|\mathcal{K}|}\left(\sum_{k \in \mathcal{K}} Z_k - \mathbb{E}_{k,A}[Z_k]\right) < K(1 - \lambda') - |\mathcal{K}|(1 - \beta_0)\right)$$

$$\le \mathrm{Pr}_{\mathbf{H_A}}\left(\frac{1}{|\mathcal{K}|}\left(\sum_{k \in \mathcal{K}} Z_k - \mathbb{E}_{k,A}[Z_k]\right) < -\lambda' + \lambda_0 + \beta_0\right)$$

where the second line follows from (9), and the third line uses the fact that $K(1 - \lambda_0) \le |\mathcal{K}| \le K$. Next, notice the fact that under $\mathbf{H_A}$, the tokens $v_1, \ldots, v_K$ are generated following the autoregressive process in Algorithm 3, and thus, the sequence, $\{Z_k - \mathbb{E}_{k,A}[Z_k]\}_{k \in \mathcal{K}}$ forms a martingale difference sequence. Using Azuma-Hoeffding's inequality, we get

$$\mathrm{Pr}_{\mathbf{H_A}}(\psi(y, \xi \mid x) < \Phi^{-1}(1 - \alpha_0)) \le e^{-|\mathcal{K}|(\lambda' - \lambda_0 - \beta_0)^2/2}.$$

whenever $\lambda' \ge \lambda_0 + \beta_0$. The above implies that the test has power $1 - \beta$ whenever $|\mathcal{K}| \ge \frac{2}{(\lambda' - \lambda_0 - \beta_0)^2} \log\left(\frac{1}{\beta}\right)$, which happens when, $K \ge \frac{2}{(1 - \lambda_0)(\lambda' - \lambda_0 - \beta_0)^2} \log\left(\frac{1}{\beta}\right)$.

$\square$

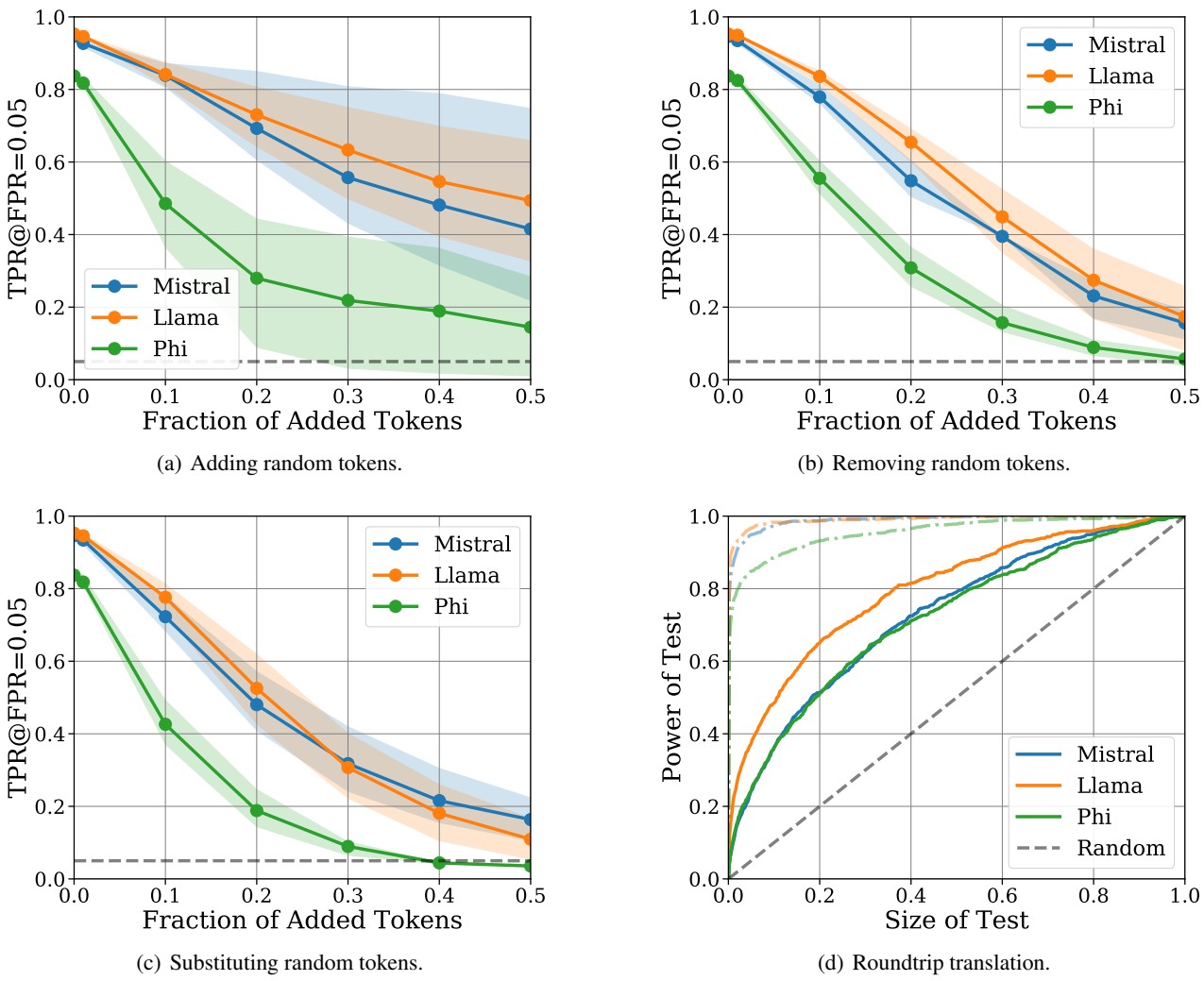

(a) Adding random tokens.

(b) Removing random tokens.

(c) Substituting random tokens.

(d) Roundtrip translation.

Figure 13. Demonstration of robustness of GaussMark to four kinds of corruptions. We demonstrate the effects of (a) random insertions of tokens, (b) random deletions of tokens, and (c) random substitutions of tokens on the rate of detection of true positives of watermarked text averaged over 3 seeds. We also consider the effect that (d) roundtrip translation (through French) has on the ROC curve and include the ROC curves of the watermarked model on uncorrupted data for reference. Note that GaussMark is relatively robust to token-level corruptions and retains nontrivial power even after the more challenging roundtrip translation attack.

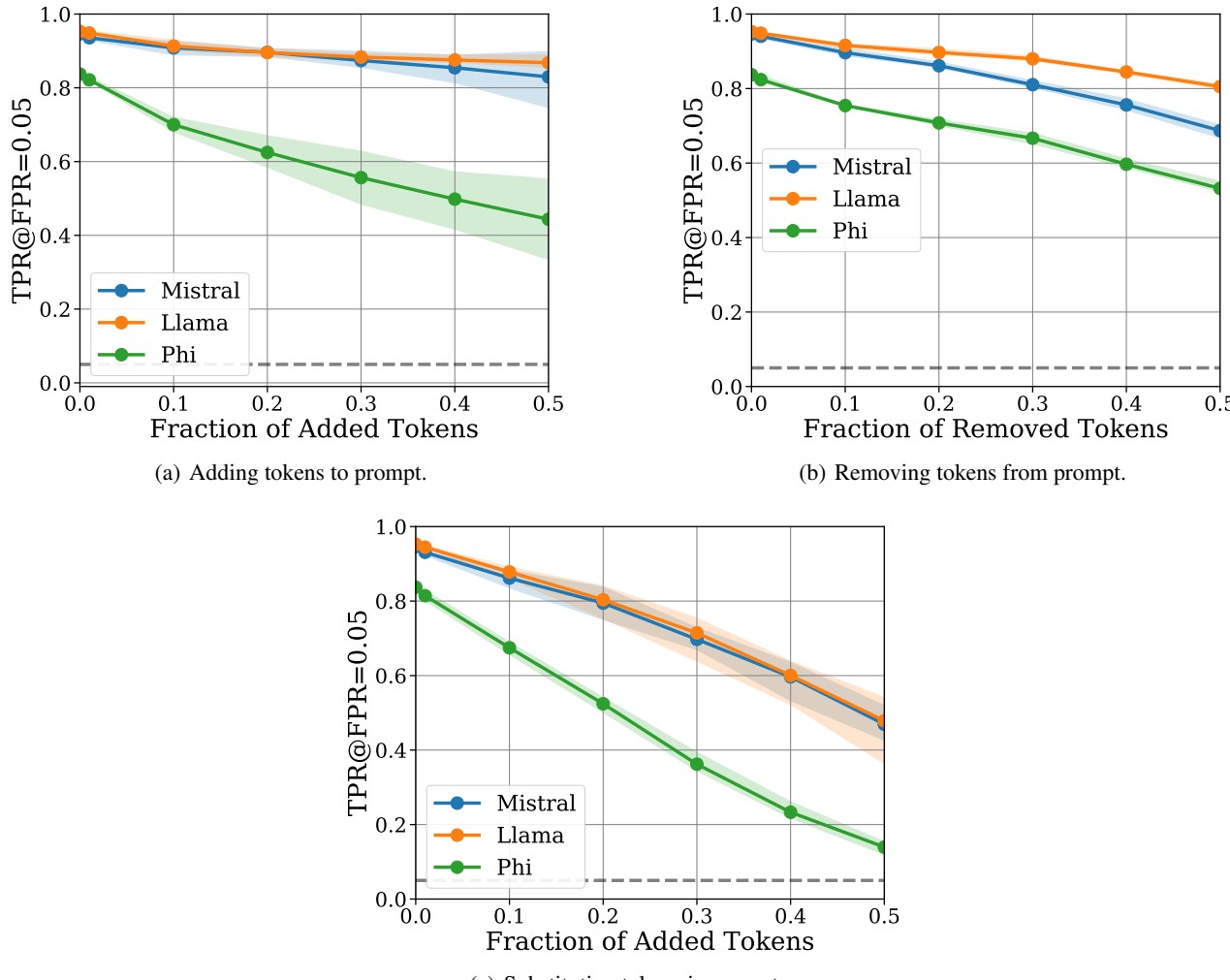

(a) Adding tokens to prompt.

(b) Removing tokens from prompt.

(c) Substituting tokens in prompt.

*Figure 14.* Demonstration of robustness of GaussMark to ignorance of prompt. We demonstrate the effects of (a) inserting, (b) deleting, and (c) substituting parts of the prompt on the fraction of detected sequences at the $p = 0.05$ level. Note that GaussMark is robust to these corruptions, as the p-values remain relatively stable across different prompt corruptions, demonstrating that knowing the prompt is not necessary for watermark detection.

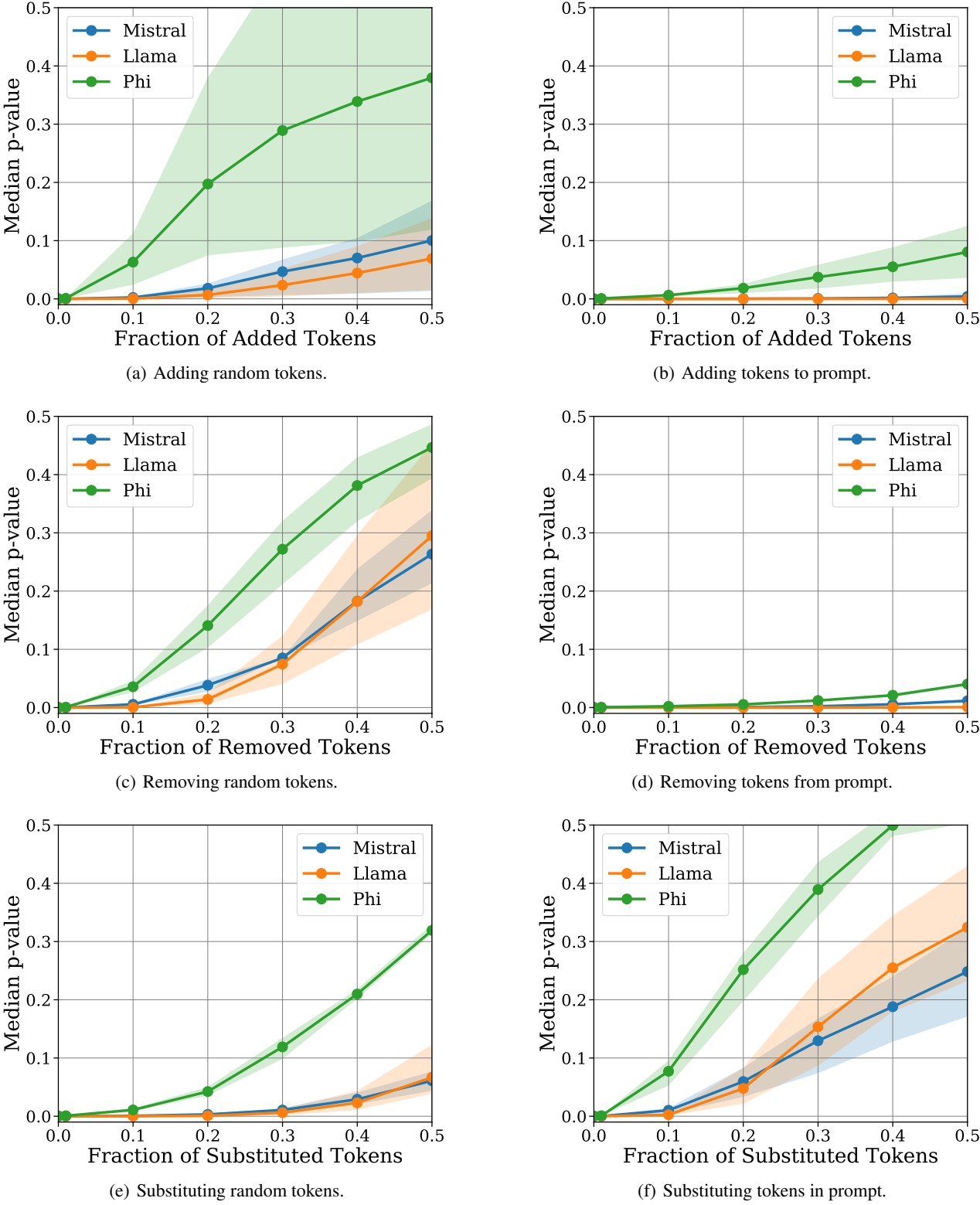

(a) Adding random tokens.

(b) Adding tokens to prompt.

(c) Removing random tokens.

(d) Removing tokens from prompt.

(e) Substituting random tokens.

(f) Substituting tokens in prompt.

*Figure 15.* Effect of token-level corruptions on the median p-values of GaussMark for each of the three models.

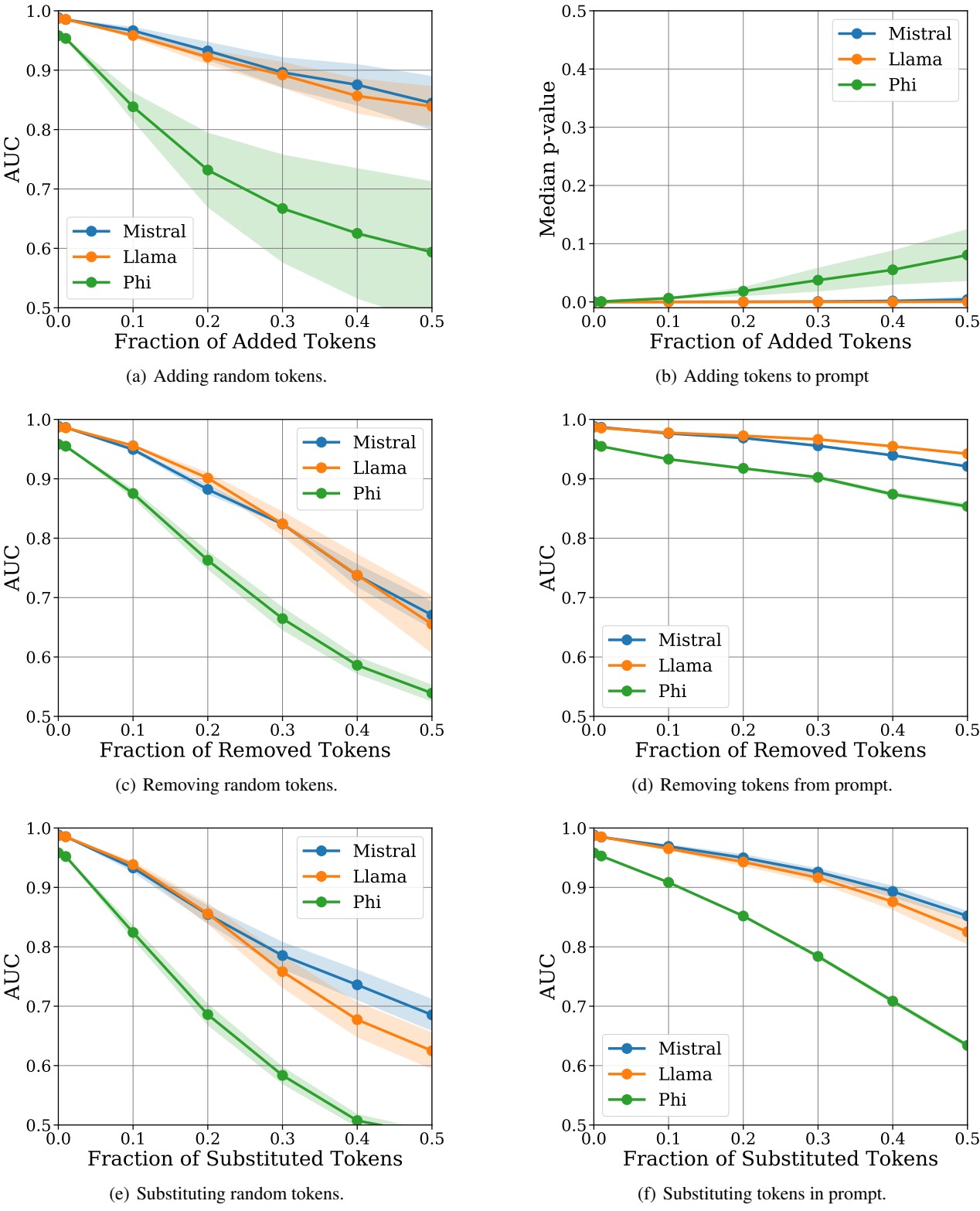

(a) Adding random tokens.

(b) Adding tokens to prompt

(c) Removing random tokens.

(d) Removing tokens from prompt.

(e) Substituting random tokens.

(f) Substituting tokens in prompt.

*Figure 16.* Effect of token-level corruptions on the AUC of GaussMark for each of the three models.

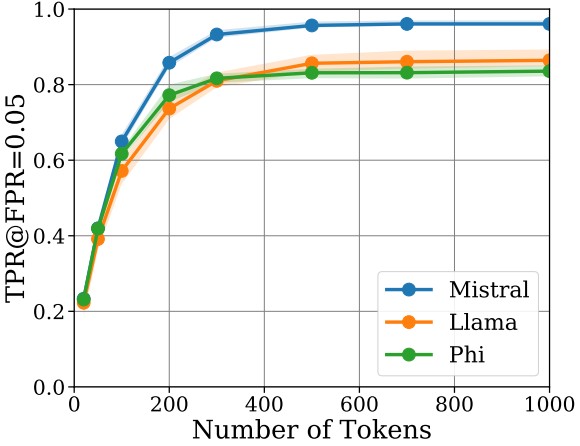

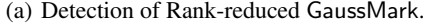

(a) Detection of Rank-reduced GaussMark.

| Var | 1e-05 | 3e-05 | 1e-04 | 3e-04 | 1e-03 |
|---|---|---|---|---|---|
| Rank 1024 | 0.8506 | 0.8491 | 0.8499 | 0.8461 | 0.7968 |
| Rank 512 | 0.8461 | 0.8514 | 0.8469 | 0.8052 | 0.7013 |
| Rank 256 | 0.8552 | 0.8469 | 0.8431 | 0.7945 | 0.6808 |
| Rank 128 | 0.8491 | 0.8446 | 0.8453 | 0.7892 | 0.6899 |
| Rank 64 | 0.8560 | 0.8431 | 0.8423 | 0.7794 | 0.6626 |
| Rank 0 | 0.8499 | 0.8469 | 0.8332 | 0.7695 | 0.6558 |

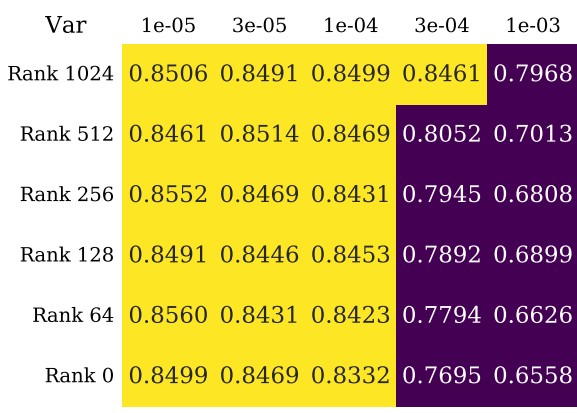

(b) Phi3.5–Mini performance on GSM–8K (`Layer 31`, `down_proj`).

*Figure 17.* Demonstration of effect of of rank-reduced GaussMark. (a) The fraction of detected watermarked responses at $p = 0.05$ for the rank-reduced version of GaussMark averaged over 3 seeds for each model. (b) The effect of rank-reduction on GaussMark.Generate as measured by the performance of Phi3.5–Mini on GSM–8K. Each curve reflects a different number of the top principal components preserved by GaussMark, with the $x$-axis being the variance of added noise. As more PCs are removed, Phi3.5–Mini can handle greater variance without a corresponding reduction in performance.

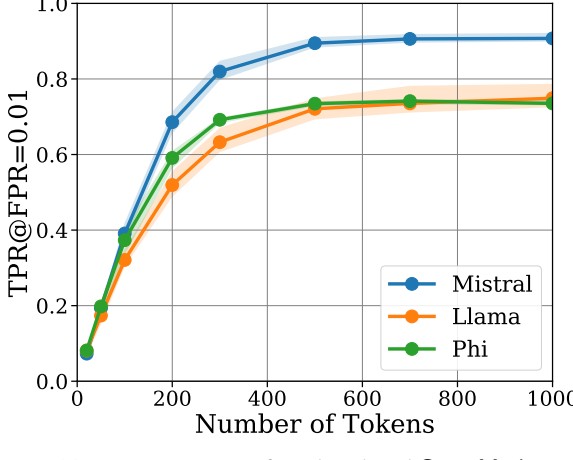

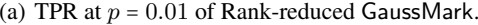

(a) TPR at $p = 0.01$ of Rank-reduced GaussMark.

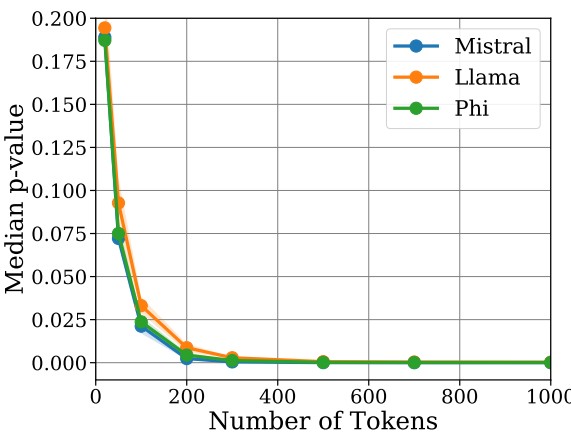

(b) Median $p$-values of Rank-reduced GaussMark.

*Figure 18.* Detectability of rank-reduced GaussMark on all three models as measured by (a) TPR at FPR 0.01 and (b) median p-values as a function of the number of watermarked tokens. The watermark is detectable for a substantial fraction of the text in all three models, with more tokens leading to increased detectability.

| Var | 1e-05 | 3e-05 | 1e-04 | 3e-04 | 1e-03 |
|---|---|---|---|---|---|
| Rank 1024 | 0.2091 | 0.0564 | 0.0023 | 0.0000 | 0.0000 |
| Rank 512 | 0.1814 | 0.0462 | 0.0009 | 0.0000 | 0.0000 |
| Rank 256 | 0.1636 | 0.0386 | 0.0006 | 0.0000 | 0.0000 |
| Rank 128 | 0.1679 | 0.0399 | 0.0005 | 0.0000 | 0.0000 |
| Rank 64 | 0.1510 | 0.0369 | 0.0005 | 0.0000 | 0.0000 |
| Rank 0 | 0.1403 | 0.0292 | 0.0003 | 0.0000 | 0.0000 |

(a) Median $p$-values of Llama3.1–8B (Layer 28, down_proj).

| Var | 1e-05 | 3e-05 | 1e-04 | 3e-04 | 1e-03 |
|---|---|---|---|---|---|
| Rank 1024 | 0.0008 | 0.0000 | 0.0000 | 0.0001 | 0.0015 |
| Rank 512 | 0.0007 | 0.0000 | 0.0000 | 0.0002 | 0.0021 |
| Rank 256 | 0.0007 | 0.0000 | 0.0000 | 0.0001 | 0.0014 |
| Rank 128 | 0.0009 | 0.0000 | 0.0000 | 0.0001 | 0.0014 |
| Rank 64 | 0.0009 | 0.0000 | 0.0000 | 0.0001 | 0.0016 |
| Rank 0 | 0.0008 | 0.0000 | 0.0000 | 0.0000 | 0.0010 |

(b) Median $p$-values of Mistral–7B (Layer 28, gate_proj).

| Var | 1e-05 | 3e-05 | 1e-04 | 3e-04 | 1e-03 |
|---|---|---|---|---|---|
| Rank 1024 | 0.5321 | 0.4436 | 0.2835 | 0.1184 | 0.0072 |
| Rank 512 | 0.4770 | 0.3798 | 0.2379 | 0.0792 | 0.0021 |
| Rank 256 | 0.4789 | 0.3683 | 0.2353 | 0.0668 | 0.0015 |
| Rank 128 | 0.4490 | 0.3918 | 0.2083 | 0.0591 | 0.0013 |
| Rank 64 | 0.4590 | 0.3662 | 0.2243 | 0.0547 | 0.0008 |
| Rank 0 | 0.4543 | 0.3561 | 0.2060 | 0.0567 | 0.0008 |

(c) Median $p$-values of Phi3.5–Mini (Layer 28, down_proj).

*Figure 19.* Effect of rank reduction on the median p-values of GaussMark for (a) Llama3.1–8B, (b) Mistral–7B, and (c) Phi3.5–Mini. As the rank of the quotient space increases, the detectability of the watermark decreases, as expected.

| Var | 1e-05 | 3e-05 | 1e-04 | 3e-04 | 1e-03 |
|---|---|---|---|---|---|
| Rank 1024 | 0.5580 | 0.5019 | 0.4230 | 0.2449 | 0.0000 |
| Rank 512 | 0.5375 | 0.4602 | 0.3867 | 0.1395 | 0.0000 |
| Rank 256 | 0.5171 | 0.4473 | 0.3768 | 0.1137 | 0.0000 |
| Rank 128 | 0.5171 | 0.4625 | 0.3844 | 0.1198 | 0.0000 |
| Rank 64 | 0.5277 | 0.4503 | 0.3836 | 0.1130 | 0.0000 |
| Rank 0 | 0.5269 | 0.4587 | 0.3707 | 0.1205 | 0.0000 |

(a) GSM–8K performance of Llama3.1–8B (`Layer 31, down_proj`).

| Var | 1e-05 | 3e-05 | 1e-04 | 3e-04 | 1e-03 |
|---|---|---|---|---|---|
| Rank 1024 | 0.4253 | 0.4056 | 0.3692 | 0.0303 | 0.0000 |
| Rank 512 | 0.4185 | 0.4003 | 0.3412 | 0.0152 | 0.0000 |
| Rank 256 | 0.4011 | 0.3867 | 0.2866 | 0.0023 | 0.0000 |
| Rank 128 | 0.3897 | 0.3609 | 0.2760 | 0.0023 | 0.0000 |
| Rank 64 | 0.3965 | 0.3723 | 0.2820 | 0.0015 | 0.0000 |
| Rank 0 | 0.3707 | 0.3306 | 0.2183 | 0.0000 | 0.0000 |

(b) GSM–8K performance of Mistral–7B (`Layer 31, down_proj`).

| Var | 1e-05 | 3e-05 | 1e-04 | 3e-04 | 1e-03 |
|---|---|---|---|---|---|
| Rank 1024 | 0.8506 | 0.8491 | 0.8499 | 0.8461 | 0.7968 |
| Rank 512 | 0.8461 | 0.8514 | 0.8469 | 0.8052 | 0.7013 |
| Rank 256 | 0.8552 | 0.8469 | 0.8431 | 0.7945 | 0.6808 |
| Rank 128 | 0.8491 | 0.8446 | 0.8453 | 0.7892 | 0.6899 |
| Rank 64 | 0.8560 | 0.8431 | 0.8423 | 0.7794 | 0.6626 |
| Rank 0 | 0.8499 | 0.8469 | 0.8332 | 0.7695 | 0.6558 |

(c) GSM–8K performance of Phi3.5–Mini (`Layer 31, down_proj`).

*Figure 20.* Effect of rank reduction in GaussMark on model performance on GSM–8K on (a) Llama3.1–8B, (b) Mistral–7B, and (c) Phi3.5–Mini. As the rank of the quotient space increases, the model quality increases as well, reflecting the decreased influence of the perturbation on the model outputs.

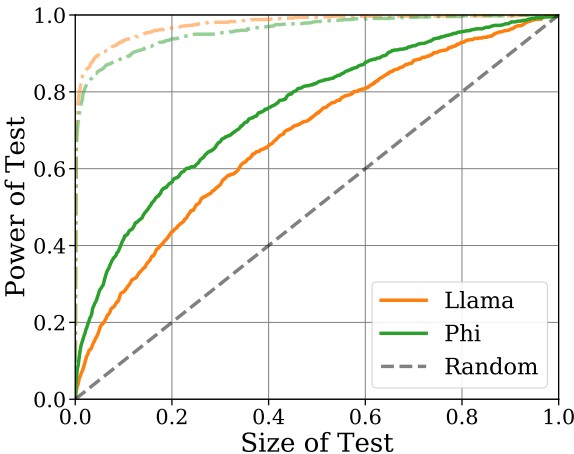

(a) Effect of roundtrip translation through French on ROC curves of rank-reduced GaussMark.

*Figure 21.* Effect of roundtrip translation on rank-reduced instantiation of GaussMark.

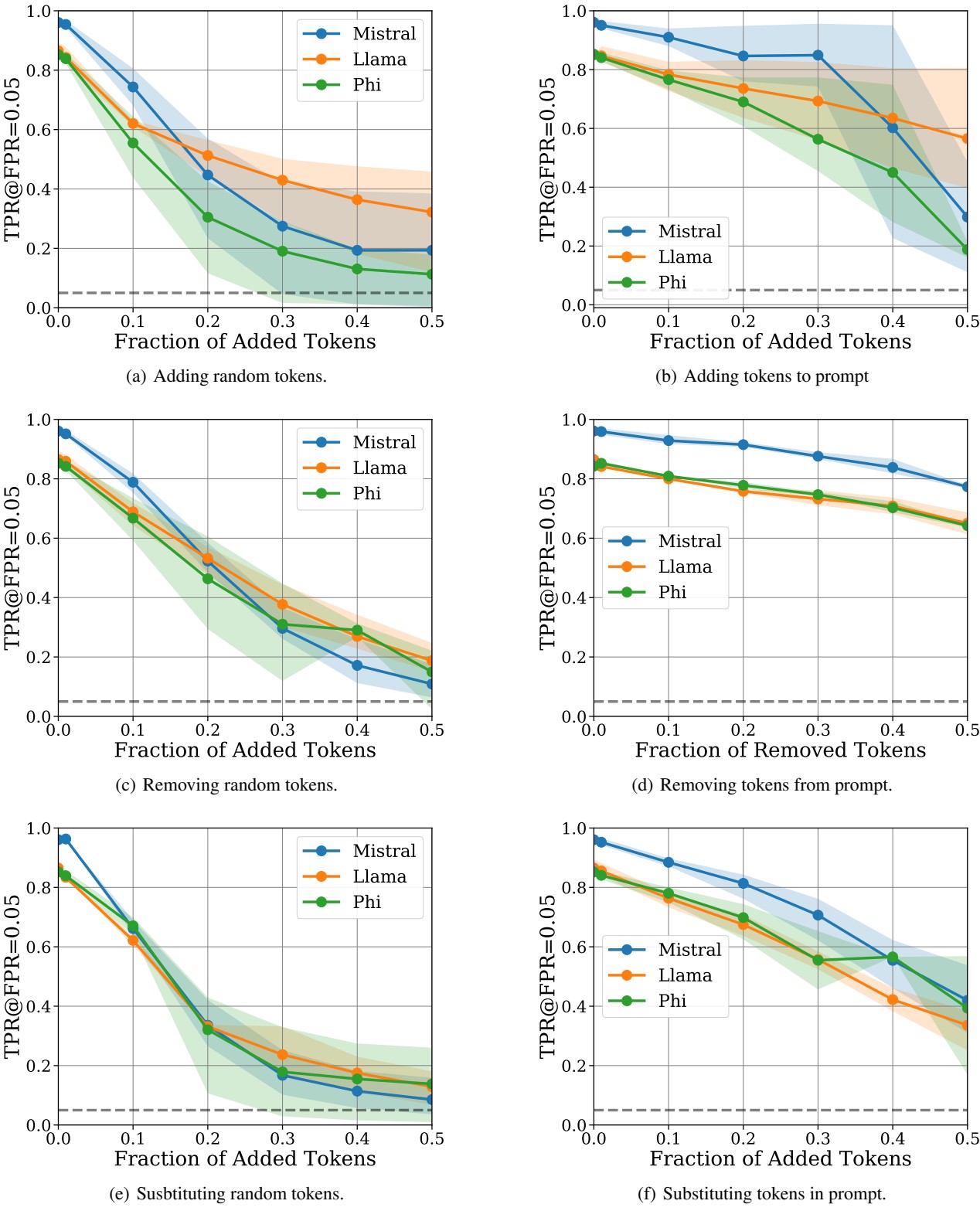

*Figure 22.* Effect of token-level corruptions on Rank-reduced GaussMark as measured by TPR at FPR $p = 0.05$.

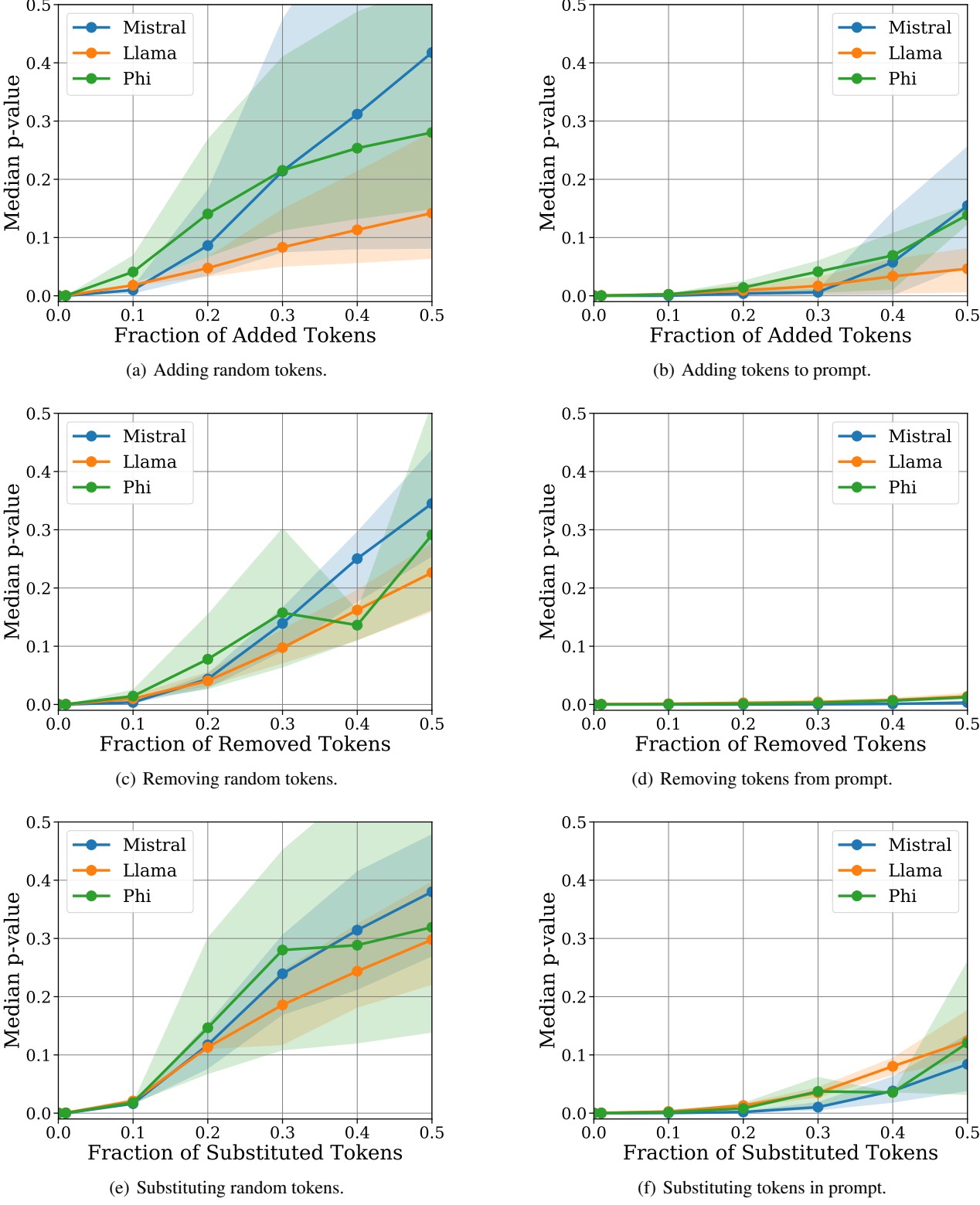

*Figure 23.* Effect of token-level corruptions on rank-reduced GaussMark as measured by median p-value.

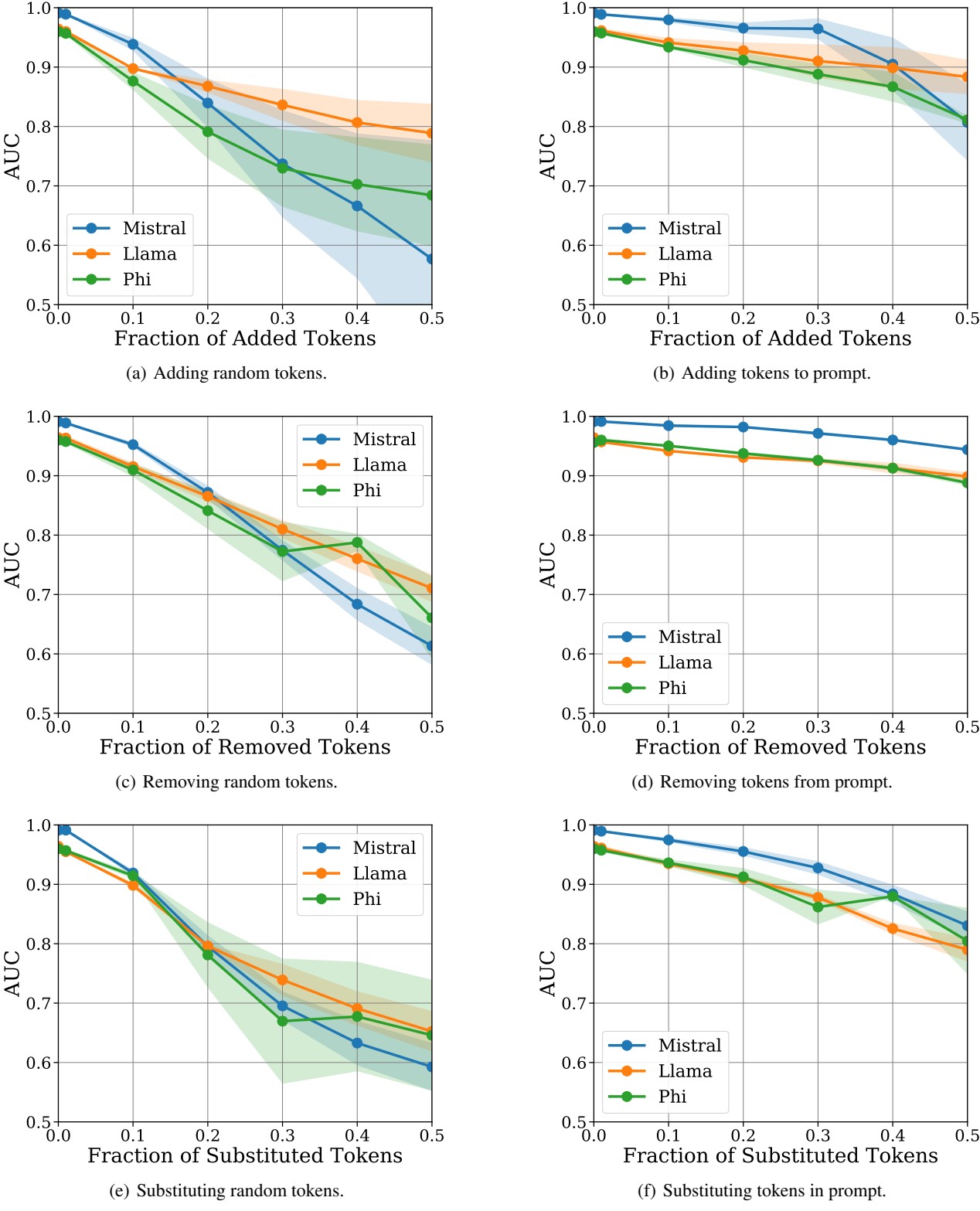

(a) Adding random tokens.

(b) Adding tokens to prompt.

(c) Removing random tokens.

(d) Removing tokens from prompt.

(e) Substituting random tokens.

(f) Substituting tokens in prompt.

*Figure 24.* Effect of token-level corruptions on rank-reduced GaussMark as measured by AUC.

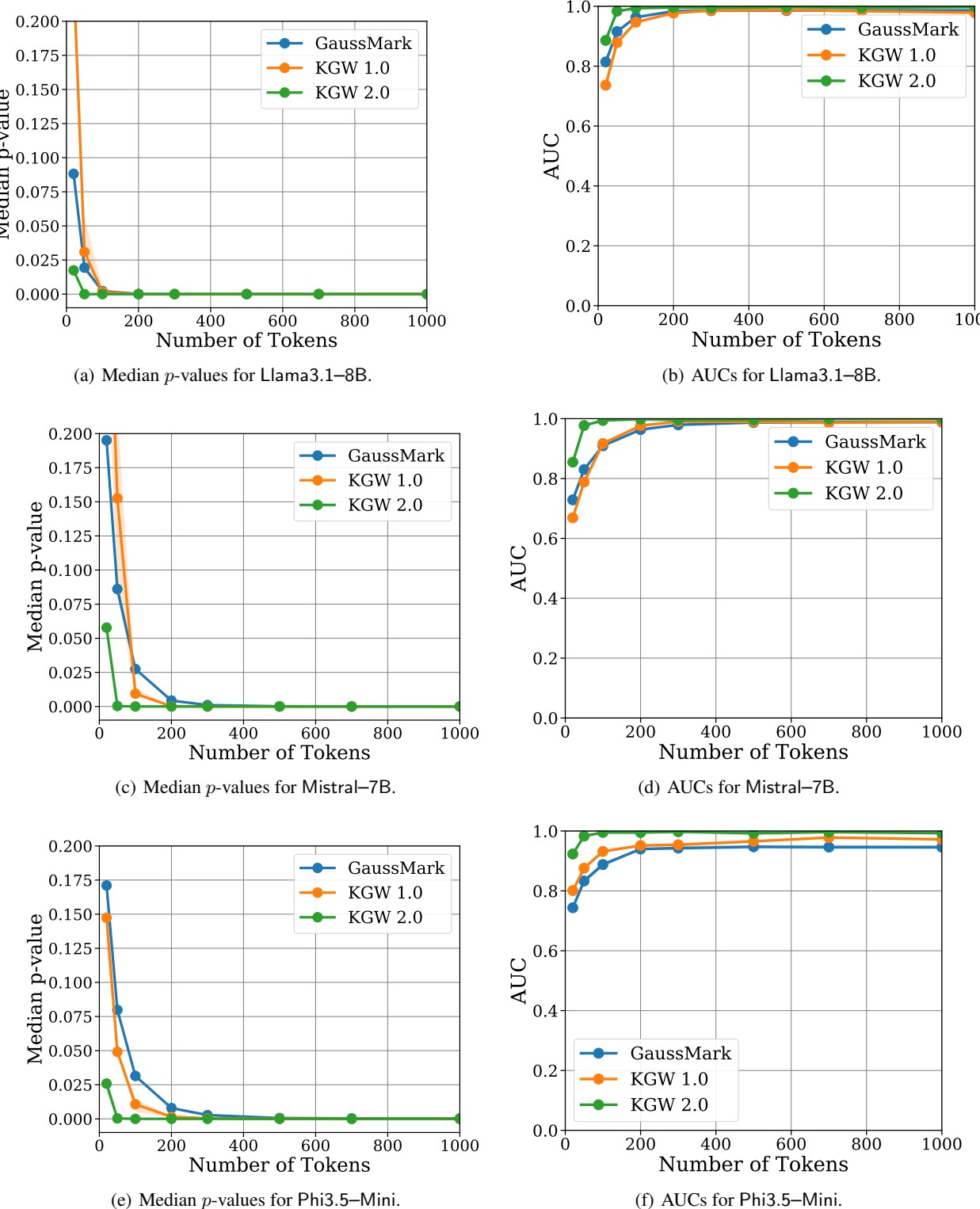

(a) Median $p$-values for Llama3.1–8B.

(b) AUCs for Llama3.1–8B.

(c) Median $p$-values for Mistral–7B.

(d) AUCs for Mistral–7B.

(e) Median $p$-values for Phi3.5–Mini.

(f) AUCs for Phi3.5–Mini.

*Figure 25.* Comparison of detectability of GaussMark with KGW-1 and KGW-2 as measured by median detection times and AUCs for Llama3.1–8B (a-b), Mistral–7B (c-d), and Phi3.5–Mini (e-f).

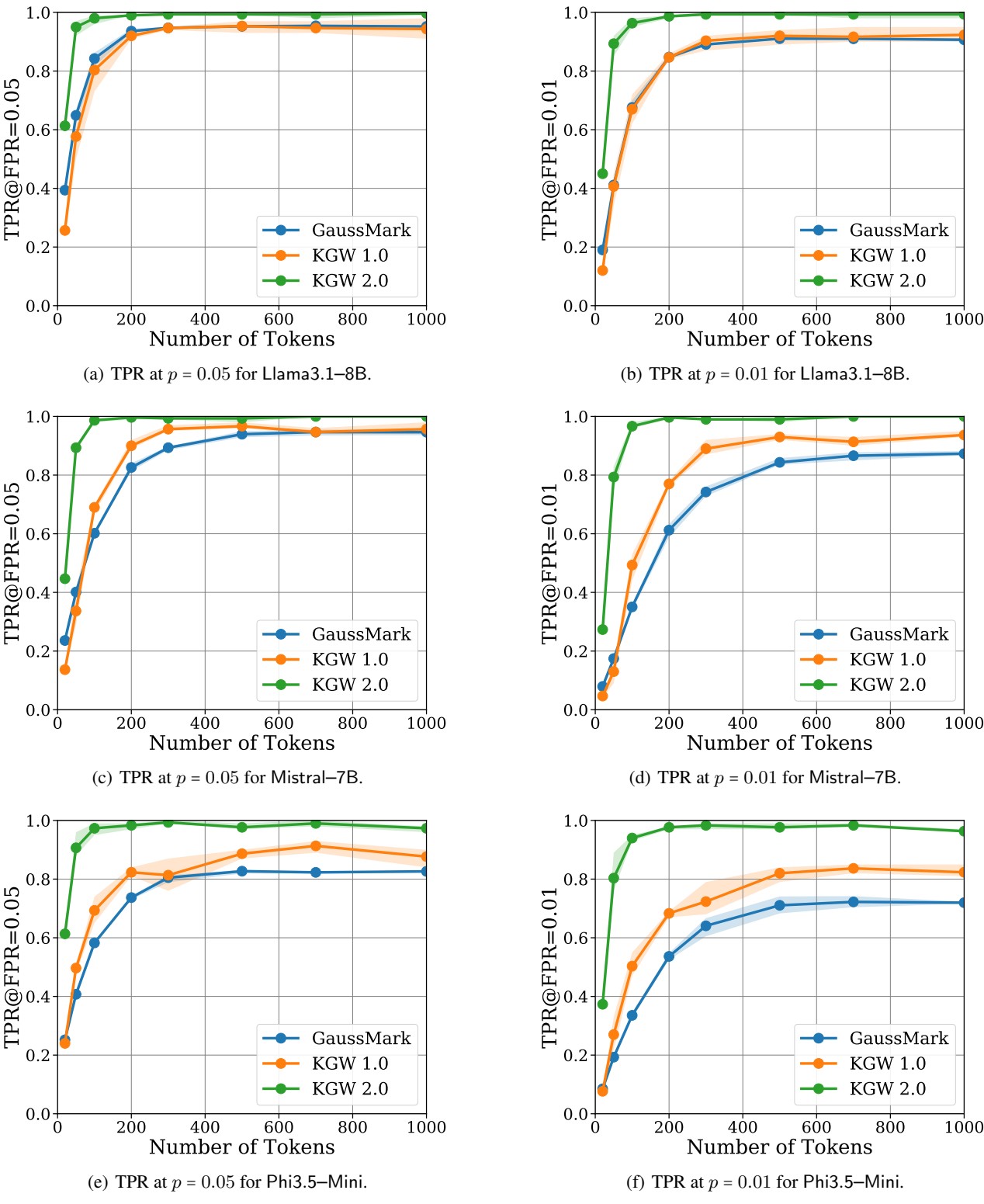

(a) TPR at $p = 0.05$ for Llama3.1–8B.

(b) TPR at $p = 0.01$ for Llama3.1–8B.

(c) TPR at $p = 0.05$ for Mistral–7B.

(d) TPR at $p = 0.01$ for Mistral–7B.

(e) TPR at $p = 0.05$ for Phi3.5–Mini.

(f) TPR at $p = 0.01$ for Phi3.5–Mini.

*Figure 26.* Comparison of detectability of GaussMark with KGW-1 and KGW-2 as measured by the TPR@FPR 0.05 and 0.01 for Llama3.1–8B (a-b), Mistral–7B (c-d), and Phi3.5–Mini (e-f).

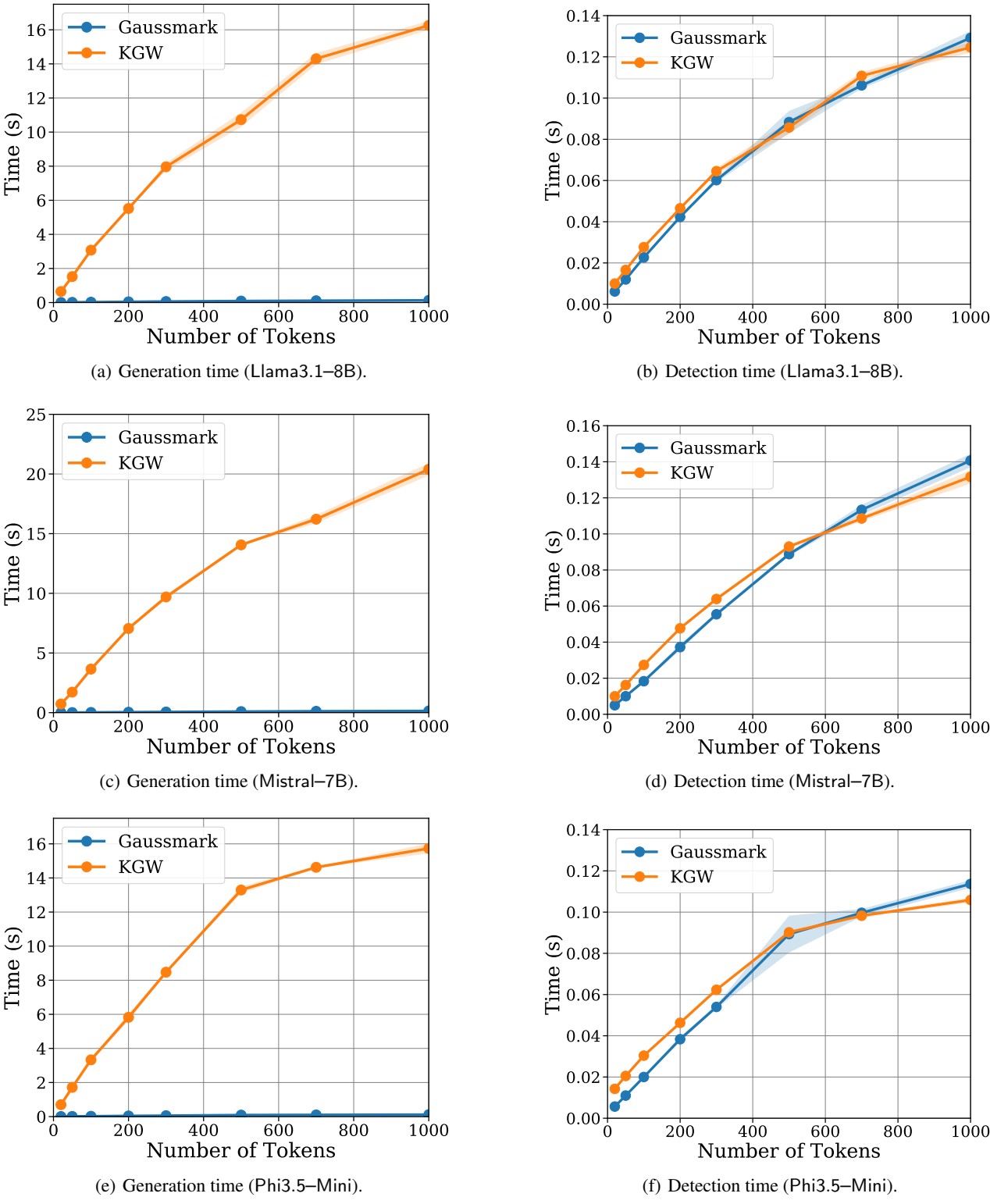

(a) Generation time (Llama3.1–8B).

(b) Detection time (Llama3.1–8B).

(c) Generation time (Mistral–7B).

(d) Detection time (Mistral–7B).

(e) Generation time (Phi3.5–Mini).

(f) Detection time (Phi3.5–Mini).

*Figure 27.* Comparison between GaussMark and KGW generation and detection times averaged across 3 seeds and 100 generations for Llama3.1–8B (a-b), Mistral–7B (c-d), and Phi3.5–Mini (e-f).

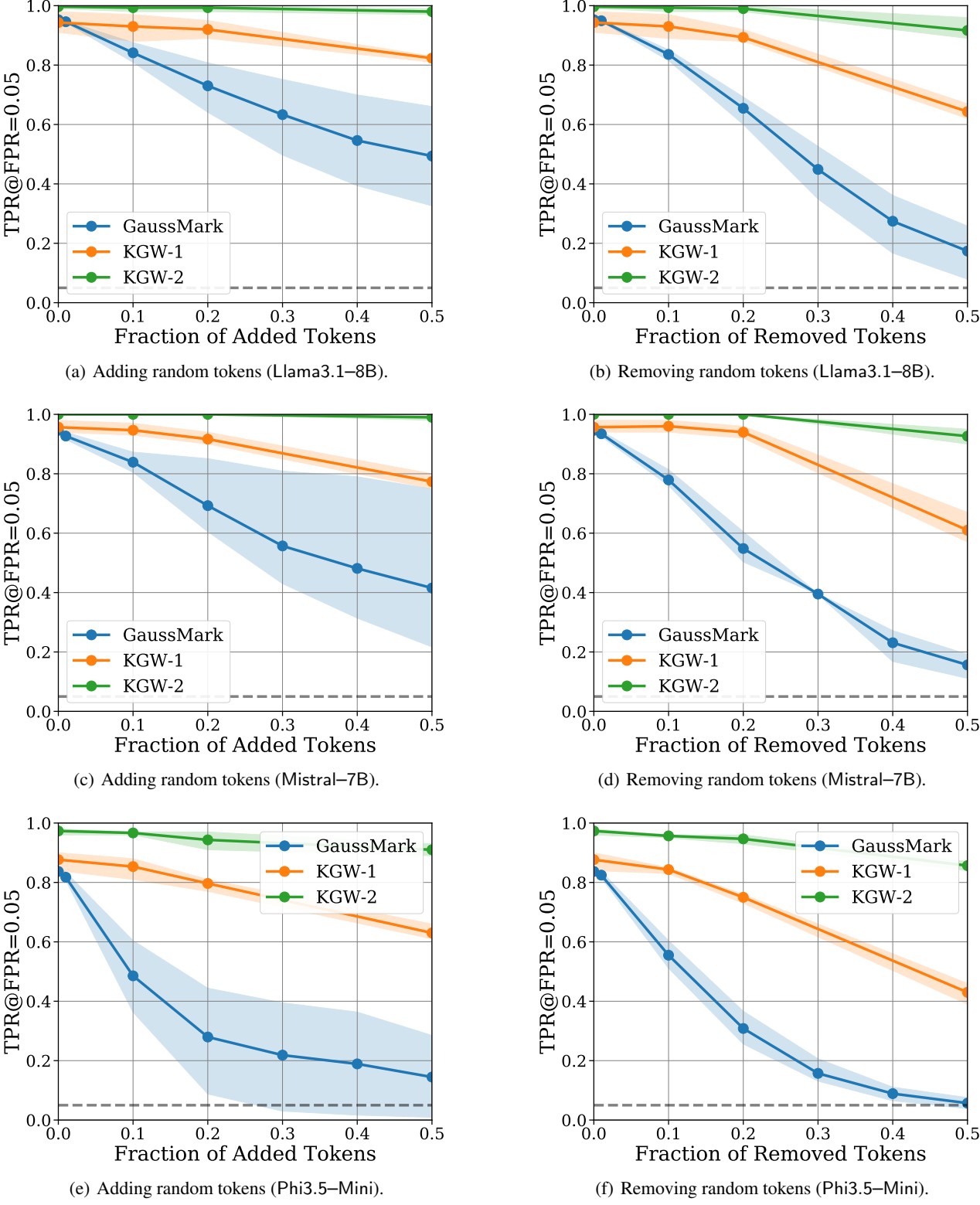

(a) Adding random tokens (Llama3.1–8B).

(b) Removing random tokens (Llama3.1–8B).

(c) Adding random tokens (Mistral–7B).

(d) Removing random tokens (Mistral–7B).

(e) Adding random tokens (Phi3.5–Mini).

(f) Removing random tokens (Phi3.5–Mini).

*Figure 28.* Comparison between GaussMark and KGW robustness to token level corruptions (adding and removing at random points) as measured by detection at FPR = 0.05 for Llama3.1–8B (a-b), Mistral–7B (c-d), and Phi3.5–Mini (e-f).

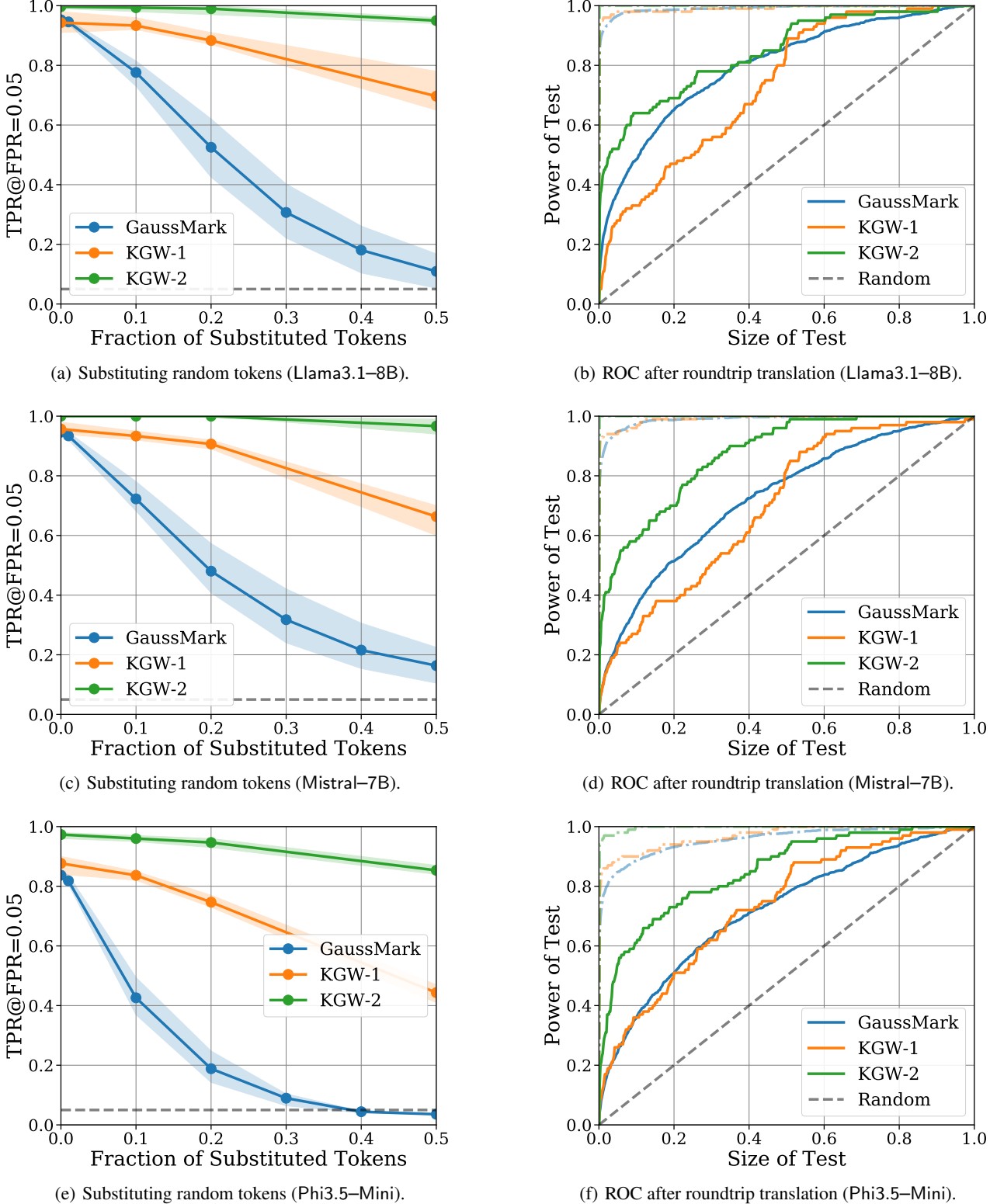

(a) Substituting random tokens (Llama3.1−8B).

(b) ROC after roundtrip translation (Llama3.1−8B).

(c) Substituting random tokens (Mistral−7B).

(d) ROC after roundtrip translation (Mistral−7B).

(e) Substituting random tokens (Phi3.5−Mini).

(f) ROC after roundtrip translation (Phi3.5−Mini).

*Figure 29.* Comparison between GaussMark and KGW robustness to token level corruptions (substituting tokens) and roundtrip translation through French for Llama3.1−8B (a-b), Mistral−7B (c-d), and Phi3.5−Mini (e-f).

## J. Sample Generated Texts

In this section, we provide samples of text completions via Mistral–7B, Llama3.1–8B and Phi3.5–Mini models. As above, the prompts are from the `realnewslike` split of the C4 dataset. For the Phi3.5–Mini model, the prompts were provided along with the following instructions prior to each prompt:

```
<|system|>
    You are a helpful assistant. <|end|>
<|user|>
    Complete the following prompt: <|end|>
<|assistant|>
```

In the examples below, the text under the heading "Base Model" represents a random text completion sampled from the original (unperturbed) model. The text under the heading "Watermarked Model" shows a random text completion produced by the watermarked model, where the watermark parameters are as described in Table 2.

### J.1. Example 1

> **Input Prompt**:
>
> `Sci-Fi fans will rejoice with today's news as details about the upcoming game, Doctor Who:  The Etern`

---

**Mistral–7B Generations**

**Base Model:** ity Clock, have been released. The game is the first cross-platform adventure game in the series' history and was co-created by TimeGate Studios and gaming legends, Doctor Who writers Steven Moffat and Steven Thompson.

Doctor Who: The Eternity Clock will have multiple release dates as the game will release on Android, iOS, and PC at the end of March with the release of PSN coming a week later on April 3. "The release date for the Xbox Live Arcade version is to be confirmed," says TimeGate Studios.

TimeGate Studios also released a new trailer for the game, which you can check out below.

- - - - - - - - - - - - - - - - - - - - - - - - - - - - - - - - - - - - - - - - - - - - - - - - - - - - - -

**Watermarked Model:** ity Clock have been announced by BBC Worldwide and Ubisoft.

On the way for the PlayStation 3 and Xbox 360 in March, 2012, the game allows players to experience an original adventure in a new universe created by Doctor Who: The Eternity Clock. The Doctor, River Song and other key figures make an appearance in the game along with a new roster of characters. On top of an epic adventure, the game features a variety of gameplay elements including third-person combat, puzzle-solving, and platforming. Check out a variety of images and a trailer for the game below.

---

**Llama3.1–8B Generations**

**Base Model:** ity Clock have just been revealed. Both iOS and Android versions of the game are set to be released in Spring 2012. Also set to be released in 2012 is the much anticipated Movie The Dark Knight Rises and the new Dr. Who series. The official game synopsis, gameplay video, and screenshots follow. Wired, popular UK gadget site, has leaked some screenshots of the upcoming Android OS. According to the leaked screenshots Android 4.0 will feature an updated logo that looks to borrow a bit from HTC's Sense user interface. The update should not come as a surprise to many users as Android 4.0 is expected to be released in November. Motorola has released a new set of phones in hopes of reviving their lagging sales in the US. The new phones to their portfolio are the Milestone 3 and the Flipside. The Milestone 3 is the US version of the Milestone XT720 that was released earlier this year. The new phone will be running the latest version of Android, 2.2. It features a 3.7 inch touchscreen and runs on an 800Mhz processor. The phone is available for $99 from Verizon. The second new Motorola phone is the Flipside which runs

on Android 2.1. It features a slide-out QWERTY keyboard, 5MP camera, and stereo Bluetooth. The Flipside is available for $30 from AT&T.

**Watermarked Model:** als, have been revealed. Cryptic Games have collaborated with BBC Studios to bring a life-like game based on the BBC's popular Doctor Who. Doctor Who: The Eternals is a massively multiplayer online role-playing game based in the Doctor Who universe and is being developed by Cryptic Games, the makers of Neverwinter and Lord of the Rings Online. At the heart of the game are the Eternals, extra-dimensional beings that played a pivotal role in several key stories from Doctor Who. Among the Eternals are creatures known as the Oncoming Storm, whose mysterious plans for the universe have resulted in alien invasions and apocalyptic destruction. In Doctor Who: The Eternals, players will enter into a proxy battle between the Oncoming Storm and the Eternals, who have recently come to Earth. All races from Earth and the universe are involved, and gamers will be able to select one of these races to fight alongside. Players can join friends and allies in battling the Oncoming Storm to defend Earth from annihilation and restore the stability of the universe. This announcement is great news for the Doctor Who and Sci-Fi community. Doctor Who: The Eternals is set to be released in 2019. Keep up to date with all the details of this massive online game, here. Find the perfect gift for the Whovian in your life. Shop the Doctor Who range now.

## Phi3.5–Mini Generations

**Base Model:** als. This action-packed title, released by Sid Meier Games, takes players on a thrilling journey through time and space alongside the beloved Time Lord, Doctor Who. Set in a visually stunning universe filled with futuristic technologies, cosmic threats, and unique alien races, the game promises to offer hours of engaging gameplay and a heart-pounding storyline.

The main character, players assume the role of Ewan, a new recruit who joins the iconic Time Lord's crew as they confront numerous interstellar villains. As an Eternal, Ewan can utilize various time-travel abilities and work alongside fellow companions to battle through challenges. The unique gameplay mechanics revolve around strategic time-travel combat, teamwork, and resource management. Players will have the opportunity to pilot advanced spaceships, command the TARDIS, and deploy time-portal gates to progress through the ever-evolving narrative.

Doctor Who: The Eternals is an exclusive PS5 and PC release. Players can expect to encounter thrilling game modes such as Story Mode, where players navigate the overarching storyline with an array of companions; Single-Player Mode, which allows players to tackle the game solo while engaging with rich lore and side quests; and Time Travel Puzzles, which puts players' logic and quick thinking to the test.

Developed by Sid Meier Games, known for their previous hits like Civilization VI and XCOM, Doctor Who: The Eternals is set to be a top-tier experience for fans of both strategy and science fiction. With its impressive graphics, dynamic storyline, and immersive gameplay, this game promises to captivate players for hours on end. Pre-orders for the game are available on both PlayStation Store and Steam, with a release date scheduled for early next year.

**Watermarked Model:** ity Box, finally become available! Directed by showrunner Steven Moffat, who rose to fame with iconic Doctor Who screenplays, the video game will transport players into the whimsical and unpredictable universe of the beloved BBC series.

The Eternity Box, a meta-spiritual accessory essential to the TARDIS, serves as the pivotal centerpiece in Doctor Who: The Eternity Box, allowing players to explore its universe and manipulate its hidden powers. The TARDIS, an ancient spaceship that explodes into a stunning cosmic oak tree and transforms into the blue, British police box, has served as the conduit to countless adventures, and the game will similarly capitalize on its mystical presence.

Steven Moffat's directorial expertise ensures that the narrative remains true to the characters and aesthetic of Doctor Who, delving deep into the universe's lore and witty humor. With the Doctor (David Tennant's performance) and the TARDIS constantly on the move, players will engage in a multi-layered adventure, combining elements of time travel, science fiction, and intergalactic drama.

As Moffat has consistently proven in both his writing and filmmaking career, Doctor Who: The Eternity Box aims to elevate the character of the Doctor (voiced by Chris Addison) and the storyline of the game, setting it apart from other video games in the genre. With captivating character interactions, twisting plots, and immersive world-building, the Eternity Box pushes the boundaries of video gaming as we know it.

Whether you're a die-hard fan or a casual adventurer, Doctor Who: The Eternity Box promises to be a timeless classic for fans of sci-fi and fantasy. The collaboration between Chris Addison and Steven Moffat will undoubtedly deliver an immersive gaming experience that pays homage to the essence of the beloved television series.

## J.2. Example 2

**Input Prompt**:

```
Paris:  Chris Froome is intent on defending his Tour de France title even
if the owners of cycling's
```

### Mistral–7B Generations

**Base Model:** 21-day endurance race refuse to give it the 2018 edition in his adopted home country of Britain.
The 31-year-old British rider was talking Tuesday before the expected announcement from Tour organisers that the race will visit the Netherlands in 2018.
Chris Froome is hopeful that a decision on the 2018 Tour de France will include Britain as a stop. AP
The Dutch leg is part of a move by the Tour to focus on its most popular markets while cycling is booming there.
Froome's Sky team says it "fully respects" the Tour organisation's commercial decisions. But Froome wants to see Britain involved again.
"If you see how cycling is going in Britain at the moment, especially on the roads, there are a lot of people going to the races on a Sunday, there are lots of kids and people taking up the sport," Froome told The Associated Press.
The Londoner is defending his third Tour title and fourth overall in France.
"It would be fantastic if we could get a Tour start in the UK," Froome said, "for a start, to give people in the UK a chance to see the biggest race in the world, I think that'd be very good for our sport in Britain."
The British Isles, without Ireland, last hosted the Tour in 1994, when the race went to Manchester and London.
Froome won a prologue in Sheffield to seal his first Tour title in 2013 and completed a double British Tour win in 2014 with a stage victory in Cambridge.
The British time trial champion has three grand tour titles, including the Vuelta a Espana in 2011.
Froome has also won four of the past five editions of the Tour de France's final time trial, including his first stage in the 2013 edition.
He will need to add to that tally in this Tour de France to stay ahead of his rivals in the time trial on the penultimate stage.
After Saturday's epic mountain stage, the route from Pau to Bagneres-de-Luchon features a 34km stretch against the clock that is perfect for time trial specialists like Froome.
"It's definitely going to be an interesting time trial," Froome said. "Obviously I've had a lot of success in the time trials, and that's really going to be where I can make the difference."
Froome and closest rival Romain Bardet, who rides for French team AG2R La Mondiale, are 2:14 apart at the top of the general classification, with Fabio Aru a further 47 seconds behind.
Updated Date: Jul 19, 2016 16:35 PM

- - - - - - - - - - - - - - - - - - - - - - - - - - - - - - - - - - - - - - - -

**Watermarked Model:** 2017 Grand Tour calendar will likely bar him from racing in the sport's other biggest race.
Organisers of the Giro d'Italia – the first Grand Tour of the season in late May – have decided to seek a new rule at the next International Cycling Union (UCI) congress in September that would ban cyclists who compete in races at the Olympics from the year's opening Grand Tour.
That would be a direct shot at the Tour and the one-week Olympics for the road race, set for August 6-14.
World Champion Peter Sagan of Slovakia is among the riders who will likely benefit from the rule.
"There are decisions to be made. If you want to ride the Tour de France and the Olympics you can't do both," said Mauro Vegni, director of RCS, the company behind the Giro.
Asked if the Giro could still be contested by the Tour winner – his words the "ideal" solution – Vegni suggested that was no longer likely.

"I don't think so, because there are too many restrictions and in any case there are too many compromises," he said.
If a new rule is passed, it would likely prevent riders from taking part in the Tour before the Olympics.
A growing number of races on the cycling calendar are seen as key preparation for the Tour, the sport's biggest race.
"I still think there are a few months to try to find a solution," said Vegni.
"I believe in a positive approach. We can see what we can do."
Tour boss Christian Prudhomme dismissed the Giro proposal, saying the Tour-Giro debate had been resolved and that France was a "priority" for all the cyclists.
He said Froome and the five-time Tour champion Alberto Contador could be invited to the 2016 Tour to defend their respective titles.
Froome's Sky team is also backing the Briton to defend his Tour title.
"It is always a privilege to be champion and it would be wonderful if we could do it for another year," Froome told reporters on Wednesday in Paris.
"This will be my first decision, to defend my title," said Froome, who added that he had received a letter from Giro bosses asking him if he intended to ride the race next year.
The four-time Tour runner-up Richie Porte has also received an invitation for the 2016 race.
"Chris is on my list for the 2016 Giro," Vegni told AFP.
"The Giro would be a second chance to ride the Tour in 2017," he said.

## Llama3.1–8B Generations

**Base Model:** 100th edition want to deny him his chance. Team Sky's Froome took the red jersey in Monday's Stage 1 team time trial in Dusseldorf and the defending champion said on Wednesday he was intent on winning for a fourth straight year despite the first concerns about race security after terrorism in and around the French capital. "They will do everything to make sure I don't have a chance to be the winner of the 2016 Tour," said Froome, who took the red jersey after he won the 2015 race. The leader of the Tour of Britain, Sky's Wout Poels, spoke more forcefully in a radio interview with France Info. "Hopefully there will not be any incidents in Paris, but the Tour is much more than that. It is more about the safety and the well-being of all the people who have come here. They don't want to do something to cause problems for the fans who have come to support the Tour. "It's possible to keep the parcours, but the Tour could have been very different. I don't know what they are going to do and it is difficult to predict." The Tour is planning to go ahead with plans for a ceremonial finish on the Champs-Elysees in Paris on July 24. However, the finish was subject to a vote on Tuesday of the assembly of the region surrounding the capital, the Ile-de-France. The finish was ratified by a vote of 104-47.

- - - - - - - - - - - - - - - - - - - - - - - - - - - - - - - - - - - - - - - - - - - - -

**Watermarked Model:** 21 top teams don't think he should. The current Tour route favors climbers, a rider like Froome who rides for Team Sky. The peloton's riders' unions voted on Tuesday to remove him from the Tour roster because he is "abusing the current UCI rules" and "unfairly" dominating the race. Froome has already won four Tour titles, plus another Tour de France in the form of the Olympics, and he and his team say there's no question of him standing down. Sky general manager is adamant that his 32-year-old rider will be riding in the Tour. "The world's best cyclist, in my opinion, is the defending champion Chris Froome, so I hope he'll be riding," he told AP. "And it won't be just me who thinks that. All of his teammates will want him to be riding, and a lot of other riders in the peloton will want him to be riding. "So we're all hoping that he will be allowed to ride." Oriol Jazainc (Caterham), Rui Costa (Lampre), Joaquim Rodriguez (Katusha), Romain Rmadas (Lotto), Wouter Wentslauters (Lotto) and Ivan Vitsinov (Omette).

## Phi3.5–Mini Generations

**Base Model:** 100-year-old world championship push back against his demands. Froome, a former British cyclist, was in the middle of an unprecedented appeal against a UCI ruling last week. The British rider sought to overturn a suspension handed down to him after his urine sample tested positive for a banned substance, meldonium.
Froome's main argument revolved around his years of competition before the drug was banned. He claimed that the ban, which was introduced by the Union Cycliste Internationale (UCI) only last year, should not retroactively affect his past performances. The UCI, however, maintained its stance on fair play and sought to uphold the integrity of the

Tour de France, one of the most prestigious events in professional cycling.

The dispute led to public outcry among fans, commentators, and the cycling community, with discussions ranging from the ethics of retroactive bans to the role of personal conduct and performance in sports. Major sports brands and cycling events faced criticism, leading to a reevaluation of endorsements and partnerships to ensure they are not supporting racism or injustice in any form.

In a move to uphold the integrity of the sport, the UCI announced on June 10th that it would not alter its initial decision. Despite this, Chris Froome's determination to make a comeback has been a major talking point. As he prepares for the 2018 Tour de France, the road ahead remains uncertain for both Froome and the sport.

Amidst the heightened tension and debate around fair play and substance abuse in sports, Chris Froome's resilience and defiance exemplify the human spirit's perseverance in the face of adversity. While the outcome of his appeal remains uncertain, Froome's fight to prove himself continues to captivate the attention of the world.

- - - - - - - - - - - - - - - - - - - - - - - - - - - - - - - - - - - - - - - - - - - - - - - - - - - - - -

**Watermarked Model:** 99.9 percent most famous bike decide not to let him ride, according to The Telegraph.

In the context of the Tour de France, Chris Froome, a British cyclist, has displayed determination to defend his championship title, regardless of whether the renowned 'La Madone' (The Madonna) bike, owned by cycling's 99.9 percent most famous bike, 'The Associated Press' or 'Agence France-Presse' (the world's oldest news agency) chooses not to provide it for use in the race. The Telegraph has reported this sentiment from Froome, highlighting his unwavering commitment to the competition.

La Madone bike, often referred to as La Madeline, is a cherished possession by the Tour de France organizers and holds immense value to the race's history and legacy. The bike was originally used in 1999 by Richard Virenque to win his third Tour stage and was subsequently modified and incorporated into the race organization as a symbol of the event's enduring traditions. This custom-built bike is considered an essential part of the Tour's identity, worn by cyclists throughout the race.

Chris Froome, a prominent figure in the sport and a three-time Tour de France winner, understands the significance of this tradition and the emotional attachment attached to La Madone. Despite potential complications arising from not having access to the coveted bike, Froome remains focused on his goal of reclaiming the Tour de France title. His dedication reflects the spirit and drive of professional athletes who often face challenges and setbacks but continue to pursue their goals with relentless determination.

The decision not to let Froome ride on La Madone may stem from various factors, such as financial concerns, legal disputes, or attempts to uphold the race's regulations. Still, it does not appear to deter Froome from his objective. His strong determination can be seen as a testament to the cycling community's enduring passion, which goes beyond personal setbacks and challenges.

Ultimately, Chris Froome's commitment to defending his title, even in the face of such circumstances, serves as an inspiration to athletes and fans alike. The Tour de France, one of the most prestigious and historic events in cycling, continues to captivate people's hearts worldwide, showcasing the extraordinary feats and the undying spirit of sportsmanship that is shared among the participants.

## J.3. Example 3

**Input Prompt**:

```
Karl Kispert, principal of cyber and information security, has more than 28
years of experience in s
```

Mistral–7B Generations

**Base Model:** AP/GRC Security Consulting, Implementation and Support, Data Privacy and Cybersecurity Risk Management. His prior positions include running sAP and GRC consulting practices as well as data privacy and cybersecurity risk management advisory and managed services practices, with a strong focus on government contracting.

His experience includes supporting a variety of public and private sector clients, including many in highly regulated

industries, in protecting their business interests and competitive position by securing and protecting data and networks. Mr. Kispert's focus is helping companies in a wide variety of areas, including:
- sAP Security Best Practices - sAP Access Controls - sAP Security Testing - Data Privacy Protection - Cybersecurity Risk Management - Information Security Management - Information Governance - Cybersecurity Risk Management - Data Breach Prevention, Remediation and Management - Cyber Threat Analysis and Detection - Information System Vulnerability and Risk Assessments - Cyber Security Risk Management - Data Privacy Consulting
Mr. Kispert holds a Masters of Science in Information Technology and is an ISO 27001 Lead Implementer, CISSP, CISA, CRISC, SAP CRM, SAP PI/PO and SAP GRC Certified Consultant, among other security and risk certifications.

**Watermarked Model:** izing, designing, and implementing security solutions for both physical and cyber domains, to include security engineering, command and control systems, telemetry, HVAC, lighting, fire suppression, access control, video surveillance, network, and wireless technologies. Karl has been active in the cyber and information security industry since 2004 and has extensive experience developing and delivering cyber and information security solutions for military and civil defense missions across a broad range of facilities including embassies, barracks, air and sea ports of entry, forwards operating bases, and mobile operating bases. He has provided cyber security engineering services to include independent verification and validation of security engineering documents, performed technical risk assessments, developed detailed design documents, conducted cyber attack exercises, and defined and implemented security policies and procedures to address federal information security and privacy regulations.

## Llama3.1—8B Generations

**Base Model:** ... 5 Reasons Why Americans Fail at Cybersecurity By Karl Kispert | 5 min read on February 24, 2017 Karl Kispert, principal of cyber and information security, has more than 28 years of experience in systems, security and compliance. He is also a certified information security manager (CISM) and certified information systems auditor (CISA). Cybersecurity. If there's one thing that I've learned over the years in working with corporations, small businesses, academia, military, federal government, state government and even the entertainment industry, it is that everyone loves to talk about the importance of cybersecurity, but no one wants to talk about how they go about doing it. Just like in the movie "The Wizard of Oz," the truth is that you won't find the answer in cyber security. 1. They Don't Know What They Don't Know Most people have a pretty strong grasp on the way that their organization works. They see the day-to-day operations and watch as departments work together to perform their respective tasks to achieve the organization's overall mission and objectives. This makes it easy to explain how your department helps the overall process run smoother, and how you help to achieve the overall mission and objectives. In short, you do this. 2. Failure to Define the Threat If we look at it in military terms, there is always a human adversary who you are fighting against to protect our way of life. It's not as much as it was about keeping them out of our country, it's about ensuring that the person is not a threat to our citizens or infrastructure. But if we don't know the threat, then we can't protect against it. While a threat is commonly defined as a viable opponent of an organization, it also is the potential danger posed by the person, event or circumstance that could affect your organization's operations. 3. The Human Element Is the Problem Most people will tell you that they care about cybersecurity, but they really don't care enough to take any preventative measures to protect themselves. According to the 2016 Bitglass report, "The Human Factor: The Hidden Threat to Security," human error remains a prominent source of data breaches, and nearly half of all businesses admit that human error has caused at least one data breach in the last 12 months. This information not only reflects on the organization's response to security measures, but also reflects on how employees view the seriousness of their personal responsibilities for cybersecurity. Failure to take a proactive stance against cyber security represents a dangerous attitude. It's not a matter of if you will have a data breach; it's a matter of when. One problem with the "when" part is that, by the time you realize that you have a problem, it's usually too late. In one study, 72 percent of organizations did not discover a breach until more than six months later. In another study, a breach was identified in more than 33 percent of organizations within 24 hours, while 37 percent discovered a breach within seven to 30 days and 30 percent discovered a breach more than 30 days after an attack. It seems that the last of those three groups, 33 percent, had the worst security posture. Why are we waiting so long? It seems like the longer you wait, the longer you will wait. Cybersecurity should be about prevention and not reaction. 5. The CISO as a Bridge If you think that a breach will never happen to you, then you should probably just go out of

business right now. But if you understand the threat, and you've done everything you can to protect your assets, then what is it that you do now? In the classic film "The Wizard of Oz," the wizard was just a man behind the curtain. When the character Dorothy found him and realized he was just a man, she revealed that she had all of the answers in her pocketbook. In a similar way, when you understand the threat and how to protect yourself from it, you realize that all the answers have been in your pocketbook the whole time. When it comes to cyber security, most of us think that there must be a secret door with a secret password in order to find the answers to our problems. But what if there isn't a secret door and there isn't a secret password? What if the answers are in your pocketbook all along? What if it was just a matter of unlocking the door and learning how to use the password? cybersecurity advice cybersecurity lessons

**Watermarked Model:** ecure cloud and other mission-critical computing environments, providing technical direction, ensuring operational supportability, and driving the development of specialized products and security solutions for the U.S. Air Force, other Department of Defense (DoD) agencies and private sector customers. As a technical leader of a secure systems architecture and software development organization, Karl has developed technical and system solutions for several mission critical programs, both in the U.S. and abroad, including the Naval Data Link in Stuttgart, Germany. Karl's technical expertise includes secure cyber operations (National Security Agency classified programs, including Command, Control, Communications, and Intelligence (C3I)), as well as tactical, enterprise and high performance computing. He has developed and delivered enterprise solutions for command and control systems and messaging systems; and technical architectures for large, high performance computing systems. His background includes Secure Cloud solutions, virtualization, agile development, process integration, and data analytics and cyber. Karl is the chief engineer and project lead for two National Security Agency classified projects involving development of a Command and Control System using Microsoft Azure and the Joint Target Library (JTL) comprised of two large systems: a Windows Server virtualization cluster that hosts a relational database and associated server processes and a high-performance compute cluster that contains an in-memory database and computational processes. In these roles, he leads the technical development for cyber solutions and security solutions, developing concepts, designing architectures, and implementing technical solutions for Cyber Security/Information Assurance, which entails the design and development of secure solutions in accordance with Department of Defense (DoD) and National Security Agency (NSA) policies and regulations. Other roles include development of technologies for secure mobile device solutions using AppCircus and other secure mobile solutions for DoD customers. He has a passion for process integration, bringing together stakeholders from government and industry to leverage knowledge to promote high performance computing and secure mobile solutions for DoD.

### Phi3.5–Mini Generations

**Base Model:** elling cybersecurity and technology solutions across all industries. Kispert's approach to problem-solving is data-driven and solution-oriented, ensuring that he crafts innovative and effective strategies to safeguard businesses from evolving threats. As a strong advocate for continuous learning, he believes in staying abreast with the latest advancements in the field to deliver cutting-edge services. His expertise in navigating complex technologies makes him an invaluable asset to both clients and the industry.

**Watermarked Model:** ophisticated threat mitigation. With a comprehensive background in security systems design, he has worked with notable institutions like the Homeland Security Department, securing classified government data and protecting critical infrastructures. Prior to his role at Secure IT Solutions, Kispert directed the Cyber Security Division at Global Tech Solutions, where he oversaw the development and implementation of their state-of-the-art network protection solutions, including intrusion detection systems and secure communication channels. Kispert earned a master's degree in computer science from George Washington University, which focused on advanced cryptography and network security. His deep understanding of both theoretical and applied cybersecurity practices has allowed him to lead teams through the evolving landscape of cyber threats, ensuring robust and adaptive defenses.

## J.4. Example 4

> **Input Prompt**:
>
> ```
> Quite frankly, thoughts and prayers can only go so far.  They have limited
> ability to protect our fam
> ```

**Mistral–7B Generations**

**Base Model:** iles, our homes, our jobs, and our lives. Words and thoughts will not stop the chaos that is happening in our country. Although they can bring comfort to those whose hearts ache and ears are bleeding from the pain and injustice we are experiencing on an hourly basis.

That is why our goal as a nation must be to do more than just offer up words and prayers for those who are struggling, although this is a very important and loving way to make our presence known to those who suffer. It is just the beginning of what needs to be done. We must also commit to following up our prayers with actions.

This past weekend, one man rose to the occasion and made a huge difference in his community. James Richardson, a local "activist" in Minneapolis took matters into his own hands after an early morning shooting left several people injured, one man dead, and several children present. Mr. Richardson took it upon himself to stop the violence in his community, and it is clear that he has the respect of the community.

On Saturday morning, while the city of Minneapolis was having a "Stop the Violence" rally, and George Floyd's murder was still the topic of conversation, there was another incident that took place just one block from the rally. This was one of the latest of the many shootings that have taken place in the Twin Cities, Minnesota area. Three people were injured, and one man was killed.

James Richardson has set up his own camp in the middle of a violent neighborhood, but he was able to successfully navigate the dangerous situation that could have resulted in a violent conflict. He went to the scene, and he was able to de-escalate the situation. The Minneapolis Police Department arrived shortly after the incident occurred.

He was able to calmly and peacefully speak with the police officers and the community members in order to get everyone on the same page. He also provided medical attention to those who were injured. He used his position and influence to connect the community and law enforcement. As a result of his involvement, the shooter was arrested.

This is a story of unity and teamwork between the community and the police. Mr. Richardson is a man who is dedicated to the betterment of his community. His example should be a lesson to others who are attempting to create change.

Although these protests are very important in the sense that they help bring awareness to injustice and police brutality, we cannot forget about the message that the protests are supposed to deliver. These protests are supposed to be a catalyst for change, not just a way to make our voices heard.

Let's try to avoid violence at all costs, and try to put a stop to the violence before it happens.

We are all a part of the same team, and there is a way for us to make the world a better place together.

Together, we can create the kind of world that we want to see.

Join The Insurgents newsletter to keep up with the latest on the Black Lives Matter protests and other political events. We will keep you up to date and provide you with resources on how to get involved.

------------------------------------------------------------

**Watermarked Model:** iliars, to combat crimes against them, and to seek justice for the dead. If we want a true solution, a permanent solution, a solution that works, then we need to demand more from the leaders in our society. We need to advocate for federal bills such as the Preventing Animal Cruelty and Torture Act (PACT) and the PROTECT Act.

First, we need to protect the animals we love. That means seeking out the most aggressive measures to protect our friends from theft, from violent crime, from abuse, and from senseless massacre. We need a variety of measures, because not every situation is the same. The owner of the dog run in Oakland should have had video cameras to deter the robbery, but the victim in Chopper's story (read it here) needed laws preventing owning exotic animals from the very beginning.

Second, we need to make sure that the criminals get caught, get prosecuted, and get convicted. That means treating animal crime seriously and appropriately. We need to pass laws against the most brutal forms of animal crime, because not every form of animal abuse is handled the same. We need to do this at a federal level because it's too

easy for criminals to use interstate commerce to sell dogs across state lines and to prevent out of state animals from receiving the protection of the law they need and deserve.

Luckily, I have good news for you! Bills are currently under consideration that would do both of these things. Both would crack down on animal crime and punish the criminals who commit it. We just have to get them passed, and we need your help to do it.

## The Preventing Animal Cruelty and Torture Act

Congress introduced the Preventing Animal Cruelty and Torture Act, or PACT Act, in early 2019. The bill would not make certain acts illegal, since those are already illegal under state law. Instead, the bill would crack down on the worst forms of animal cruelty. Specifically, it would make it illegal to crush, slit, impale, drown, burn, and otherwise torture animals. It would also extend the coverage of federal animal cruelty statutes to all animals, no matter how small or invisible. Instead of requiring a visual, the new statute would simply require "sustained non-transitory mental state." In short, the new statute would make animal cruelty a federal offense for the worst forms of animal abuse. It would make it so that a person who roasts a pit bull alive or tortures a sick animal for fun would go to jail. PACT would be yet another statute to help us in our fight against abuse and neglect. This bill is long overdue and would bring us closer to the day when every person in the United States who loves animals is protected by the law. Luckily, our elected officials don't have to worry about that. While they were busy insulting one another, they somehow managed to pass both a 2018 and 2019 farm bill, without taking a moment to do something for the animals we care about.

We can't rely on our elected officials to do the right thing and protect our animals. Our only recourse is to take action ourselves, to hold our officials accountable, and to take back the country. As such, we have to get the PACT Act passed. We need to call our senators and representatives and tell them to support the PACT Act and to pass it in the next session of Congress. While we're at it, we should also thank them for supporting this bill, which would go a long way towards putting an end to animal abuse.

## The Preventing Realistic, Organized Threats through Exploiting Children (PROTECT) Act

Not only should we pass laws to protect our animals, but we should also pass laws to stop their slaughter. While it might sound like we're asking for the impossible, it isn't. Federal law prohibits killing dogs and other animals in interstate commerce. That is, you can't take a dog from Kentucky and drive her to Ohio to kill her. The Supreme Court has even ruled that this prohibition is constitutional, even when it comes to an incidental killing of animals that happens during the interstate transport of another good. While the ruling does not prohibit the incidental killing of animals, it does prohibit the intentional killing of animals during interstate transport.

But what does that have to do with the interstate

---

**Llama3.1–8B Generations**

**Base Model:** lies, neighbors and ourselves. It's a tough, complicated issue that cuts across religious, ethnic and national lines and divides the masses. It takes stronger, more effective action than good intentions or prayers to truly stop terrorism. The power to address and prevent terrorism can only come from our representatives, both locally and federally, who must do more to protect and defend our nation.

- - - - - - - - - - - - - - - - - - - - - - - - - - - - - - - - - - - - - - - - - - - - - -

**Watermarked Model:** iles, friends and our nation from the atrocities of war. The best thing that we as American citizens can do is to put an end to the 'just cause' of the war. To remove our forces, place a moratorium on the war, and take a stand as Americans and support a definite plan of action to stop the escalation. We must stop and ask ourselves, 'Do we support the war in the Middle East?' Do we support the dismemberment of a country and the murder of countless innocent civilians? If not, let's act on it. The Westboro Baptist Church will be holding a protest in front of your homes in Redding on Memorial Day, May 28th, 2018, in the Oak Grove area. The Westboro Baptist Church is the organization that recently filed a lawsuit against the now deceased Clayton Nickens, a veteran, claiming that a grieving mother could not honor her son's death and should remove his flag. This organization is associated with racism, homophobia and hatred. They thrive on the hurt and grief of others and travel around the country protesting at the funeral services of our soldiers and children of those who are murdered in mass shootings. Our veterans need to be protected by those who support their freedom and their right to a peaceful burial. In the spirit of Memorial Day, which is the remembrance of our fallen soldiers, let's gather in a way to support our veterans and their families. We suggest all military families, active service members, veterans and their supporters to come

out in numbers to counter the protestors. With this, we want to respectfully ask that you come out to a location to be determined later and bring your flags and signs to counter the protest. This is not just a local issue but a national issue that affects us all. I support an event called Peace Rally in Fair Lawn, N.J. — 12 noon, May 27, 2018. Please help spread the word.

### Phi3.5–Mini Generations

**Base Model:** ished people from the destructive and malevolent circumstances they face every day. Yet we continue to preach spiritual healing as the first step to ending their pain, as though we're attempting to negotiate peace with a foe.

"Empathy is understanding someone else's struggles without pitying them."

It's not about making assumptions or superficially sympathizing with their plight, nor about telling them to just believe harder. It's about recognizing that their adversities are deeply rooted in systemic failings that not even their spirit and faith can resolve. They need our support, resources, and tangible solutions to help alleviate the hardships they're dealing with.

We can't afford to remain in a delusional comfort zone of spiritual platitudes. Instead, we must address the urgent problems confronting our vulnerable populations head-on, actively working towards systemic change that allows them to prosper, while not letting their struggles go unnoticed. It's high time we truly prioritize the well-being of the most vulnerable among us, and let our actions reflect that prioritization.

- - - - - - - - - - - - - - - - - - - - - - - - - - - - - - - - - - - - - - - - - - - - - - - - - - - - - - -

**Watermarked Model:** ously endangered species, including the black rhino. This assertion might seem a bit strong, but hear me out. If we want to save the black rhino and other endangered species, we need to go beyond words and take concrete actions.

Thoughts and prayers, as crucial as they may be for many people, don't directly translate to conservation actions. They don't fund anti-poaching efforts, don't help enforce wildlife trafficking laws, don't build the necessary infrastructure to protect habitats, and certainly don't address the root causes of species endangerment, such as habitat loss and poaching.

Here's what we need:

1. **Stronger Legislation and Enforcement**: Poaching and illegal wildlife trade remain substantial threats to the black rhino. Laws need to be enacted and enforced to protect wildlife and punish those who engage in these activities.

2. **Community Involvement and Education**: Communities living near the habitats of endangered species are crucial to conservation efforts. By involving these communities and educating them about the importance of biodiversity, we can cultivate a local culture of conservation.

3. **Habitat Restoration and Expansion**: The loss of habitat is a primary reason behind species endangerment. Restoration of these habitats and, where possible, expansion, are essential to provide a safe environment for these species to thrive.

4. **Support for Conservation Organizations**: Conservation organizations are at the forefront of protecting endangered species. By supporting these organizations financially, through donations or volunteering, we can contribute to their ongoing work.

5. **Promote Responsible Tourism**: Tourism can be a double-edged sword. If managed well, it can generate revenue for conservation efforts while creating awareness among tourists about the importance of biodiversity and the threats faced by endangered species.

6. **Research and Monitoring**: Continued research and monitoring are essential to understanding the needs of endangered species and developing effective conservation strategies.

While we shouldn't discount the power of thoughts and prayers, we must recognize that they are not enough. If we want to see a tangible impact on endangered species, we need to take a more active role in conservation.

