# OpenReview forum: "GaussMark: A Practical Approach for Structural Watermarking of Language Models"
_ICML.cc/2025/Conference — ICML 2025 poster_

### Official Review · Reviewer_x6rs · 2025-03-12

**Overall Recommendation:** 2

**Summary:**

This paper proposes a watermarking for large language models. The watermarking injection process has an almost negligible overhead and applies to LLMs of any structure. The authors observed that a small perturbation to the parameters of LLMs will not significantly affect the model's performance. Based on this, the authors inject the watermarking into the model's weights. Specifically, in the watermarking injection phase, sample the watermarking from the Gaussian distribution, then add it to the parameters of the model (similar to model merging). In the detection phase, based on hypothesis testing, the watermarking is detected by calculating statistical information related to the gradient of the parameter part of the watermarking injection. A lot of theoretical analysis is done in the paper, and experiments show that this watermarking has advantages in detection success rate, impact on text generation, and anti-attack.

## Update after rebuttal

The concern about "the effect of increased noise on model performance is not provable" has not been solved, so I maintain the original score.

**Claims And Evidence:**

The core innovation of the paper and the analysis of experimental conclusions are clear.

**Essential References Not Discussed:**

The discussion on the robustness of the LM watermark is necessary, and diverse adversarial attacks against the watermark can be found in [1].

- [1] Liu, et al. On Evaluating The Performance of Watermarked Machine-Generated Texts Under Adversarial Attacks.

**Experimental Designs Or Analyses:**

The reviewer has checked the soundness and validity of the experimental designs. The experimental setup and analyses are reasonable.

**Methods And Evaluation Criteria:**

This paper has used multiple datasets to empirically demonstrate the negligible influence on the original model. However, this evaluation method is ultimately limited, for example, the effect of the added parameter noise on the safety alignment of the model is not considered.

**Other Comments Or Suggestions:**

- Detailed discussion and analysis are needed to prove small perturbations cannot destroy the safety alignment of the model.
- In the watermarking detection phase, it is necessary to obtain the user's prompt, which is almost impossible in practice due to privacy and timeliness. Although the experiment considers the case of prompt damage, this is also under the condition of knowing the real prompt.
- The watermark detection algorithm and the examples are all based on single-round dialogue scenarios. However, in actual situations, users often gradually guide the model in the form of multiple rounds of dialogue. In this case, how to detect the watermark? For example:
    - user: please introduce the LLMs safety.
    - assistant: LLMs safety includes ... (1000tokens)
    - user: thank you, please summarize in one paragraph.
    - assistant: ok, ...(400tokens)
- Typos:
    - The title of Figure 13-b does not match its horizontal axis.
    - Should the horizontal and vertical axes in Figure 5 be TPR and FPR?
    - The colors of low p-values in Figure 9 and 10 should be consistent.

**Other Strengths And Weaknesses:**

Strengths:

- It is simple to implement the proposed watermark algorithm. Watermarking injection and detection do not require much memory and computational overhead.

- Rich theories to prove the correctness of detection metrics.

- The redundancy of weight information is used to directly add watermarking to weights without forcing the selection of tokens, which maximizes the retention of language characteristics. Experiments have also proved this.

Weaknesses

- The reviewer's biggest concern is that the effect of increased noise on model performance is not provable, so there is a possibility, though a very low probability, of damaging model performance.

- The choice of parameters of watermarking and the location of watermarking injection rely on experience, so extensive experiments are needed to prove the applicability of experience, which includes more models and more different types of datasets, such as translation and code.

- Need more types of attacks to detect the robustness of watermarkings, such as emoje-attack or document-level attacks that copy text to large amounts of human text.

**Questions For Authors:**

Please see the comments and weaknesses.

**Relation To Broader Scientific Literature:**

The paper introduces the related work in detail and the proposed method is novel.

**Theoretical Claims:**

The reviewer has checked the correctness of the core theoretical claims of the paper.

---

> ### Author Rebuttal · Authors · 2025-03-27
>
> We thank the reviewer for their careful attention to our work.
>
> ## Model Quality: Further benchmarks and safety
>
> We agree that more benchmarks, such as translation or code, would be better and we acknowledge in lines 407-408 2nd column that such checks are necessary before deployment.  We also agree that safety benchmarks would be beneficial to consider.  On the other hand, note that the suite of evaluations we did perform approximately matches or exceeds much of the empirical watermarking literature and the realities of resource constraints surrounding compute limit our ability to conduct a wider sweep.  We agree that considering the effect of GaussMark on safety is an interesting direction for future research.
>
> ## Further corruption attacks and detection of multi-round conversations
>
> Regarding further corruption attacks, we agree that more sophisticated attacks would be interesting to study, but again note that the set of attacks we did investigate is fairly large relative to many of the other works proposing watermarking schemes.  In particular, emoji attacks amount to structured token-level insertions and are likely less able to damage GaussMark than the random insertions we did investigate.  While certainly more sophisticated document-level attacks are relevant to the investigation, note that we simulate a weaker version of this attack by adding random text to the beginning of the examined corpus.  Further experiments are, of course, always better, but are subject to the same resource constraints mentioned above.
>
> On the subject of multi-turn dialogue, we believe that this is partially addressed by our discussion in lines 376-381 2nd column, where we mention that ignorance of the prompt does not substantially affect our ability to detect the watermark.  For more detailed discussion of our robustness experiments, please see Appendix F.
>
>
> ## Typos
>
> Thank you for catching these; we will fix them in the revision.

---

### Official Review · Reviewer_XB7Z · 2025-03-13

**Overall Recommendation:** 4

**Summary:**

This paper introduces GaussMark, a novel watermarking scheme for language models. The approach involves adding small Gaussian perturbations to a single MLP layer during generation and using statistical tests based on Gaussian independence to detect watermarked text. The watermarking scheme comes with formal statistical guarantees and is efficient to implement in practice. The authors test GaussMark on Llama3.1-8B, Mistral-7B, and Phi3.5-Mini, showing it can effectively watermark text while maintaining performance on the SuperGLUE, GSM-8K, and AlpacaEval-2.0 benchmarks.

**Claims And Evidence:**

The paper's claims are generally well-supported by both theoretical analysis and empirical evidence.
- Statistical validity: The authors provide formal proofs about the power of the GaussMark test under an assumption that the language model can be approximated as a linear softmax model.
- Minimal impact on generation quality: The empirical evaluations on SuperGLUE, GSM-8K, and AlpacaEval-2.0 convincingly demonstrate that GaussMark has minimal impact on model performance.
- Efficient implementation: The timing measurements clearly show that GaussMark has negligible impact on generation latency and reasonable detection time, outperforming alternative approaches like KGW by orders of magnitude in generation time.
- Robustness to corruptions: The paper presents evidence that GaussMark is somewhat robust to token-level corruptions and roundtrip translation, though it is less robust than some alternative approaches (e.g., KGW-2).

**Essential References Not Discussed:**

I am not aware of other key references that are not discussed.

**Experimental Designs Or Analyses:**

The experimental designs are sound and comprehensive.
- The authors use standard benchmarks and datasets (C4 for generation, SuperGLUE, GSM-8K, AlpacaEval-2.0 for evaluation).
- Ablation studies thoroughly explore the effects of different hyperparameters.
- Error bounds and multiple seeds are used to account for stochasticity.
- Corruption experiments (token insertion/deletion/substitution, roundtrip translation) are comprehensive and systematically test different types and levels of text modification.

One minor limitation is that the generation evaluation could benefit from more diverse text domains beyond just C4.

**Methods And Evaluation Criteria:**

The methods and evaluation criteria are appropriate and very comprehensive.
- The evaluation across multiple models (Llama3.1-8B, Mistral-7B, Phi3.5-Mini) provides good coverage of different model sizes and architectures.
- The benchmarks (SuperGLUE, GSM-8K, AlpacaEval-2.0) appropriately assess different aspects of model quality.
- The corruption experiments (token insertion/deletion/substitution, roundtrip translation) are comprehensive and test realistic scenarios.

**Other Comments Or Suggestions:**

I think the paper is generally very well written. One minor suggestion is that in my opinion the main text could benefit from a slightly more thorough description of the method hyperparameters $\sigma$ and $\theta$, see questions.

**Other Strengths And Weaknesses:**

Other strengths:
- The paper is comprehensive and clearly written.
- The method is practical, with little computational overhead, and easily deployable, requiring no modifications to inference pipelines.

Weaknesses:
- The method is not as robust to more sophisticated attacks (e.g. roundtrip translation). Other attacks, e.g. targeted or non-consecutive paraphrasing could be more thoroughly explored.
- The paper could benefit from more detailed discussion about the effect of the choice of model parameters to be perturbed, see questions.
- The method may not be as applicable to smaller models, where weight perturbations may have larger effects.

**Questions For Authors:**

1. How valid is the linear softmax assumption for models that differ from the standard Transformer architecture, e.g. convolutional long-context models or models that combine global and local operations (e.g. sliding window attention)? This may be crucial for determining whether the method may lead to greater performance degradation on different models.
2. Do the authors have a general method for choosing $\sigma$, the noise variance, and $\theta$, the model parameters on which to apply the watermarking procedure, beyond simply testing the different options empirically?
3.  Have the authors investigated how well GaussMark performs on models that have undergone quantization or other post-training modifications? This may be relevant for deployment scenarios where models are often compressed for efficiency.

**Relation To Broader Scientific Literature:**

The paper is well-situated within the watermarking literature for language models. The authors situate their work clearly within the landscape of prior work and clearly identify their novel contributions:
- Unlike token-level approaches like Kirchenbauer et al. (2023) and Kuditipudi et al. (2023), the proposed method makes edits to (few) model parameters.
- The authors mention drawing inspiration from work on model weight manipulation, including model soups (Li et al., 2023; Sun et al., 2023; Fernandez et al., 2024; Sharma et al., 2024).

**Theoretical Claims:**

I checked the high-level proof ideas for the main theoretical claims in the paper, Propositions 3.1 and 3.3. The theoretical claims in the paper appear sound as far as I can tell and use standard statistical techniques.
- Proposition 3.1 (statistical validity) uses the rotational invariance of Gaussian distributions to show that the test statistic follows a standard normal distribution under the null hypothesis.
- Proposition 3.3 (power bound) derives bounds on the statistical power under the linear softmax model assumption. The authors are careful to state the details of their assumptions and Lemma I.6 justifies why this assumption may be reasonable in practice.

---

> ### Author Rebuttal · Authors · 2025-03-27
>
> We thank the reviewer for their careful attention to our work.
>
> ## Choice in Perturbation Parameters
>
> We agree with the reviewer that a more thorough understanding of which parameters to perturb would be beneficial and included Figures 7-12 as some empirical guide as to the effect of layer, parameter, and variance on both detectability and model quality.  At present, we do not have a strong theoretical intuition for which parameters are optimal to watermark beyond the notion that higher dimensional parameters with variance neither too high nor too low work the best.  We suspect, as we allude to in Line 438 2nd column, that the optimal choice is to perturb multiple weights at once and it is an interesting direction for future research to better understand and develop a more general intuition for this question.
>
> ## Validity of linear softmax assumption beyond transformers
>
> This is a good empirical question and we are not sure as to the answer as we only experimented with transformer architectures.  We suspect that recurrent networks likely have complicated nonlinear effects that may reduce the power of Gaussmark, but that convolutional operations, being linear, may still allow for the linear softmax assumption to apply, at least in later layers, but this is entirely guesswork and this is an interesting direction for future work.  Note that this approximation is only important to theoretically bound the power of the test, and statistical validity holds unconditionally.
>
>
>
> ## Quantized models
>
> The question of precision is very good and we did not experiment with this so we are not sure of the implications.  We suspect that our method would still work (although possibly with reduced power) as several weights can handle larger variance without reducing text quality (cf. Fig 11-12).  Optimizing Gaussmark for quantized models is an excellent direction for future work.

---

### Official Review · Reviewer_36HD · 2025-03-13

**Overall Recommendation:** 3

**Summary:**

This paper proposes a watermarking scheme for large language model output (LLMs) using gradient-based test statistics. The properties of the proposed method, GaussMark, are analyzed theoretically to derive signficance levels and power of the test. The method and its properties (efficiency, quality-preservation, and detectability) are empirically verified using three language models.

**Claims And Evidence:**

There are four main claimed properties of the method, GaussMark

* It is statistically provable. I found no substantial flaws in the theory part, but the derivation relies on coarse assumptions that might not hold for real-world models. For instance it uses a linear approximation.
* It does not deteriorate the quality of the outputs. I think the references and the empirical results support this point well. However, I think this should rather depend on the layer or the model where GaussMark is applied. But generally I am convinced that the method can be applied to models without deterioration in quality
* It is robust to simple post-processing techniques. Empirically verified through experimental evaluation, but no theoretical results concerning this point.
* It is very efficient. I think this claim is established well (it is conceptually straightforward to see) and experimentally demonstrated in Figure 3. However, I profoundly disagree with the claim that detection has the same memory requirements as inference made (in Section 5, l.434). For detection, a backward-pass is required which usually consumes substantially more memory.


However, as the authors acknowledge, there are already a lot of watermarking approaches in the literature. I think this necessitates further competitive comparison concerning the key criteria: detectability, performance-degradation, and robustness to modifications. There is no direct comparison to any method in the main paper. This is an essential weakness to me as I think this contribution needs to be judged in relation to existing methods.

**Essential References Not Discussed:**

see above, otherwise I just think more methods should be studied empirically.

**Experimental Designs Or Analyses:**

There seems to be on issue regarding hyperparamter selection: The hyperparameter values (sigma, and selection of layer) are not independent of the benchmarks, as it seems the same data and metrics were used to select them as far as I understand. It would be more sound to use a validation subset of the benchmark for selecting sigma/the layer according to the performance.

**Methods And Evaluation Criteria:**

I found no major issues with the experimental setup besides lack of baselines.

**Other Comments Or Suggestions:**

## Rebuttal update

increased score from 2 -> 3 based on rebuttal (see below)

**Other Strengths And Weaknesses:**

Writing and Clarity: The (main) paper is very well-written, besides a little too many references to the appendix, especially in the theoretical statements. Some parts, especially at the end of Section 3, just read like a concatenation of references to the appendix, which is not a nice read. In terms of the contents, think the paper could also fit a journal format well. There is a lot of theory in the appendix. I don’t think the theoretical contribution fits the conference review format as it would be very time-consuming to completely review.

Scope: A fundamental limitation of the proposed method is its requirement for access to model gradients. Most watermarking schemes derive their utility from working with black-box API access – for instance, when detecting if students have used ChatGPT or Claude for assignments. Since this method requires gradient access, it cannot be applied to the predominant use case of commercial API-accessible models, severely limiting its practical impact.

On the positive side, I appreciate that the proposed water-marking method is straightforward to implement and has no overhead in generation time. The topic of watermarking further is timely and relevant.

**Summary** While the approach shows some interesting theoretical properties and can be easily implemented, there are concerns regarding a performance comparison to other methods, novelty in contrasts to earlier works using similar statistics, practicality/scope, and theoretical consistency that significantly impact its contribution.  While the paper's execution and writing is generally good and the topic is timely, I currently cannot provide an accept score because of the many issues raised.

**Questions For Authors:**

*  Captions in Figure 5: Why are the x axes labeled “Size of test” in Figure 5? I think these are TPR/FPR curves, right?

*  To preserve utililty, the variance of GaussMark has to be chosen very small (e.g., 1-e5). For efficient generation quantization of weights is common. However, when using quantized models, e.g., with 16-bit floats, the variance used is far lower than o the representable precision (machine epsilon for fp16 is around 1e-3). Do you think for GaussMark’s performance full-precision computations are necessary?

**Relation To Broader Scientific Literature:**

The methodology and the test statistic appears to be directly adapted from Pawelczyk et al. (2025), "Machine Unlearning Fails to Remove Data Poisoning Attacks" (ICLR 2025; preprint available since June 2024), which established a nearly identical test statistic approach using Gaussian samples and input gradients. The current work essentially applies this existing methodology to LLM watermarking without proper acknowledgment. Even the key insight regarding improved performance with higher input dimensionality was previously established in the ICLR 2025 paper. The absence of proper citation and comparative discussion substantially diminishes the novelty claim.

**Theoretical Claims:**

The paper contains a potentially concerning theoretical inconsistency regarding the noise parameter σ:

In Section 3.2 (before Proposition 3.3), the authors justify the linear model assumption by arguing it holds when σ is "sufficiently small"
Later, when discussing results, they analyze the case where σ → ∞, and state that larger sigmas are required for the method to work properly.
This contradiction undermines the theoretical foundation of the work. Which condition is actually required for the method to function properly? The paper needs to resolve this fundamental inconsistency.

The entire method seems to rely on the assumption in eqn. (3) that seems relatively coarse and drastic. Details of the approximation are deferred to the Appendix, but I think this part is crucial to understand the statistical grounding and reliability of the method.
I have the impression that a lot of theoretical effort was invested in a method that is based on this uncertain approximation. Therefore, I am not sure how valueable the theoretic results are in practice.

---

> ### Author Rebuttal · Authors · 2025-03-27
>
> We thank the reviewer for their careful attention to our work.
>
> ## Comparison to other watermarking schemes
>
> Due to space, we deferred an extensive comparison with Kirchenbauer et al to appendix H, alluded to in the paragraph beginning in line 406.  Please see that for results and explanation of why we chose this scheme as a benchmark.  We are happy to move some of those comparisons to the main body subject to space constraints.
>
> ## Linear approximation
>
> We emphasize that while our motivation and power bounds rely on simplifying assumptions, in Proposition 3.1, *the statistical validity of the test holds without assumption*, in particular not requiring the linear softmax approximation to hold.  The power bound is a theoretical result to guide our understanding of the effect of problem parameters, but is not directly relevant to practice; we verify empirically that Gaussmark has good power in detecting watermarked text.  The discussion following Proposition 3.3, wherein we take $\sigma \to \infty$, is meant to serve as a guide for intuition and we give a finite bound on how large $\sigma$ must be in Corollary I.5, to which we allude in line 298.  Furthermore, this intuition is reflected empirically in Figure 7, where we see that when variance is high for early layers we see a *decrease in power, manifesting through increased p-values when the variance increases*, suggesting that the lack of linear approximation hurts the test’s power.  In the revision we will include additional discussion clarifying this point.
>
> ## Memory Requirements
>
> We agree that a full backward pass requires more memory than a forward pass, but we only keep track of gradients with respect to a single parameter matrix, usually in a late layer.   We are happy in the revision to clarify that choosing a parameter at a much earlier layer results in additional memory as well as provide empirical results on the memory cost of detection relative to inference.
>
> ## Choice of parameters depending on benchmarks
>
> We believe there may be some confusion as to when a validation set is important.  Our goal is to find watermarking parameters that do not adversely affect model quality (as measured by the chosen benchmarks) while still ensuring detectability, not to claim generalization.  As such, including the entire benchmark set in evaluation is *more stringent* as it reduces the standard error of the estimate of model quality, requiring the watermarked model to perform better relative to the benchmark in order to qualify as a model that does not reduce quality.
>
> ## Relation to Pawelczyk et al 2025
>
> Thank you for pointing this out; we agree the (very nice) work of Pawelczyk et al 2025 is relevant. We will discuss it in our revision, summarized here.
>
> First, we emphasize that while GUS perturbs *data* and takes gradients wrt inputs, GaussMark perturbs *parameters* and takes gradients wrt the same, as is appropriate for these very different settings.  Indeed, that work considers unlearning and thus aggregates the inner product test statistic over multiple data points on a model that is passed through some noisy channel induced by the unlearning process, while ours focuses on a fixed data point and evaluating the likelihood that this datum is generated by a watermarked model.  We make several novel theoretical claims: (1) We prove statistical validity under virtually no assumptions in the watermarking setting; (2) We prove power bounds in toy models that somewhat approximate LM’s and the intuition we glean from these broadly aligns with our empirical findings.  In addition to theory, we extensively evaluate our proposed approach empirically in a very different setting from the mentioned paper.  Furthermore, while that work does mention the blessing of dimensionality, they do this only in the Gaussian setting; this is subsumed by our strong log-concavity analysis but does not approximate the LM setting as well as the linear softmax model we consider.
>
> Second, Section 3.1 discusses how our test is, heuristically, a computationally efficient approximation of the minimax optimal test, giving theoretical grounding to our approach.  In particular, GaussMark naturally emerges as the ‘right test’ for structural watermarking rather than being directly inspired by GUS. In fact, the resemblance of the two suggests that the GUS may be theoretically grounded as well, although this direction appears out of scope for that paper.
>
> TL;DR: we agree this work is relevant and will add a discussion and proper reference to  Pawelczyk et al 2025 in the revision, but we disagree that the existence of the aforementioned work diminishes the novelty claim.
>
> ## Necessity of knowing model gradients
>
> This is a good point and is acknowledged as a limitation in line 415, 2nd column.  Please see that paragraph for why we believe the white-box setting is reasonable.
>
> ## Questions
>
> Thank you for catching the typo in the caption.  Please see our response to XB7Z regarding quantization.

---

> > ### Comment · Reviewer_36HD · 2025-04-04
> >
> > I thank the authors for the rebuttal. However, some points are still not quite clear to me:
> > a) The linear softmax assumption vs. the assumption in Eqn. (3) vs. the linear softmax model in Definition 3.2.: I was referring to the assumption in eqn. 3 specifically, are they the same? Also can the authors explain in simple terms, which the validity /p-value of the test does not depend on either?
> >
> > I unfortunately don't quite understand the response regarding hyperparameter selection on the benchmarks. It seems that the authors argue that they don't want to show generalization (i.e., to other benchmarks). This would be unfortunate because I think that we would like to have a hyperparameter setting, that works for general purpose settings (e.g., and LLM behind and API).
> >
> > Regarding related work, I am aware that the unlearning setting has some differences. I nevertheless wanted to underline the technical similarities, which I think make this work an important reference and that the contribution (basically transforming it to another setting + extending some theorecal properties) should be appropriately framed as such.
> >
> > I still remain a bit sceptical about the writing, which is very dense (this also becomes clear in the reply, where the authors claim they "allude" to certain points -- apparently this was not sufficient for reader's like myself, familiar with LLM, watermarking, unlearning etc. in general, but not a statistics expert to understand these points).
> > skeptical.
> >
> > I am still willing to increase my score if the authors can provide convincing and intuitive responses for my first two points.

---

> > > ### Author Response · Authors · 2025-04-04
> > >
> > > We thank the reviewer for their engagement and further questions. Please find our response below:
> > >
> > >
> > > > Linear Softmax Assumption vs (3)
> > >
> > > To clarify, the heuristic approximation in (3) is not the same as the linear softmax assumption.  In particular, (3) is intended to heuristically motivate the proposed test as a computationally efficient, linearized approximation to the optimal test in this setting.  This approximation is reasonable whenever the variance of the noise $\xi$ is small relative to the Frobenius norm of the Hessian of the log-likelihood at parameter theta.  Note that even granting the linear softmax assumption, equation (3) remains an approximation due to the lack of the linear dependence of the normalizing factor on the parameter theta.  To be clear: (3) is introduced only to motivate our test as a linear approximation to the optimal Neyman-Pearson likelihood ratio test with arbitrary differentiable dependence on theta and this approximation is not required to hold for any of our formal results, especially not the proof of statistical validity that holds unconditionally, i.e., our p-values are correct even if these assumptions do not hold (see the proof of Proposition 3.1).
> > >
> > > The linear softmax assumption in Section 3.2 is made to give a theoretical bound on the power of the proposed test in an analytically tractable setting that sufficiently approximates transformers so as to give some intuition as to when we expect Gaussmark to be powerful in detecting watermarked text.  We again emphasize that this linear softmax assumption is only used in the power computation and is not related either to the motivation for the test (which relies on a linear approximation of the log-likelihood itself, which is a stronger condition than linear softmax due to the aforementioned normalizing factor) or the statistical validity of Gaussmark in any way.
> > >
> > > The reason the statistical validity holds without either (3) or the linear softmax assumption is fundamentally due to the rotational invariance of the Gaussian.  Under the null, $y$ and $\xi$ are independent, so, conditioning on $y$, it holds that $\langle \xi, \nabla \log p_\theta(y) \rangle \sim N(0, \sigma^2 \| \nabla \log p_\theta(y) \|^2)$; thus, dividing the inner product by the norm of the gradient and $\sigma$, we get that our test statistic, under the null hypothesis, is distributed according to $N(0, 1)$.  This holds regardless of the distribution of the $y$ and thus, for any distribution of $y$, as long as the $y$ are independent of $\xi$, we can condition on the value of $y$ and the preceding calculation holds.  The fact that the distribution of the statistic under the null is independent of the precise distribution within the null is precisely what allows us to guarantee statistical validity.
> > >
> > >
> > >
> > > > Parameter Selection and Generalization to other benchmarks
> > >
> > > We apologize for the confusion and agree that the purpose of including the benchmarks is to suggest that we expect the comparison between watermarked and base models to generalize to other notions of model quality.  To this end, we are happy to include in the revision additional results demonstrating the correlation in model performance across these benchmarks for base and watermarked models with different chosen parameters.  Such correlation can already be observed to some extent in the disaggregated SuperGLUE scores included in the supplement.  Moreover, to clarify, we picked our watermarked models based on SuperGLUE/GSM8K/p-value constraints and measured their performance on AlpacaEval, which already provides some evidence of this generalization.  We will better clarify this point in the revision.
> > >
> > > > Discussion of related work
> > >
> > >
> > > We completely agree that Pawelczyk et. al. is a relevant reference and will definitely add proper citations and discussion in the final version of our paper. From our response, we simply wish to convey that the existence of this work does not diminish the technical novelty of the current paper due to the difference in setting, additional theoretical results, and substantial empirical evaluation.
> > >
> > > > Clarity of presentation of statistical concepts
> > >
> > > We apologize if some of the technical writing is overly dense; due to space constraints we did defer many of the proofs to the appendix.  In the revision we will endeavor to give a more comprehensive introduction to the statistical concepts and proof techniques used, subject to space constraints.

---

### Official Review · Reviewer_nqFw · 2025-03-16

**Overall Recommendation:** 2

**Summary:**

The paper proposes GaussMark, a structural watermarking method for LLMs that perturbs model weights with Gaussian noise during generation. The authors claim this approach addresses limitations of token-level watermarking by embedding watermarks directly into model parameters. Detection leverages hypothesis testing on gradients of perturbed weights, with some statistical arguments.

**Claims And Evidence:**

Authors claimed that new method GaussMark: is designed to fix multiple problems of token-level watermarking, like "generation latency, detection time, degradation in text quality, or robustness". However, there are many mis-interpretation of prior token-level watermarking methods. See "Relation To Broader Scientific Literature" for details.

The theoretical claims are not rigorously proved. See "Theoretical Claims".

**Essential References Not Discussed:**

See above points for mis-interpretation of prior works.

**Experimental Designs Or Analyses:**

No issues identified.

**Methods And Evaluation Criteria:**

It makes sense.

**Other Comments Or Suggestions:**

No other comments.

**Other Strengths And Weaknesses:**

One notable strength of GaussMark is its novelty. While previous watermarking techniques often manipulate the token sampling process at the logits level, GaussMark takes a different approach by directly perturbing the model's parameters. This departure from logits-based and investigation on adding noise into model parameters manipulation is indeed novel.

Despite the novelty, GaussMark exhibits several weaknesses concerning its problem framing and theoretical foundation.

**Questions For Authors:**

Why does TPR saturate in Figure 2a?
Is it possibly because the fundamental approximation eq (3) of both theory and method became invalid in long sequence?

**Relation To Broader Scientific Literature:**

In the abstract and introduction, the authors argue that previous methods suffer from issues such as generation latency, detection time, text quality degradation, and lack of consideration for text structure. However, some of these claims regarding prior works are exaggerated or inaccurate. For example, the paper criticizes methods like Kuditipudi et al. (2023) for "extremely slow generation and detection," which is a mischaracterization. Kuditipudi's approach, based on manipulating logits and using efficient hash-based detection, is actually quite fast. Furthermore, the claim of text quality degradation in token-level
 watermarking is also misleading, e.g. "Unbiased Watermark for Large Language Models," have been developed to avoid performance drops.


The paper's also claim that prior token-level watermarking methods ignores  "inherent structure of text" and GaussMark addresses that by embedding watermarks into model weights is vague. It's not evident how this directly leverages or incorporates the inherent structure of language itself, as opposed to, for instance, semantic watermarking. On the other hand, semantic-based watermark, as seen in "Watermarking Conditional Text Generation for AI Detection: Unveiling Challenges and a Semantic-Aware Watermark Remedy", explicitly handle the language structure. Even though this paper cited some semantic watermark works, it only discuss its relationship with paraphrasing attacks, and didn't compare two approaches in terms of language structure.


Therefore, the paper's initial framing of the problem and the necessity for a fundamentally different approach like GaussMark is built upon a somewhat flawed understanding of the current state of the art.

**Theoretical Claims:**

The theoretical justification for GaussMark relies on a problematic approximation:

$\log\bigl(p\_{\theta+\xi}(y|x)/\mathbb{E}\_{\xi^{\prime}\sim\nu}\bigl[p_{\theta+\xi^{\prime}}(y|x)\bigr]\bigr)\approx\langle\xi,\nabla_\theta\log p_\theta(y\mid x)\rangle$.

Author claim that this approximation is valid only under the assumption that $\sigma \ll 1$ and that the higher-order terms, particularly $\frac{\sigma^2\left\|\nabla_\theta\log p_\theta(y\mid x)\right\|^2}{2}$, are negligible.

However, the norm of the gradient, $\lVert\nabla_\theta\log p_\theta(y\mid x)\rVert$, is substantial in realistic scenarios with long generated texts and complex, high-dimensional language models.

The paper does not adequately address the conditions under which this approximation remains valid for practical LLMs and text generation lengths.  The claim of $\sigma \ll 1$ being sufficient is likely too simplistic, and a more stringent condition, possibly dependent on text length and model complexity, like $\sigma\ll\frac{1}{L\sqrt{n}}$, might be necessary for the approximation to hold, a consideration absent in both the theory and experimental analysis, as all following bounds builds upon such approximation.

---

> ### Author Rebuttal · Authors · 2025-03-27
>
> We thank the reviewer for their careful attention to our work.  We wish to clarify a few points.
>
> ## The linear approximation
>
> We wish to emphasize that while the motivation and power bounds do rely on some simplifying assumptions, as we demonstrate in Proposition 3.1, *the statistical validity of the test holds under virtually no assumptions*, in particular not requiring the linear softmax approximation to hold.  The power bound is an important theoretical result to guide us in understanding the effect different problem parameters have on the efficacy of our test, but is not directly relevant to practice; indeed, we verify empirically that Gaussmark has relatively good power in detecting watermarked text.  The discussion following Proposition 3.3, wherein we take $\sigma \to \infty$, is meant to serve as a guide for intuition and we give a finite bound on how large $\sigma$ must be in Corollary I.5, to which we allude in line 298.  Furthermore, we believe that this intuition is reflected empirically e.g. in Figure 7, which reports p-values of detection for different choices of variance and watermarked parameter; in this figure, when the variance is high for early layers (where we expect the linear approximation to be weaker) we see a *decrease in power, manifesting through increased p-values when the variance increases*, suggesting that the lack of linear approximation is indeed reducing the power of the test.  In the revision we will include additional discussion in the main body clarifying this point.
>
> With respect to the motivation, note that the linear approximation allows us to find a computationally efficient test statistic that (if the approximation is sound) will perform approximately as well as the minimax optimal test, which is likely computationally inefficient.  We agree (and will clarify in the revision) that the quality of the linear approximation depends on the operator norm of the Hessian which implicitly grows with sequence length, but this motivation is intended as a non-rigorous motivation for a statistic we find works quite well empirically.
>
> To summarize: in the event that the linear approximation does not hold, there is no effect on statistical validity and while there may be an effect on empirical power, we demonstrate conclusively that one may choose parameters that do not suffer from this too much.
>
> ## The importance of fast generation and detection
>
> Regarding the speed of generation of watermarked text, it is important to differentiate between theoretical and empirical latency.  We agree that the approach of Kuditipudi et al is theoretically fast with respect to generation, but they do not report generation times themselves.  One key advantage of Gaussmark (cf. lines 173-175, 2nd column) is that it can be immediately integrated into preexisting generation pipelines, like vLLM that allow for substantially faster generation than more naive approaches.  The approach of Kirchenbauer et al is at least as fast in terms of generation as that of Kuditipudi et al and, as we show in Figure 27, the former is orders of magnitude slower in generation empirically (cf. Line 1406).  Furthermore, as noted in Kuditipudi et al, detection time grows quadratically in text length and their reported detection times are significantly slower than ours with much shorter texts.
>
> ## Semantic Watermarking
>
> We are happy to include additional discussion of other semantic watermarking schemes.  The paper mentioned by the reviewers is certainly relevant, but note that their results do not in any way dominate ours.  Indeed, as an example, their approach (as summarized by Table 1 in that paper) leads to substantial declines in text quality, despite improving on pre-existing work, in contradistinction to GaussMark, which we demonstrate does not lead to reduced text quality on a wide variety of benchmarks.
>
> ## Why the TPR saturates in Figure 2a
>
> This is a good question.  Note that the answer is likely *not* because the linear approximation becomes invalid for long sequences, but rather because the power calculation predicts that our test is only consistent with high dimension, not with increasing sequence length.  Were the linear approximation breaking down the cause of this phenomenon, we would expect the power of the test to potentially decrease (as we discuss above in reference to Figure 7), which we do not observe.
>
> We suspect that the power could be improved by adding noise to multiple parameters at once, as we suggest in Line 438, 2nd column; this is an interesting direction for future research.

---

### Official Review · Reviewer_xKDx · 2025-03-24

**Overall Recommendation:** 5

**Summary:**

The paper introduces GaussMark, a watermarking scheme that embeds a subtle signal into a language model by additively perturbing its weights with a small Gaussian noise. Instead of operating at the token level, GaussMark leverages the inherent structure of text by modifying a single MLP layer within the model. The method formulates watermark detection as a statistical hypothesis test, where a test statistic based on the inner product between the noise vector and the gradient of the log-likelihood is computed. The authors provide rigorous statistical guarantees (e.g., via Propositions 3.1 and 3.3) and validate the approach with extensive experiments on several modern language models using standard benchmarks (SuperGLUE, GSM–8K, and AlpacaEval–2.0). Their empirical evaluation shows that the scheme maintains high text quality and very low latency in both generation and detection.

## update after rebuttal

I have read the other reviews and the authors' responses. I think this paper meets the bar of ICML, so I recommend acceptance.

**Claims And Evidence:**

Claims:

- GaussMark is practical and efficient, with essentially no impact on generation latency.
- It provides formal statistical guarantees for detection, ensuring control over false positive rates.
- The method is robust to common token- and sequence-level corruptions and does not degrade model quality.

Evidence:
- Theoretical results (e.g., Proposition 3.1 establishes the test’s validity and Proposition 3.3 provides bounds on detection power) back up the claims.
- Experimental results demonstrate that watermarking does not harm downstream performance and that detection remains fast and reliable even under various corruptions.

**Essential References Not Discussed:**

N/A

**Experimental Designs Or Analyses:**

- Experiments are conducted on multiple language models (e.g., Llama3.1–8B, Mistral–7B, Phi3.5–Mini) with clear comparisons between watermarked and unwatermarked variants.

- The design includes measuring generation latency, detection latency, and performance on language understanding and reasoning benchmarks.

**Methods And Evaluation Criteria:**

Methodology:

- The watermark is embedded by perturbing a chosen model weight with Gaussian noise. Detection uses a normalized statistic that is compared against a threshold derived from the Gaussian CDF.

- This approach sidesteps the latency issues seen in token-level watermarking methods by integrating directly into the inference pipeline.

Evaluation:

- The evaluation is comprehensive, employing standard benchmarks (SuperGLUE, GSM–8K, AlpacaEval–2.0) and covering various aspects such as quality, speed, and robustness.

- Experiments also include ablation studies (e.g., on the effects of hyperparameter choices and rank-reduction variants) to illustrate the trade-offs between watermark strength and model performance.

**Other Comments Or Suggestions:**

See listed above.

**Other Strengths And Weaknesses:**

Strengths:

- Simplicity & Efficiency: The method is conceptually straightforward and easily integrated into existing pipelines without slowing down generation.

- Theoretical Rigor: Provides clear statistical guarantees and theoretical analyses that enhance credibility.

- Practical Impact: The approach is designed with deployment in mind, including fast detection and minimal impact on model quality.

**Questions For Authors:**

See listed above.

**Relation To Broader Scientific Literature:**

This paper leverages established ideas from statistical hypothesis testing and recent empirical findings on weight perturbations and model merging.

**Theoretical Claims:**

The theoretical claims appear sound within the stated assumptions. However, the dependence on linear approximations may limit applicability in regimes where non-linear effects are significant.

---

> ### Author Rebuttal · Authors · 2025-03-27
>
> We thank the reviewer for their careful attention to our work.  One point on the theoretical claims that we wish to clarify is that the statistical validity of our test does not depend on the linear approximations being sound and so the test’s statistical significance holds under virtually no assumptions.  It is only the power of the test for which we require linear approximations (or local strong log-concavity, cf. the paragraph beginning in line 300) to justify theoretically.  Note that the bound on the power of the test is not necessary for practical deployment as long as the true positive rate of detection is relatively high at a fixed false positive rate (as we verify empirically).

---

### Decision · Program_Chairs · 2025-05-01

**Decision:**

Accept (poster)

**Comment:**

This paper introduces GaussMark, which is a new approach for watermarking LLMs. The main algorithmic idea of adding Gaussian noise to the input is simple but natural, the technical analysis that relies on a Gaussian independence test is highly trivial. Extensive experiments further demonstrate the applicability of this new approach.

After the rebuttal, most reviewers are pleased with this work. Therefore, I am happy to recommend its acceptance. We encourage the authors to incorporate the discussions and the promised revisions into the final version of this paper.